# Evaluation of ambient ammonia measurements from a research aircraft using a closed-path QC-TILDAS spectrometer operated with active continuous passivation

Ilana B. Pollack[1], Jakob Lindaas[1], J. Robert Roscioli[2], Michael Agnese[2], Wade Permar[3], Lu Hu[3], and Emily V. Fischer[1]

[1]Department of Atmospheric Science, Colorado State University, Fort Collins, Colorado, 80523, USA
[2]Aerodyne Research Inc., Billerica, Massachusetts, 01821, USA
[3]Department of Chemistry and Biochemistry, University of Montana, Missoula, Montana 59812, USA

*Correspondence to*: Ilana B. Pollack (ipollack@rams.colostate.edu) or Emily V. Fischer (evf@rams.colostate.edu)

**Abstract.** A closed-path quantum cascade tunable infrared laser direct absorption spectrometer (QC-TILDAS) was outfitted with an inertial inlet for filter-less separation of particles and several custom-designed components including an aircraft inlet, a vibration isolation mounting plate, and a system for optionally adding active continuous passivation for gas-phase measurements of ammonia ($NH_3$) from a research aircraft. The instrument was then deployed on the NSF/NCAR C-130 aircraft during research flights and test flights associated with the Western wildfire Experiment for Cloud chemistry, Aerosol absorption and Nitrogen (WE-CAN) field campaign. The instrument was configured to measure large, rapid gradients in gas-phase $NH_3$, over a range of altitudes, in smoke (e.g., ash and particles), in the boundary layer (e.g., during turbulence and turns), in clouds, and in a hot aircraft cabin (e.g., average aircraft cabin temperatures expected to exceed 30 ℃ during summer deployments). Important design goals were to minimize motion sensitivity, maintain a reasonable detection limit, and minimize $NH_3$ "stickiness" on sampling surfaces to maintain fast time response in flight. The observations indicate that addition of a high frequency vibration to the laser objective in the QC-TILDAS and mounting the QC-TILDAS on a custom-designed vibration isolation plate were successful in minimizing motion sensitivity of the instrument during flight. Allan variance analyses indicate that the in-flight precision of the instrument is 60 ppt at 1 Hz corresponding to a $3\sigma$ detection limit of 180 ppt. Zero signals span ±200 pptv, or 400 pptv total, with cabin pressure and temperature and altitude in flight. The option for active continuous passivation of the sample flow path with 1H,1H-perfluorooctylamine, a strong perfluorinated base, prevented adsorption of both water and basic species to instrument sampling surfaces. Characterization of the time response in flight and on the ground showed that adding passivant to a "clean" instrument system had little impact on the time response. In contrast, passivant addition greatly improved the time response when sampling surfaces became contaminated prior to a test flight. The observations further show that passivant addition can be used to maintain a rapid response for *in-situ* $NH_3$ measurements over the duration of an airborne field campaign (e.g., ~2 months) since passivant addition also helps to prevent future build-up of water and basic species on instrument sampling surfaces. Therefore, we recommend the use of active continuous passivation with closed-path $NH_3$ instruments when rapid (> 1 Hz) collection of $NH_3$ is important for the scientific objective of a field campaign (e.g., sampling from aircraft or another mobile research platform). Passivant addition can be useful for maintaining optimum operation and data collection in $NH_3$-rich/humid environments or when contamination of sampling surfaces is likely, yet frequent cleaning is not possible. Passivant addition may not be necessary for fast operation, even in polluted environments, if sampling surfaces can be cleaned when the time response has degraded.

## 1 Introduction

Ammonia ($NH_3$) is the dominant alkaline gas in the atmosphere and plays an important role in many atmospheric processes. Major sources of atmospheric $NH_3$ include agricultural activities (e.g., application of fertilizer and volatilization from animal

wastes) (e.g., (Galloway et al., 2003; Pinder et al., 2007; Reis et al., 2009; Erisman et al., 2008; Balasubramanian et al., 2015; Leen et al., 2013), light duty gasoline vehicles equipped with three-way catalytic converters (e.g., (Kean et al., 2009; Burgard et al., 2006), biomass burning (e.g., (Hegg et al., 1988; Bray et al., 2018), water and sewage treatment plants, and some industrial production activities (e.g., chemical production plants (Zhu et al., 2015)). Atmospheric reactions of $NH_3$ with

acids formed from the oxidation of sulfur dioxide ($SO_2$) and nitrogen oxides ($NO_x = NO + NO_2$) can lead to formation of fine particulate matter (Behera and Sharma, 2010; Fenn et al., 2003), which has strong implications for human health, regional air quality, atmospheric visibility, radiative forcing, and nitrogen deposition in sensitive ecosystems (Pope, 2002; Zhu et al., 2015; Erisman et al., 2008; Asman et al., 1998; Krupa, 2003; IPCC, 2007).

Anthropogenic $NH_3$ emissions are becoming increasingly important to study due to intensification of agricultural activities and animal husbandry (e.g., concentrated animal feeding operations) (Galloway et al., 2008).  While $NH_3$ is regulated under the Gothenburg protocol in some parts of the world (e.g., http://www.unece.org/environmental-policy/conventions/air/guidance-documents-and-other-methodological-materials/gothenburg-protocol.html), it remains an unregulated pollutant in the U.S. (Gilliland et al., 2008). Having instruments that can collect high-sensitivity, fast-response

*in-situ* measurements of $NH_3$ are essential for directly measuring $NH_3$ emissions fluxes (e.g., from animal husbandry, agricultural fertilization) and eddy covariance fluxes (e.g., associated with deposition/evaporation processes), characterizing concentrations and emissions rates in plumes (e.g., from urban areas with emissions dominated by traffic, concentrated animal feeding operations, and wildfires), and sampling from mobile platforms (e.g., instrumented aircraft and ground-based vehicles). There are several techniques and types of instruments that can be used for rapid measurements of atmospheric

$NH_3$, including mass spectrometric methods (e.g., (Nowak et al., 2007)) and optical methods based upon open path absorption (e.g., (Miller et al., 2014; Ni et al., 2015)), closed path absorption (e.g., (Griffith and Galle, 2000; Ellis et al., 2010; Leen et al, 2013; Martin et al., 2016; Leifer et al., 2017)), and photoacoustic spectroscopy (e.g., (Schmohl et al., 2002; Pushkarsky, et al., 2002; Pogány et al., 2009)). The mass spectrometric method has been effectively leveraged aboard research aircraft (Nowak et al., 2007; Nowak et al., 2010), and the compact footprint associated with the photoacoustic

approaches are useful for many applications. Open and closed path direct absorption approaches are highly applicable to the $NH_3$ concentration ranges expected during ambient monitoring. Open-path instruments typically have lower power consumption, higher data collection rates, and no time delays or sampling surface interactions due to inlet tubing (e.g. (Miller et al., 2014)). Closed-path systems afford the advantage of minimal data loss when sampling in potentially high particle/aerosol conditions such as in dust, smoke, precipitation/icing, and salt deposition events (e.g., (Sun et al., 2015; Leen

et al., 2013)), and allow for more control over environmental influences (e.g., temperature, pressure, and water vapor). They are also able to be directly zeroed and calibrated during operation.  However, closed path systems typically rely on inlet tubing to supply ambient air to the instrument, and thus the effects of inlet tubing length (and inlet complexity, especially when deployed aboard airborne research platforms) on the instrument time response can be significant. To add to these existing challenges, $NH_3$ is notorious for being a "sticky" molecule. Its ability to readily adsorb and/or desorb from sampling

surfaces makes it a difficult gas-phase species for which to measure large, rapid changes and it is a particularly difficult measurement for which to determine accurate *in-situ* background, or zero, levels. Recent laboratory studies showed dramatic improvements in $NH_3$ transmission through a commercially-available, closed-path spectrometer, and thus dramatic improvements in measurement time response, when the instrument's sampling surfaces were actively and continuously passivated with a strong perfluorinated base (e.g., 1H,1H-perfluorooctylamine) (Roscioli et al., 2016). The passivant coating

works by extending a nonpolar chemical group into the sample flow path that prevents the adsorption of both water and basic species to the sampling surfaces, yet the passivant chemical does not interfere with $NH_3$ detection due to the highly-specific nature of detection.

    Here we describe the first opportunities to evaluate and carefully characterize the effects of adding passivant to an optical

absorption based, closed-path gas-phase $NH_3$ monitoring system aboard a research aircraft. We note that although this is not the first aircraft deployment of an optical-based $NH_3$ monitor (Leen et al, 2013; Hacker et al., 2016; Schiferl et al, 2016; Miller et al., 2015; Leifer et al., 2017), this is the first opportunity to evaluate the effects and the value of adding a passivant to the sample stream of an optical-based instrument for airborne measurement applications. We start by characterizing the performance of the non-passivated instrument aboard a research aircraft in flight (e.g., precision, detection limit, motion

sensitivity, and stability over time) with respect to fluctuations in cabin pressure, cabin temperature, and changes in altitude.

We then evaluate the instrument time response with and without active continuous passivation under a variety of operating conditions. We report several different methods for zeroing the instrument and offer recommendations for each method based on the sampling environment. Lastly, we provide recommendations for using active passivation for atmospheric measurements of $NH_3$ with optical-based, closed-path instrumentation in a variety of field measurement scenarios and environmental conditions.

## 2 Methods

### 2.1 Airborne sampling

Airborne measurements were collected aboard the NSF/NCAR C-130 research aircraft during the Western wildfire Experiment for Cloud chemistry, Aerosol absorption and Nitrogen (WE-CAN) field campaign in summer 2018 and during 16 test-flight hours prior to the WE-CAN deployment. The aircraft conducted seventeen research flights of roughly 6-8 hour duration between 20 July and 31 August in 2018, three test flights of 2-3 hour duration between 21 September and 29 September in 2017, and two test flights on 13 July and 17 July in 2018. Research flights were conducted in the northwestern U.S. with aircraft operations based out of Boise, Idaho; test flights were conducted in the northeastern Colorado Front Range based out of the National Center for Atmospheric Research (NCAR) Research Aviation Facility in Broomfield, Colorado. The 2018 test and research flights were conducted under average ambient temperature and humidity conditions expected for summer in Idaho and Colorado; 2017 test flights were performed under lower than average ambient temperature (e.g., average ambient temperature was 12 ℃) and higher than average relative humidity (e.g., the average relative humidity was >70%) conditions for Colorado. WE-CAN research flights provided a number of opportunities to evaluate multiple aspects of the $NH_3$ instrument in flight and within concentrated smoke plumes. Several missed approaches performed at Greeley-Weld County Airport during the test flight period provided several opportunities to characterize the instrument's time response with and without passivant under a variety of instrument operating conditions. Aircraft maneuvers were performed during several of the test flights and are used here to assess instrument precision, detection limit, and motion sensitivity in flight. Changes in instrument zero signal level with cabin pressure, cabin temperature and changes in altitude were extensively tested during the test flights by overflowing the inlet tip with $NH_3$-free air for large periods of the flight. The aircraft also often sampled ambient air in the free troposphere during the test flights and when in transit to wildfires during the 2018 deployment; these measurements are used for evaluating different methods for zeroing the instrument, ambient $NH_3$ levels in the free troposphere, and the instrument detection limit.

### 2.2 Instrumentation

The flight-ready, closed-path, optical-based $NH_3$ monitoring system described here consists of a combination of commercially-available and custom-built components including: 1) a commercially-available infrared absorption spectrometer that serves as the heart of the $NH_3$ monitor, 2) a commercially-available inertial inlet that acts as a filter-less separator of particles from the sample stream, 3) a custom-built aircraft inlet, 4) a custom-designed vibration isolation mounting system for the spectrometer, and 5) an optional system for adding passivant to the sample stream.

### 2.2.1 NH₃ detection

These experiments utilized a compact, single-channel, closed-path, quantum-cascade tunable infrared laser direct absorption spectrometer (QC-TILDAS), model TILDAS-CS, for measuring $NH_3$. The QC-TILDAS is commercially available from Aerodyne Research Inc., and is described in detail in the literature (McManus et al., 1995; McManus et al., 2007; McManus et al., 2010; Zahniser et al., 1995). Briefly, a high sample flow rate (e.g., >10 standard liters per minute, SLPM, with standard conditions defined as 760 Torr and 0˚C) is drawn through a 76-m multipass absorption cell into which the output of a single-mode quantum cascade laser is coupled. The optical output of the laser is swept across an $NH_3$ absorption feature located at 967.34634 cm$^{-1}$. This strong rotational-vibrational (ro-vibrational) absorption feature is within the Q-branch of the $v_2$ band of $NH_3$. The instrumental linewidth is typically 0.012 cm$^{-1}$ (360 MHz) FWHM and is largely defined by the

operating pressure of the absorption cell, which was held constant at 40 Torr for these experiments. The laser is scanned across $0.315$ cm$^{-1}$, and the resulting absorption features are detected on fast mercury cadmium telluride (MCT) detector (Vigo System). The pressure- and doppler-broadened spectral peaks are fit using a Voigt lineshape model. The QC-TILDAS is capable of up to 10 Hz collection of absolute $NH_3$ concentrations. $NH_3$ mixing ratios were typically collected at 10 Hz, and subsequently averaged to 1-Hz for WE-CAN research flights and for selected analyses described below.

Figure 1a is a schematic of the $NH_3$ monitoring system as it was configured for use on the NSF/NCAR C-130 aircraft. With WE-CAN objectives in mind, the instrument was configured to measure $NH_3$ over a range of altitudes, in smoke (e.g., large and rapid gradients in $NH_3$ and in ash and particle-rich air), while systematically following a smoke plume in the mixed boundary layer (e.g., in turbulence with frequent turns), and during summer near Boise, ID (e.g., large, rapid gradients in cabin temperature and average temperatures often exceeding 30 ºC). In ground-based field studies and laboratory experiments, the optical bench of the QC-TILDAS is ideally operated between 20 and 25˚C for maximum stability of the laser power, optical alignment, and spectroscopic absorption signal. In anticipation of cabin temperatures exceeding 30˚C, the set point for the QC-TILDAS optical bench temperature was intentionally set to the upper end of this range at 25˚C.

Further, the instrument flow path was purged with ultrahigh purity (UHP) $N_2$ overnight and when there was no power or access to the aircraft in order to keep the sampling surfaces as clean as possible. Previous experiments (Nowak et al., 2007) found that the rise/decay characteristics of $NH_3$ instruments operated during aircraft missions (e.g., running for several hours, sitting idle overnight, and running again the next day) were only reproducible when a flow of clean, dry $N_2$ was used to purge the inlet during periods of instrument inactivity. Therefore, anytime the instrument is powered off, a purge flow of 40 standard centimetres per minute (sccm) of UHP $N_2$ is introduced just upstream of the pressure control valve to flush the instrument in the reverse direction of the sample flow path, as indicated in Fig. 1a. We found that the instrument could reach a stable zero signal level more quickly when it had been purged overnight compared to times when the instrument sat idle without purge flow. The instrument response time could also be maintained for a longer operational time period (e.g. weeks to months) between cleanings when the instrument flow path was purged between uses.

**2.2.2 Inertial Inlet**

The $NH_3$ QC-TILDAS is typically operated with a heated inertial inlet positioned upstream of the spectrometer to provide filter-less separation of particles >300 nm from the sample stream, as shown in Fig. 1a. Coupling an inertial inlet with a QC-TILDAS has been well established following several laboratory and ground-based field experiments (Ellis et al., 2010; Ferrara et al., 2012; Tevlin et al., 2017; von Bobrutzki et al., 2010; Zöll et al., 2016). The inertial inlet is described in detail by Ellis et al. (2010) and Roscioli et al. (2016). Briefly, the inertial inlet used in these experiments consists of a quartz tube (12.7 mm o.d., 10.4 mm i.d.) with an integral, conical-shaped critical orifice roughly 1 mm in diameter positioned at about half the length of the tube, as shown in Fig. 1a. After passing through the orifice, gas (and particulates) are accelerated to a higher speed at a lower pressure (between 40 and 100 Torr) through the latter half of the 12.7 mm quartz tube, and then pass into a second quartz tube (25.2 mm o.d., 22.2 mm i.d.) that is sleeved around the 12.7 mm tube. The sample flow is split into two branches with approximately 90% of the total flow through the critical orifice (denoted by the blue arrow in Fig. 1a) being forced to make an 180˚ turn around the edge of the 12.7 mm tubing to continue to the spectrometer, and the other 10% (denoted by the orange arrow in Fig. 1a) being dumped via the straight section of 25.2 mm tube into the main pumping system. The inertia of particles with aerodynamic diameters greater than ~300 nm is too large to follow the gas stream around the 180˚ turn, thereby forcing the particles into the 10% of the flow stream that is directed to the pumping system. Ellis et al. (2010) reported that the inertial inlet, which acts like a form of virtual impactor, removes more than 50% of particles larger than 300 nm. A tee positioned immediately upstream of the critical orifice allows for pressure measurements using a baraton transducer (range 0-1000 Torr), which is used in determining the sample flow rate, and an auxiliary draw that allows the dead volume around the base of the conical-shaped critical orifice to be actively flushed. The flow rate of the auxiliary draw ranges from 160 to 500 sccm with changes in ambient pressure at the inlet tip. The inertial inlet is housed in a fiberglass enclosure, with the inside of the enclosure maintained at 40°C."

**2.2.3 Aircraft Inlet**

A heated aircraft inlet constructed of perfluoroalkoxy fluoropolymer (PFA) allows for maximum transmission of $NH_3$ from the inlet tip to the spectrometer, as shown in Fig. 1a. Inlet components are housed inside a 36 cm long stainless steel strut that extends beyond the boundary layer of the aircraft, and consists of: 1) an inlet tip made of 6 cm long 3/8" o.d., ¼" i.d. PFA tubing, 2) a machined PFA block serving as an injection manifold for calibration and passivation gases, and 3) a 24 cm length of 3/8" o.d., ¼" i.d. PFA tubing leading down the length of the strut. Another 71 cm length of 3/8" o.d., ¼" i.d. PFA tubing directs the sample flow from the base of the inlet strut to the enclosure containing the inertial inlet, which is mounted to an equipment rack inside the aircraft cabin. The inlet tip is designed to be mounted with a standard stainless steel, bored-through compression fittings from the inside of the strut to minimize protrusions from the inlet end cap and to maintain the inlet tip at 35˚C in flight to prevent it from freezing. The tip of the PFA inlet tubing was cut with a slight rear-facing bias from the direction of flight and extended only ~6 mm from the face of the inlet end cap to minimize particle ingestion and disruption of the boundary layer near the inlet tip. The PFA injection block located just inside the inlet strut end cap has outer dimensions of 8.3 cm long x 3.6 cm wide x 1.5 cm thick and a 6-cm long ¼" i.d. inner sample channel to match the i.d. of the tubing used for the inlet tip and sample line. Aircraft inlets of similar design and construction were used during the 2014 Front Range Air Pollution Experiment (FRAPPE) aboard the C-130 aircraft and during the 2017 Utah Winter Fine Particulate Study aboard the NOAA Twin Otter. Another 36 cm long segment of 3/8" o.d., ¼" i.d. PFA tubing then brings sample flow from the inertial inlet box to the QC-TILDAS, which is co-located in the same equipment rack inside the aircraft cabin. As shown in previous studies, PFA tubing and fittings are used wherever possible along the sample pathway, and tubing lengths are kept to a minimum and heated wherever possible (Neuman et al, 1999; Schmohl et al., 2001; Mukhtar et al., 2003; Leifer et al., 2017). Components housed within the aircraft inlet strut are heated to 40˚C. The tubing between the inlet strut and the inertial inlet and between the inertial inlet and the QC-TILDAS are not actively heated; however, they are wrapped in flame-resistant polymer felt (DuPont Nomex) for thermal isolation. Although several reports specifically highlight the benefits of heated inlet components for measurements of $NH_3$ (Ellis et al., 2010; Nowak et al., 2007; Tevlin et al., 2017), these sections of tubing were left unheated in order to reduce the instrument's power load on the aircraft and because the residence time in these segments of tubing is short (e.g., <0.15 s due to the high sample flow rate and low pressure) and aircraft cabin temperatures were anticipated to exceed 30˚C in flight.

### 2.2.4 Vibration isolation system

To reduce motion sensitivity in-flight, the QC-TILDAS was mounted on a custom-designed, vibrationally-isolated plate (e.g., Fig. 1b) before being mounted in the aircraft equipment rack as shown in Fig. 1c. The mounting plate consisted of eight total vertically mounted wire rope vibration isolators (Enidine) that allow the spectrometer enclosure (containing the optical cell, laser, MCT detector, and optical bench) to float in all three dimensions and remain isolated from direct contact with the aircraft equipment rack. Two isolators (Enidine, WR3 series) were mounted along the inner face of each of the fore-aft facing legs of the mounting plate frame and spaced 13 cm apart; two additional isolators (Enidine, WR4 series) were positioned on top of the outboard and inboard legs of the mounting plate frame base and spaced 22.5 cm apart. The QC-TILDAS is then affixed to the mounting frame via the isolators as indicated in Fig. 1b and 1c. The advantage of this "springed" mounting plate is that it mitigates the effects of high frequency vibrations endemic to the aircraft, which can result in acoustic noise in the instrument, uncontrolled vibration of the optical bench, and general misalignment of the optics. In addition, it relieves strain on the instrument chassis when the frame of the aircraft equipment rack flexes during aircraft maneuvers. Hard stops were added to the mounting plate frame to satisfy aircraft crash loads in the forward and aft directions. Hardware was selected based on structural analysis calculations including finite element analysis given the vibrations and the loads expected aboard the NSF/NCAR C-130 aircraft. The mounting plate is similar in design to previous vibration isolation apparatuses used on research aircraft with Quantum Cascade Laser Systems (QCLS) (B. Daube, Personal Communication, 2017); although, this specific mounting system was unique to the $NH_3$ instrument deployed during the 2017 and 2018 WE-CAN flights. Additionally, the laser objective, an optic that guides the laser beam into the optical cavity, is vibrated at a high-frequency (~200 Hz) to wash out etalon fringe motion induced by aircraft accelerations. The frequency and amplitude of the vibrations applied to the laser objective can be adjusted to accommodate a variety of moving platforms. Several months prior to installation on the C-130 aircraft, a cabled tilting system (located at Aerodyne in Billerica, MA) was used to simulate the effects of in-flight forces on the QC-TILDAS as it was mounted to the vibration isolation plate within the aircraft equipment rack. Additional tests were performed on the hangar floor in Broomfield, CO immediately before installation of the instrument aboard the aircraft by manually tipping and shaking the equipment rack. The "tilt and shake" tests performed in both locations were conducted with the instrument powered on and operating under "zero" measure conditions by overblowing the sample inlet port with $NH_3$-free air. The center frequency and frequency sweep of the piezoelectric stack mounted on the laser collection objective as well as the general optical alignment within the enclosure were optimized during these tests. From these tests, we additionally learned that external forces acting on the inlet and outlet tubing associated with the sample stream were putting strain on the optical bench and resulting in notable deviations from zero in the $NH_3$ absorption signal. However, this motion sensitivity could be minimized by keeping tubing lengths to a minimum and reinforcing the strain relief of the sample tubing connected to the QC-TILDAS enclosure inlet and outlet ports (e.g., rigidly securing all flexible tubing to the frame of equipment rack with cable ties) prior to installation on the aircraft.

### 2.2.5 Passivant addition system

Owing to the "stickiness" of $NH_3$, the QC-TILDAS and inlet sampling surfaces can be compromised when they are coated with as little as a single monolayer of adsorptive matter; the build-up can cause the instrument's time response to gradually become slower (Roscioli et al., 2016). Although sampling surfaces can be periodically refreshed by cleaning them with solvents, frequent cleaning may not always be possible or practical during field intensives. One solution, recommended by Roscioli et al. (2016), is to actively and continuously passivate the instrument sampling surfaces with a chemical coating that prevents adsorption of water and basic species. Therefore, we designed and deployed a non-commercial system that allowed for the option of passivant addition to the sample stream to this $NH_3$ instrument system. This work reports the first-time application of adding passivant to a closed-path, optical-based $NH_3$ instrument aboard a research aircraft, and serves as an evaluation of the flight-ready instrument's time response on the ground and in-flight under a variety of conditions and with and without passivant.

For these tests, the flight instrument is outfitted with an option for adding 1H,1H-perflurooctylamine ($C_8H_4F_{15}N$, CAS Number: 307-29-9) vapor to the sample stream using a similar apparatus as that used in laboratory experiments by Roscioli et al. (2016). 1H,1H-perflurooctylamine (purchased from Synquest Laboratories and used without further purification) is a

liquid at room temperature and pressure (e.g., 20°C and 760 Torr). Vapors of the passivant are entrained in a 200 sccm flow of UHP $N_2$ regulated with a mass flow controller (Alicat) and introduced to the sample flow path as close to the inlet tip as possible. Given the configuration for WE-CAN, this means that the C8 passivant is injected roughly 10 cm downstream of the aircraft inlet tip via the middle port of the PFA injection block as shown in Fig. 1a. The liquid passivant is contained in a PFA impinger (Savillex), and a pair of stainless steel quick connects (Swagelok) are used to connect the impinger inline between the mass flow controller and the injection port (e.g., Fig 1d). $N_2$ carrier gas was intentionally not bubbled through the liquid to avoid splattering the liquid onto the impinger walls and/or lodging droplets of passivant in the delivery tubing. When disconnected, the quick connects isolate the supply of passivant from the flow path allowing it to be safely removed from the aircraft overnight and for refilling. Tests without passivation could also be easily performed simply by disconnecting the quick-connects from the impinger and re-connecting them to each other without the impinger in line. Bypassing the impinger in this manner allows for constant dilution of the sample stream by a known amount of $N_2$ carrier gas flow regardless of whether the passivant chemical is being added. For future applications, a set of solenoid valves can be added to the impinger system for automated computer control of passivant addition or passivant bypass. For the laboratory experiments and WE-CAN flights described here and with the passivant liquid held near room temperature (e.g., 25°C), the typically usage rate of the C8 compound was ~5 grams in 20 hours.

A C7 version of the passivant chemical, 1H,1H-perfluroheptylamine ($C_7H_4F_{13}N$, CAS Number: 423-49-4), was used in a separate set of tests performed in the laboratory between the 2017 and 2018 flight period, and is discussed further in Sect. 4.3. Disadvantages of adding passivant to the sample stream for $NH_3$ measurements include the use of hazardous materials and the cost of consumable chemicals. Both the C8 compound (1H,1H-perfluorooctylamine) and the C7 version (1H,1H-perfluoroheptylamine) are strong corrosives, contain an amine group, and are highly fluorinated. While these chemicals pose no immediate danger when properly handled, the long-term exposure effects are unknown. In addition, these compounds have been identified as potentially potent, long-lived greenhouse gases (Hong et al., 2013); although, we anticipate the environmental impacts to be minor given the small quantities and low addition rates used for this application.

**2.3 Calibration**

The PFA injection block is configured such that calibration and passivation gases can be introduced to the sample flow path within 6-12 cm of the inlet tip (e.g., Fig. 1a). The QC-TILDAS is calibrated via standard addition to the sample stream with a known concentration of $NH_3$ generated from a temperature-regulated ($40 \pm 0.1$°C) permeation device filled with anhydrous $NH_3$ (Kin-Tek), and zeroed by overflowing the inlet tip with a source of $NH_3$-free air. As described in detail in Ellis et al. (2010), three-way switching solenoid valves and a vacuum manifold are used to actively flush zero and calibration gases from the injection tubing when measuring ambient air. A continuous flow of UHP $N_2$ at 40 sccm was sufficient to transport all of the $NH_3$ vapor emitted from the permeation device to the inlet tip. The stability of the permeation device was maintained overnight when there was no power and access aboard the aircraft by removing it to a laboratory where it could be kept heated and under a constant (40 sccm) flow of UHP $N_2$. The emission rate of the permeation device was calibrated before and after the test flight period using the NOAA ultraviolet (UV) optical absorption system (Neuman et al., 2003). The average emission rate measured with the NOAA system before and after the WE-CAN deployment period was $407 \pm 10$ ng min$^{-1}$. The NOAA calibration has a reported uncertainty of $\pm10\%$, which is mainly due to the uncertainty in the 185 nm absorption cross section for $NH_3$ used for interpreting results from the optical absorption system (Neuman et al., 2003). In this work, we refine the uncertainty of the NOAA calibration of the emission rate of the permeation device used in these experiments by utilizing more recent assessments of the $NH_3$ absorption cross section reported in the literature. Here, we use a weighted average of the $NH_3$ absorption cross sections reported by Froyd and Lovejoy (2012) ($4.67 \pm 0.08 \times 10^{-18}$ cm$^2$), Chen et al. (1998) ($4.7 \pm 0.5 \times 10^{-18}$ cm$^2$) and Cheng et al. (2006) ($4.7 \pm 0.5 \times 10^{-18}$ cm$^2$). The weighted mean utilized here ($4.7 \pm 0.1 \times 10^{-18}$ cm$^2$) is in agreement within the uncertainties with the value reported by Neuman et al. (2003) (e.g., $4.4 \pm 0.3 \times 10^{-18}$ cm$^2$). Combining in quadrature the $\pm2\%$ uncertainty associated with the weighted mean of the absorption cross section, the $\pm2.5\%$ uncertainty in the stability of the permeation device between pre- and post-project calibrations with the NOAA UV optical absorption system, and a conservative estimate of $\pm6\%$ for other sources of uncertainty associated with the NOAA calibration system, we determine a total estimated uncertainty of $\pm7\%$ for the emission rate of the permeation device used in these experiments. The permeation rate of the $NH_3$ permeation device was specifically selected for $NH_3$

concentrations expected while sampling concentrated plumes of wildfire smoke during WE-CAN (e.g., mixing ratios $\geq 50$ ppbv). During flight, the instrument sample flow rate varies with altitude due to changes in ambient pressure upstream of the critical orifice inside the quartz inertial inlet. As a result, calibration concentrations range from 50 ppbv at 620 Torr (e.g., on the ground in Broomfield, CO at 1.729 m AMSL (above mean seal level)) to 100 ppbv at 310 Torr (e.g., near 7.4 km AMSL). Since the orifice inside the inertial inlet is truly critical, the in-flight sample flow rate can be calculated as $F_{alt} = F_{grd}*(P_{alt}/P_{grd})$ from a pre-flight measurement of pressure and sample flow on the ground using a primary flow calibration unit (DryCal Definer 220) and the pressure measured in flight just upstream of the critical orifice inside the inertial inlet using a 0-1000 Torr Baratron pressure transducer (MKS Instruments, model 722B), as shown in Fig. 1a.

## 2.4 Power, weight and space

The instrument system described above in the configuration that it was utilized aboard the C-130 aircraft requires the space of an entire NSF/NCAR G-V aircraft equipment rack (approximate dimensions 21.5" W x 28" D x 50" H). The equipment without the rack weighed approximately 150 kg and included a 30 kg uninterruptable power supply (UPS) and a 10 kg display laptop. The total power used by the instrument system was 1600 watts, with roughly one third of this total (600 watts) being dedicated to the main pumping system (Agilent, model Triscroll 600, 100 lbs installed). It is possible that the power, weight and space required for this instrument system can be reduced for future deployments by eliminating the UPS and display laptop. It may also be possible to reduce the size of the pump if different field applications allow for a lower sample flow rate to be used.

## 3 Methods for measuring instrument zero

Ellis et al. (2010) recommended that background checks of the $NH_3$ instrument would optimally be performed by removing $NH_3$ from ambient air while keeping the humidity level constant. Historically, ambient $NH_3$ monitors have been zeroed in several ways, including overblowing the inlet tip with dry synthetic air or UHP $N_2$ from a cylinder (Nowak et al., 2010; von Bobrutzki et al., 2010), overblowing the inlet tip with chemically scrubbed ambient air sources (Nowak et al., 2007; Nowak et al., 2006; Fehsenfeld et al., 2002), sampling through oxalic acid coated filters (Norman et al., 2009), or passing ambient air through heated metal catalysts (Norman et al., 2007; Tevlin et al., 2017). However, significant effort and cost can be required to routinely generate a large enough supply of $NH_3$-free air to overblow the inlet of the $NH_3$ instrument described here (e.g., >10 SLPM), and at low enough zero levels to be considered truly $NH_3$-free by instruments with low detection limits of a few hundred pptv or less. Our evaluation and recommendations reported below take these factors into consideration.

In the months leading up to the 2017 test flights, we tested five different sources of $NH_3$-free air in the laboratory. First, we used dry, ultrapure synthetic "zero" air (UZA) from a compressed gas cylinder (AIRGAS). We assert that this bottled source of UZA provides a measure of the "true" instrument zero to which we then compared all other tested sources. Second, we tested the output of a commercial zero air generator (ZAG) (Teledyne, model 701H). Third, we tested a chemical $NH_3$ scrubbing system that included a compressor pump (KNF Neuberger) for pushing ambient air through a single all-metal trap filled with an $NH_3$ scrubbing reagent (Permapure). Fourth, we tested 4 Å molecular sieve (Delta Absorbents). And, fifth, we combined the chemical scrubber in tandem with the 4 Å molecular sieve. Scrubber materials were contained individually in separate traps constructed from KF-40 stainless steel tubing and vacuum fittings (LDS Vacuum). The endcaps were outfitted with stainless steel mesh screens to prevent solid scrubber materials from migrating towards the instrument inlet and the spectrometer. Traps were warmed from the outside using heating tape (Omega Engineering) for selected experiments. Traps were cleaned prior to being filled with scrubbing media by rinsing the surfaces with water and then ethanol before baking the empty housings and fittings overnight at 250 ˚C.

We found UZA cylinders to provide the lowest and most reliable zero measure. The ZAG was also able to achieve a zero signal level consistent with that measured from a UZA cylinder. Although not ideal for use in flight owing to its weight, space and power needs, the ZAG is a plentiful and cost-effective source of $NH_3$-free air for laboratory and ground-based

field experiments. We also found the ZAG useful for pre-flight operations exceeding 3 hours and for ground-based maintenance days aboard the aircraft, so that consumable cylinders could be conserved. For the chemical scrubber, we elected to use a hygroscopic phosphoric acid scrubbing reagent (Permapure) that does an acid-base neutralization reaction to remove gas-phase $NH_3$ from an ambient air sample, as this product had been found to be successful for generating a source

of $NH_3$-free air in previous airborne field campaigns (Nowak et al., 2007). However, we found that this particular chemical scrubbing reagent could not achieve a true zero on its own. The lowest possible zero level achieved with a fresh trap of the chemical scrubber was 200 pptv above true, which is larger than the on-ground detection limit (100 pptv, see Sect. 3.1) and on par with the in-flight detection limit (200 pptv). The $NH_3$ signal from the chemical scrubber increased with usage and was closer to 1 ppbv after several days. Cleaning the trap and refreshing the scrubbing media reagent did not result in any

improvement. It is possible that the volume of the chemical scrubber trap was not large enough to scrub all of the $NH_3$ from the supply of ambient air required to overflow the instrument. However, similar offsets above true zero were observed in a separate experiment where the chemical scrubber was supplied with UZA from a cylinder instead of ambient air via the compressor, suggesting that the scrubbing reagent actually outgassed small amounts of $NH_3$. An independently purchased supply of this chemical scrubbing media produced similar results confirming that the original reagent material had not been

compromised prior to these tests. In a fourth test, we used 4 Å molecular sieve to remove $NH_3$ from an ambient air sample. While this was able to achieve a true zero, the volume of material that could be used and the flow rate required to overflow the instrument inlet limited the lifetime of the trap. The lifetime of this trap could be extended by increasing surface area contact time with the absorbing material, which could be achieved by increasing the length of the trap or by linking multiple traps in series. A disadvantage of 4 Å molecular sieve is that it also absorbs water. This means that the operational lifetime of

the molecular sieve is greatly limited in humid environments, and since the molecular sieve dries the $NH_3$-free air source the output is no different than that of a bottled source of UZA or the ZAG. In a fifth experiment, the chemical scrubber was used to remove the bulk of $NH_3$ from ambient air and followed in series by a trap filled with 4 Å molecular sieve to remove any remaining $NH_3$. The combined trap was successful in achieving a true zero, and the lifetime of the trap was noticeably longer (2-4 hours) than that of molecular sieve alone (1-2 hours).

Although the chemical scrubber (operated with or without the molecular sieve) may be the closest option to a source of $NH_3$-free ambient air, the $NH_3$ scrubbing reagent is hygroscopic and needs to be carefully monitored for condensate accumulation even when heated. Heavy contamination of the instrument sampling surfaces is possible from species outgassing from the chemical scrubbing reagent, especially if the chemical reagent becomes moisture saturated. Contamination of the instrument

from the scrubber system is hard to predict and time consuming to remedy once it happens. While the molecular sieve effectively absorbs residual $NH_3$ in the combined trap, it does not prevent contamination from compounds larger than 4 Å that could result from the moisture-saturated chemical reagent. A contamination of this type was observed during the 2017 test flight period and is described in Sect. 5. Further, while the cost of molecular sieve is negligible and the material can easily be regenerated, the non-regenerative chemical reagent can significantly accumulate in expense over time depending on

usage rate. In our tests, the chemical scrubbing reagent lasted a few weeks before being compromised. However, the actual lifetime depends on how $NH_3$-rich and humid the operational environment is and how consistently the trap is heated.

Following this assessment, we elected to only evaluate and compare the UZA cylinder and the combined chemical scrubber/molecular sieve during the 2017 test flights. Even though cylinders are rapidly consumed at the high sample flow

rate and thus were replaced prior to each of our flights, we ultimately found this source to be the most convenient and cost-effective method for zeroing the $NH_3$ instrument system during WE-CAN test and research flights. As might be expected, cylinders of dry UHP $N_2$ produced the same zero signal level as a bottled source of UZA, and thus can be utilized as an alternative (and sometimes less costly) $NH_3$-free source for zeroing the instrument. It should also be noted that the calibration signal from standard addition of $NH_3$ on top of a zero background signal generated with UHP $N_2$ can be different

by as much as 10% from a calibration signal measured on top of a background produced with UZA owing to differences in pressure (or collisional) broadening of the $NH_3$ spectral lines in the $\nu_2$ absorption band with different carrier gases (Pearman and Garratt, 1975; Owen et al., 2013).

## 4 Evaluation of the non-passivated instrument on the ground and in flight

### 4.1 Precision, detection limit, and stability

Continuously overflowing the inlet tip with $NH_3$-free air during one of the test flights allowed for characterization of any flight-induced artifacts in the detected absorption signal above a constant, low-level background. An overflow > 500 sccm (e.g., the difference between the flow of zero air being supplied to the inlet and the instrument's sample flow) was maintained to ensure that the sample stream was truly $NH_3$-free during this test. The measured zero signal level on the ground and in flight was the same when tested with a UZA cylinder, a UHP $N_2$ cylinder, or the combined chemical reagent/molecular sieve scrubber. Ambient signal levels were checked periodically for a few minutes at a time throughout the flight to confirm that measured ambient levels were greater than or equal to the measured zero signal level. Measured zero signal levels were consistently within a factor of three of the in-flight instrument precision with changes in altitude up to the 5 km AGL (or ~6 km AMSL, which was the upper range of the C-130 with the WE-CAN payload). Ambient measurements were consistently greater than zero and frequently above the 200 pptv detection limit, even in the free troposphere.

Figure 2 depicts two time segments of data collected at 10 Hz while measuring $NH_3$-free air. One is a 10-minute segment collected in flight in the boundary layer near 1.4 km AGL (e.g., Fig. 2a); the other is a 2-hour segment collected in the laboratory (e.g., Fig. 2b). The instrument was operated without passivant in both cases. Instrument precision, calculated as the Allan deviation or the square root of the Allan variance (Werle et al., 1993), is 430 pptv at 10 Hz and 60 pptv at 1 Hz in flight and 130 pptv at 10 Hz and 40 pptv at 1 Hz on the ground. The Allan variance from the 10-Hz data collected in flight reflects the effects of a frequency-swept vibration applied to the laser objective to reduce motion sensitivity due to optical feedback from the objective. The frequency and amplitude of the vibration were specifically tuned to minimize motion sensitivity at 1 Hz for this flight application. If we define the detection limit as three times the instrument precision, then the estimated 1-Hz detection limits are 180 pptv in flight and 120 pptv on the ground. Therefore, vibrations associated with the C-130 in flight lead to larger detection limit by a factor of 1.5. Since the instrument was not operated without the vibration isolation mounting plate or high frequency vibration of the laser objective during the test flight period, we have no direct comparison for how these added features impacted the measurements. The same precision and detection limit are determined from an Allan variance analysis of $NH_3$ data at a constant mixing ratio of 50 ppbv, collected when calibration gas was added to the sample stream. With averaging, the Allan variance plots suggest that the in-flight detection limit can be reduced to 75 pptv over and averaging period of 5 seconds and 60 pptv over 10 seconds.

Figure 3 shows that zero and calibration signal levels were largely insensitive to in-flight fluctuations in cabin pressure and temperature and changes in altitude within ±200 pptv. For these experiments, the instrument inlet was continuously overflowed with $NH_3$-free air for the duration of a 3-hour pre-flight exercise prior to take off. Overflowing the inlet was purposefully done to keep the instrument system free of contaminants (e.g., exhaust from other aircraft and ground-based support equipment) prior to sampling in flight. A slight positive trend was observed in the zero signal level with increasing altitude during an ascent profile between the ground in Broomfield, Colorado and 6.5 km AMSL; however, there is little variability in the mean zero signal measured during constant altitude legs and the $3\sigma$ standard deviation of each mean is within ±200 pptv of zero. Constant altitude legs were performed roughly every 1 km for a period of 5-10 minutes during the ascent profile. The more apparent increasing trend between $NH_3$ zero signal level and increasing altitude between these straight and level flight legs likely reflects a change in the measured zero signal level due to motion sensitivity of the instrument during accelerations associated with the aircraft's ascent, as described in detail in the next section. Overall, changes in the zero signal level with altitude are largely within ±200 pptv over the entire altitude range tested. We note that the true detection limit of the instrument in flight may be better represented by the full range of variability about the mean zero signal level from the observations in Fig. 3 (e.g., an instrument detection limit of 400 pptv).

Instrument stability was evaluated over the 2-month duration of the WE-CAN intensive and test flight period. Zero signal levels drifted <10% from average, and calibration signal levels drifted <2% from average over the entire period. Variability in the calibration correction factor applied to the measured data to generate final $NH_3$ mixing ratios is <1%. We estimate that

the total uncertainty associated with the reported 1-Hz measurement is ±12% of the measured $NH_3$ mixing ratio plus the 200 pptv detection limit, where the uncertainty of the measured $NH_3$ mixing ratio is calculated by quadrature addition of the associated individual uncertainties. The individual uncertainties include: the permeation rate of the calibration source as measured by the NOAA UV calibration system (±7%), the stability of calibration (±2%) and zero (±10%) measurements over the deployment period, changes in $NH_3$ mixing ratio with changes in flow rates (e.g., ±1% each for dilution of calibration gas into the sample flow, calibration gas addition to the sample flow, and passivant addition to the sample flow as measured with the DryCal flow calibrator), and changes in $NH_3$ calibration signal with changes up to 10 Torr in QC-TILDAS optical cell pressure (±1%). Changes in measured $NH_3$ signal with deliberate changes in optical cell temperature over a 5°C range were <0.1%, and thus are considered negligible in the uncertainty calculations.

## 4.2 Motion sensitivity

Figure 4 shows the sensitivity of the QC-TILDAS to in-flight accelerations measured during the test flight period. Maneuvers performed at 2.6 km AGL while measuring $NH_3$-free air at 10-Hz show excursions in the $NH_3$ signal with respect to acceleration (denoted here as $\Delta NH_3/g$). The $\Delta NH_3/g$ observed for each direction of motion is determined from the slope of a scatter plot of $NH_3$ (in ppb) versus acceleration (in g). Slopes are 3 ppb/g, 1 ppb/g and 5 ppb/g for the side-side, up-down, and fore-aft motions, respectively. These excursions are not due to real changes in $NH_3$ mixing ratios in the ambient air sample; instead they reflect an artifact in the measured zero signal associated with physical movement of the absorption cell optics with accelerations in the side-side, up-down, and fore-aft motions of the aircraft during flight. In particular, excursions in the absorption measurements are likely affected by micro displacements in the distance between the laser and laser objective. Owing to the orientation of the laser objective with respect to the direction of flight in the spectrometer box, micro displacements in the fore-aft direction are expected to have the largest effect. For atmospheric research objectives, accelerations during ascent/descent, turbulence and turns are of particular importance since most flight plans require sampling in the mixed boundary layer, transecting emissions plumes, and performing spirals and/or sawtooth-shaped vertical profiles. Figure 4 shows that the largest accelerations of aircraft motion, specifically during turbulence and turns, are experienced in the vertical plane. Scaling accelerations observed during turbulence in the mixed boundary layer at 0.3 km AGL and turns performed near 2.6 km AGL to the $\Delta NH_3/g$ observed in all three dimensions during maneuvers indicates that changes in $NH_3$ signal during turbulence and turns are < 50 pptv, a factor of four less than the detection limit. It should also be noted that large accelerations in the up-down and fore-aft dimensions are also significant at the onset of vertical ascent. Accelerations measured in the up-down and fore-aft motions at the onset of a 1000 ft/min vertical ascent were measured to be 0.4 g and 0.08 g, respectively. Given the slopes above, these accelerations correspond to a maximum change in $NH_3$ zero signal level of 400 pptv during ascent, which is consistent with the variability in zero signal level observed in Fig. 3 when ascending between constant altitude legs.

## 4.3 Time response of the non-passivated instrument

The time response of the $NH_3$ instrument can be determined from a step change response in $NH_3$ concentration, as described in detail by Zahniser et al. (1995), Ellis et al. (2010) and Roscioli et al. (2016). Briefly, for these experiments, a step change in $NH_3$ concentration is generated by switching off calibration gas while overblowing $NH_3$-free air at the inlet tip. Ellis et al. (2010) and Roscioli et al. (2016), showed that the decrease in $NH_3$ mixing ratio was well represented by a bi-exponential decay of the functional form in Eq. (1):

$$y = y_0 + A_1 exp\left(\frac{-(t-t_0)}{\tau_1}\right) + A_2 exp\left(\frac{-(t-t_0)}{\tau_2}\right) \quad (1)$$

where $y_0$ represents the mixing ratio reached at the end of the decay, $A_1$ and $A_2$ are constants that sum to the stable mixing ratio of $NH_3$ prior to the calibration being switched off, and $\tau_1$ and $\tau_2$ are the decay time constants. $\tau_1$ is typically fast and has been referred to in the literature to correspond largely to the gas exchange time of the flow path and optical cell (~ 0.4 s); $\tau_2$ can be significantly slower and is commonly associated in the literature with the interaction of $NH_3$ molecules with sampling surfaces (Zahniser et al., 1995; Ellis et al., 2010; Miller et al., 2014; Roscioli et al., 2015). The instrument time response can

then be quantified as the time ($t$) that it takes for the $NH_3$ calibration signal to return to some percent of the final zero signal level after the calibration gas was switched off ($t_0$). Given the double exponential nature of the decay, the ratio of $A_2$ to ($A_1 + A_2$), defined as parameter $D$ and reported as percent in previous works (Ellis et al., 2010), can also be a useful tool for describing the fraction of $NH_3$ slowed in reaching the QC-TILDAS due to interactions with the sampling surfaces.

Instrument time response is commonly reported as the $1/e$, 75%, and 90% signal recovery times (with the latter denoted here as $t_{90}$). With an overall instrument uncertainty of ±12%, this instrument's time response is adequately characterized using $t_{90}$. Further, owing to this particular instrument's low detection limit and robust stability over time, we also report for reference $t_{99}$, the response time associated with a 99% signal recovery.

All step change profiles were measured with the instrument configured for use aboard the aircraft, as shown in Fig. 1, and were collected during pre-flight operations on the ground prior to the test flights or in the laboratory between test flight periods. Table 1 summarizes the coefficients and corresponding $1\sigma$ standard deviations from a bi-exponential fit of each time profile shown in Fig. 5. Also included in Table 1 are the resultant values for *%D*, $t_{90}$ and $t_{99}$ extrapolated from the fit coefficients, with uncertainties for these values reflecting propagation of the $1\sigma$ standard deviations of the fit coefficients and an uncertainty of ±0.1 s for $t_0$. From Fig. 5, it appears that a bi-exponential fit does not always do a good job of approximating the observations. Indeed, reduced chi-square values from bi-exponential fit of the decay profiles ranged from 0.4 to 1.3. A triple exponential decay with the functional form shown in Eq. (2):

$$y = y_0 + A_1 exp\left(\frac{-(t-t_0)}{\tau_1}\right) + A_2 exp\left(\frac{-(t-t_0)}{\tau_2}\right) + A_3 exp\left(\frac{-(t-t_0)}{\tau_3}\right) \tag{2}$$

produces better fits to the time profiles shown in Fig. 5. Albeit, the coefficient associated with the third time constant ($A_3$) is small (e.g., $A_3$ is <5% on average of the sum of the coefficients (e.g., [$A_3/(A_1 + A_2 + A_3)$]) and <23% on average of the sum of the coefficients associated with the latter two time constants (e.g., [$A_3/(A_2 + A_3)$]). While the physical basis for using a triple exponential fit is not forthright, it is possible that there is more than one time constant associated with the gas exchange rate through the sample flow pathway, the interaction of $NH_3$ molecules with the sampling surfaces, or a combination of these effects. In the case of multiple time constants associated with the gas exchange rate, it is possible that different residence times could arise from the different pressure regimes of the sample flow pathway (e.g., the portion of the sample flow path at ambient pressure upstream of the critical orifice in the inertial inlet versus the portion of the sample flow path downstream of the critical orifice at pressures between 40 and 100 Torr). In the case of $NH_3$ molecules interacting with the sampling surfaces, additional time constants could be related to variability in the level of cleanliness along the sample flow path. For example, inlet tubing and components were cleaned/replaced following contamination, but the optical cell in the QC-TILDAS was not; thus, more than one time constant might be most plausible, especially for the "typical" and "contaminated" time profiles that were collected following contamination. For consistency with the approaches used in the peer-reviewed literature for characterizing the time response of a QC-TILDAS instrument and for ease of comparison to the values reported by Ellis et al. (2010) and Roscioli et al. (2016), we show the results of the bi-exponential fits in Table 1. However, the possibility remains that the time profiles collected here are not perfectly represented by the bi-exponential air-surface exchange model described by Eq. 1. Therefore, we also utilize the observations in Fig. 5 to directly derive the 90% and 99% signal recovery times (denoted as $t_{90, obs}$ and $t_{99, obs}$). In this case, uncertainties reflect the $\Delta t$ spread in the observations associated with the 90±1% and 99±1% signal recovery levels, where ±1% on the signal recovery level corresponds to ±0.5 ppbv for a 50 ppbv step change, which is well within the limit of detection.

As indicated in Fig. 5 and Table 1, the instrument time response has a clear dependence with the cleanliness of the instrument sampling surfaces. Specifically, an instrument with "clean" sampling surfaces has a much faster time response ($t_{90, obs}$ < 1 s) compared to an instrument with "dirty, or "contaminated", sampling surfaces ($t_{90, obs}$ = 143 s). This effect is apparent regardless of how the $t_{90}$ is determined.

## 4.4 In-flight measurements with the non-passivated instrument

$NH_3$ was measured over a range of altitudes, including several kilometers in the free troposphere (e.g., Fig. 6a). Measured $NH_3$ mixing ratios were as much as 80 ppbv during missed approaches at Greeley-Weld County Airport and in the boundary layer (< 1.5 km AGL) over animal husbandry and agricultural operations in northeastern Colorado during the test flights. In contrast, $NH_3$ mixing ratios in the boundary layer near Akron, Colorado were around 1 ppbv following a few days of rain during the 2017 test flights. We were also fortunate to have the opportunity to sample clear air in the free troposphere for a 10-20 minute period during each of the 2017 and 2018 test flights. A histogram of the $NH_3$ measured in the free troposphere over the northeastern Colorado Front Range during the test flights (e.g., Fig. 6b) indicates that free tropospheric $NH_3$ mixing ratios were frequently greater than 0.4 ppbv during the September 2017 test flight period and frequently greater than 1 ppbv in July 2018 following a period of higher ambient temperatures and less rain. It should be noted that calibrations at very low $NH_3$ mixing ratios (e.g., sub 1 ppbv to 10 ppbv) were not performed during the test flights because the in-flight calibration source was optimized for the $NH_3$ mixing ratios expected in concentrated wildfire smoke during WE-CAN research flights (e.g., >50 ppbv). However, calibrations of the QC-TILDAS performed by the manufacturer and during separate experiments in the laboratory prior to installation on the aircraft show linearity within the instrument uncertainty for $NH_3$ calibration mixing ratios ranging from a few ppbv to hundreds of ppbv. All the same, further measurements are recommended for assessing sampling biases that could arise during field measurements of low mixing ratios of $NH_3$ in clean environments following long periods of exposure to near source level concentrations. The potential for an adsorption-related "memory effect" of $NH_3$ (e.g., Williams et al., 1992) on the sampling surfaces following long-term exposure to high concentrations of $NH_3$ is discussed in following sections.

## 5 The effects of adding passivant

### 5.1 Passivated instrument time response on the ground

### 5.1.1 "Clean" vs. "dirty" instrument conditions

As indicated in Fig. 5 and Table 1, the improvement in instrument time response when passivant is added has a clear dependence with how "clean" or "dirty" the instrument system is. More specifically, there is no difference in time response for the "clean" instrument when operated with and without the C8 passivant, yet the time response increasingly improves with passivant addition to an instrument with increasingly compromised sampling surfaces (e.g., Fig. 5a compared to Fig. 5c). In these tests, the "clean" case refers to a new instrument that had only been operated for a few months after being built and always under relatively pristine conditions (e.g., operated in a laboratory with dry $NH_3$-free air). In this "clean" case, the instrument can recover from a 50 ppbv step change in $NH_3$ in $t_{90} < 1$ s regardless of whether passivant is applied. The step change profiles collected under "typical" and very dirty, or "contaminated", operating conditions demonstrate that adding passivant can greatly improve the overall instrument time response and that the effect of adding passivant can be increasingly beneficial as sampling surfaces are further exposed to "dirty" sampling conditions. In this study, the instrument response is rigorously tested with a single step change of $NH_3$ created by turning off a 50 ppbv calibration gas mixture. We note that such large variations in $NH_3$ mixing ratio may not been full applicable to field measurements in unpolluted regions away from concentrated sources of $NH_3$. As described by Ellis et al. (2010), large gradients in $NH_3$ may be less impacted by surface interactions because "clean" sampling surfaces only have a finite number of adsorption sites that could be quickly filled under high $NH_3$ conditions. At lower $NH_3$ concentrations, a greater fraction of $NH_3$ molecules may interact with the inner surfaces. This could explain why passivation did not help to increase the response time of the instrument.

Before the instrument response could be characterized in flight with passivant under typical ambient operating conditions during the 2017 test flight period, the instrument sampling surfaces experienced a case of extreme contamination. The contamination was likely caused by the chemical scrubbing reagent, which had been used in several prior experiments to evaluate a scrubbed source of $NH_3$-free air, that was compromised by exposure to excessive moisture (e.g., the 2017 test flight period was particularly cold and rainy). Fortuitously, this "contaminated" case, albeit an atypical and non-optimal

operating condition, presented a unique opportunity to test the power of passivation for improving, or in this case recovering, instrument time response. Under "contaminated" conditions, accumulation of $NH_3$ on the sampling surfaces was so severe that $NH_3$ was more prone to sticking on the contaminated inlet surfaces rather than being transmitted to the spectrometer. The time response of the contaminated system was so long (e.g., hours) that it could not be accurately measured (e.g., the instrument would need to run for several hours to achieve 99% signal recovery, which was not possible to accomplish in the time frame of the aircraft operations). Thus, Fig. 5c shows the time response of the "contaminated" instrument after one cleaning, where a 99% signal recovery could be observed within a few hours. The "contaminated" surfaces degraded the instrument time response from $t_{90} < 1$ s to 180 s and $t_{99} = 37$ s to 1200 s. As shown in Fig. 5c, adding a continuous flow of the C8 passivant to the contaminated system brought $t_{90}$ back to 7 s and $t_{99}$ to 180 s. Table 1 also shows a similar value of %D in the passivated case compared to a factor of 4 difference in %D for the non-passivated case when comparing the "contaminated" versus "clean" systems indicating that the time response of the contaminated system is dominated by interaction of $NH_3$ with the instrument sampling surfaces ($\tau_2$). While the proportion of the time response governed by the slow, "adsorptive", term was typically quite low ($D < 10\%$), the magnitude of the step change concentration utilized here is large (e.g., 50 ppb), so caution should be taken when extrapolating these results to ambient observations away from concentrated source regions.

Even though adding passivant to an already "contaminated" instrument cannot instantaneously reset the sampling surfaces to near pristine "clean" condition, the results in Table 1 show that passivant addition has a greater factor of improvement for increasingly "dirty" instruments (e.g., a factor of 2 improvement for "typical" conditions, and a factor of 25 improvement for "contaminated" conditions). Although $NH_3$ accumulation should be avoidable during normal operation even in polluted environments with frequent checks of the step-change time response and cleanings, we conclude that the option of adding passivant can be a useful tool for recovering instrument time response when fast measurements are required and routine maintenance/cleaning is not possible (e.g., when contamination occurs before or during a research flight or at a remote field site).

It should also be noted that the instrument's time response following the contamination event could only be fully recovered by replacing contaminated tubing and performing multiple cleanings of any non-replaceable components between the inlet tip and the QC-TILDAS optical cell. Cleaning consisted of vigorously rinsing components with deionized water several times, followed by a few rinses with 200-proof ethanol, and finally blowing out each component for several minutes with compressed UZA or UHP $N_2$ until all traces of solvent were gone.

### 5.1.2 The effects of increasing passivant concentration

Previous experiments performed in the laboratory showed improvements in instrument time response with increasing addition of C8 passivant (Roscioli et al., 2016). In these experiments, Roscioli et al. (2016) observed nearly a factor of 6 improvement in $t_{90}$ when passivant addition was increased from 1 ppm to 40 ppm. Our tests, reported in Fig. 7 with the instrument operated under near "typical" conditions, confirm that the instrument time response can be increasingly improved by increasing the amount of passivant added at the inlet. While the improvement increases with increasing addition of passivant chemical, the improvements observed for this particular instrument system appear to be exponential with limited improvement when passivant concentrations exceed 60 ppmv.

### 5.1.3 The effects of humid vs. dry sampling conditions

In all cases presented in Table 1, except for the "contaminated" case, $D$ is $< 10\%$ indicating that the time response of the instrument as configured for these experiments is not overly dominated by surface interactions, even when operated under typical in-field ambient measurement operating conditions and without passivant. This is likely because the instrument sampling surfaces are heated to prevent adsorption of water and basic species, as described in Ellis et al. (2010). All the same, it should be noted that the time response tests reported here were performed on top of a sample stream of $NH_3$-free air supplied from a synthetic, dry bottled source or generated using a hygroscopic scrubbing media. Sample humidity is alleged to increase the relative importance of $NH_3$ surface interactions (Ellis et al., 2010), and thus differences in the instrument time

response determined in a laboratory using dry air compared with that determined from data collected in a moist field environment could be significant (Nowak et al., 2007). However, previous observations with respect to sample humidity are inconsistent (Pogány et al., 2016). One study showed that humidity addition increased surface interactions (e.g., more $NH_3$ adsorption on sampling surfaces, (Ellis et al., 2010)), while another study showed $NH_3$ adsorption on sampling surfaces to

decrease with increasing water content (Vaittinen et al., 2014). Although the differences may be attributed to whether the sampling surfaces were sufficiently heated to prevent water adsorption on the sampling surfaces, we were compelled to perform a few basic tests of the effects of humidity on this instrument's time response with and without passivant addition. We only measured two extreme relative humidity conditions for these tests, even though the relationship of surface interactions may be non-linear and vary greatly depending on the fraction of water vapor added as suggested by Pogány et al.

(2016) and Vaittinen et al. (2018). This was done by overflowing a sample of dry versus 80% humidified air at the inlet tip. Dry air was sourced directly from a UZA cylinder; humidified air was generated by passing 9.2 SLPM of UZA through a bubbler filled with dionized water and allowing it to re-mix with 2.2 SLPM of dry UZA before overflowing the humidified air mixture at the inlet tip. Humidifying the sample stream to 80% instead of 100% was intentional to avoid condensation of saturated water vapor onto the sampling surfaces while still providing a rigorous test of the instrument time response in a

humid environment (e.g., the average annual relative humidity in morning in the continental U.S. is ~80%).

Figure 7 shows a slight difference in instrument time response between the humidified versus the dry air sample when the instrument is operated with heated inlet surfaces and without passivant, although the difference is close to being within one standard deviation of the measurement uncertainties. The observations in Fig. 7 are consistent with that observed by Ellis et

al. (2010), which were performed using similar instrumentation and similar magnitude step changes in $NH_3$ mixing ratios. The time responses for both the dry and humidified air samples are increasingly improved when an increasing amount of the C8 passivant is added to the sample stream. This trend is similar to that observed by Roscioli et al. (2016) in dry air samples with the exception of a plateau in the time response improvements at high concentrations of passivant addition. Improvements in time response with passivant addition for the humidified air sample seem to be limited to the same ~60

ppmv threshold as the dry air sample. Overall, the differences in time response between the humidified and dry air samples appear to be small when using a heated inlet system regardless of whether passivant has been added. All the same, we reiterate that a caveat of these tests is that the humidity levels tested here may not provide enough information to fully characterize the effects of passivant addition over the full range of dry to humid sampling conditions. Further characterization of the humidity dependence with and without passivant addition is recommended prior to future

deployments of this instrument system (or similar QC-TILDAS instruments) in humid field environments.

### 5.1.4 Other possible passivant chemicals

Table 1 demonstrates a similar improvement in instrument time response when a C7 passivant, 1H,1H-perfluorheptylamine, is implemented instead of the C8 passivant. Of significant difference is the usage rate of the C7 passivant, which is ~2 hours per gram compared to ~4 hours per gram for the C8 compound at a typical passivant addition flow rate of 200 sccm and

room temperature (~22°C). Longer carbon chain versions of the passivant chemical are expected to produce similar improvements in time response; however, longer chain species are likely to be more difficult to introduce into the sample stream due to reduced volatility.

Prior studies have shown inlet coatings such as a halocarbon wax (Yokelson et al., 2003) and SilcoNert 2000 (Pogány et al.,

2016) can prevent the adsorption on $NH_3$ and water vapor on instrument sampling surfaces. While current coating technology can provide relatively non-sticky surfaces, we note that in field environments, these surface treatments can quickly become overcoated with dust, salt, and other condensables, that ultimately compromise their non-stick properties. Continual re-application of a non-stick coating via the active continuous passivation method described here mitigates this issue.

## 5.2 The effects of adding passivant in flight

### 5.2.1 Test flights in the Colorado Front Range

Missed approaches at the Greeley-Weld County Airport in Colorado (40.4375˚ N, 104.633056˚ W, and 1,432 m AMSL) were performed on multiple occasions during 2017 and 2018 and provided opportunities to sample large, rapid gradients in
gas-phase $NH_3$ while operating the instrument with and without passivant addition. The aircraft often sampled emissions from nearby concentrated animal operations located south and east of the Greeley-Weld County Airport during the 29 September 2017 (e.g., flight tracks in Fig. 8) and 13 July 2018 flights. The aircraft reached a minimum altitude of 50 m AGL during the missed approaches; the maximum $NH_3$ mixing ratios were intercepted between 300 and 400 m AGL, and wind speeds were consistently between 4 and 5 m s$^{-1}$.

$NH_3$ enhancements observed near 13:30 Mountain Daylight Time (MDT) and 14:00 MDT during the 29 Sept 2017 flight represent intersects of the same $NH_3$ plume, yet differ by whether the "contaminated" instrument was operated without (e.g., cyan shaded areas near 13:30 MDT in Fig. 8) or with (e.g., orange shaded areas near 14:00 MDT in Fig. 8) the C8 passivant. Passivant addition is turned off by bypassing the impinger using the quick-connect fittings as described in Sect. 2. The large
drop in $NH_3$ signal at 13:23 MDT when the passivant impinger is bypassed indicates that the passivant coating is quickly stripped from the sampling surfaces (in about ~60 s) when the coating is not actively and continuously applied. As depicted in the cyan shaded area of the time series in Fig. 8, $NH_3$ mixing ratios measured by the non-passivated, "contaminated" instrument are significantly reduced. Only 5 ppbv was observed out of the ~45 ppbv of $NH_3$ expected during this plume transect suggesting that only ~10% of ambient gas-phase $NH_3$ molecules were actually transmitted to the QC-TILDAS while
~90% were adsorbed to the "contaminated" sampling surfaces. In contrast, a time series of $NH_3$ enhancements sampled during plume transects collected with a "clean" instrument system during the flight on 13 July 2018 (e.g., Fig. 9) shows little difference in the amount of $NH_3$ measured by the spectrometer when the instrument is operated with passivant (e.g., orange shaded area at 13:46 MDT in Fig. 9) and without passivant (e.g., cyan shaded area at 13:55 MDT in Fig. 9). Therefore, we attribute the reduced transmission of $NH_3$ to the QC-TILDAS in the non-passivated and "contaminated" instrument system
to increased retention of $NH_3$ due to adsorption on the walls of the "contaminated" sampling surfaces.

To further test this hypothesis, we quantify the relative amount of $NH_3$ ejected from the sampling surfaces when passivant is re-introduced to the sample stream (e.g., the red shaded areas in Fig. 8) following a non-passivated plume intersect (e.g., the cyan shaded area in Fig. 8 that represents a period of sampling large $NH_3$ mixing ratios without passivant). Ejection occurs
when passivant is re-added to the system and any $NH_3$ molecules that have adhered to the sampling surfaces during the non-passivated sampling period are released from the surfaces and replaced with the passivant coating. The ejection signal is a measure of the $NH_3$ molecules that are kicked off of the sampling surfaces and transmitted to the QC-TILDAS, and manifests in the time series as an intense increase followed by a slightly less intense decrease in measured $NH_3$ mixing ratio and an exponential decay (as depicted in Fig. 8). By adding up the amount of $NH_3$ ejected from the sampling surfaces with
the amount of $NH_3$ measured by the non-passivated instrument, we can quantify the total amount of $NH_3$ that should have been transmitted to the spectrometer for a specific sampling event (aka. $NH_3$ expected) compared to $NH_3$ actually detected ($NH_3$ measured) to quantify how much $NH_3$ was absorbed to the sampling surfaces when the instrument was operated without passivant. To do this, we integrate the highlighted segments of Fig. 8, which represent plume transects sampled with and without passivant and ejection; results are reported in Table 2. From these results, we can make the following
conclusions. First, the $NH_3$ ejected from the sampling surfaces when passivant is re-added accounts for nearly all of the missing $NH_3$ detected by the QC-TILDAS when the instrument system is operated without passivant. Second, passivant addition is capable of recovering a near-optimal instrument sampling capability even when the instrument is operated under non-optimal cleanliness conditions. These results are evident from the small percent difference between expected versus measured $NH_3$ during the feedlot plume intersects. Third, the likelihood of $NH_3$ adsorption to sampling surfaces increases
with increasing build-up of water and basic species on the sampling surfaces, thereby decreasing the amount of $NH_3$ transmitted to the QC-TILDAS when the instrument is operated without passivant. This is demonstrated by a factor of six greater fraction of $NH_3$ ejected from the sampling surfaces during the 2017 "contaminated" case compared to the 2018 "clean" case. In general, for both cases, the small percent difference in measured and expected $NH_3$ additionally

demonstrates that the NH$_3$ plumes used for each case analysis likely originated from the same feedlot source and was intersected by the aircraft under similar meteorological (e.g., wind) conditions.

Additional qualitative information was unintentionally gained from a Proton Transfer Reaction Time of Flight Mass Spectrometer (PTR-ToF-MS) that was simultaneously deployed aboard the C-130 aircraft. The PTR-ToF-MS was operated with H$_3$O$^+$ reagent ion and positioned just aft of the QC-TILDAS-based instrument in the aircraft cabin. While previous studies have shown a PTR-ToF-MS to be successful in quantifying *in-situ* NH$_3$ mixing ratios from airborne measurement platforms (Müller et al., 2014; Norman et al., 2007), the PTR-ToF-MS deployed during WE-CAN was not optimized for measuring NH$_3$ and seemed to suffer from a previously-identified issue (e.g., high background signal correlated with the instrument's ion source (Müller et al., 2016)). Since the primary objective of the PTR-ToF-MS during WE-CAN was to measure non-methane volatile organic compounds in smoke, the PTR-ToF-MS instrument inlet is non-specific and not calibrated for NH$_3$. However, we were able to kinetically calculate an NH$_3$ mixing ratio from the raw PTR-ToF-MS instrument signal collected during the flight on 29 Sept 2017, and thus the PTR-NH$_3$ measurement (e.g., blue line in Fig. 8) can be used to qualitatively confirm the QC-TILDAS-based instrument observations in this study. Passivant was not added to the PTR-ToF-MS; active continuous passivation was only applied the QC-TILDAS-based instrument during the selected times described above. It is clear by visual comparison to the PTR-ToF-MS that the non-passivated, "contaminated" QC-TILDAS instrument did not capture all of the expected ambient NH$_3$. This is evident from the differences in measured NH$_3$ mixing ratios reported in Fig. 8 during the time period between 13:20 and 13:23 when the QC-TILDAS was operated without passivant. During this time period the PTR-ToF-MS consistently measured more NH$_3$ than the QC-TILDAS, with the enhancement measured by the PTR during the plume intersect at 13:30 MDT showing an expected mixing ratio of ~45 ppbv. According to PTR-NH$_3$, the integrated NH$_3$ signal during the plume intersect at 13:30 MDT was only 14% less than the integrated NH$_3$ signal measured during the plume intersect at 14:00 MDT, and thus a significant enhancement in NH$_3$ should have been observed by the QC-TILDAS-based instrument. However, the non-passivated, "contaminated" QC-TILDAS-based instrument measured only a fraction of the NH$_3$ expected during the plume transect at 13:30 MDT, with the only attributable difference being NH$_3$ molecules adsorbing to the sampling surfaces. It should be noted that a large and variable background, which appeared to be inversely correlated with the H$_3$O$^+$ ion signal, prevented PTR-NH$_3$ from being determined for any of the 2018 WE-CAN test and research flights. NH$_3$ mixing ratios for the test flight on 29 Sept 2017 could only be calculated from the PTR-ToF-MS raw instrument signal because the H$_3$O$^+$ ion signal was ~10% lower in magnitude and significantly more stable than ion signal during the 2018 flights.

For further perspective, we would like to highlight that NH$_3$ measurements could only be collected with the "contaminated" instrument during the test flight on 29 September 2017 because of the option to add passivant to the sample stream. The contamination occurred only hours before take-off leaving too little time to disassemble and clean the instrument sampling surfaces prior to flight. In this case and without the option for passivant addition, the alternatives would have been to either cancel the flight and potentially miss an ideal sampling opportunity for addressing the project's scientific goals or conduct the research flight without NH$_3$ measurements. The latter would have been a significant loss if the NH$_3$ measurements were central to answering the project's scientific questions, as was the case for the 2018 WE-CAN field campaign.

### 5.2.2 Research flights in concentrated wildfire smoke during WE-CAN

Concentrated wildfire smoke plumes from the South Sugarloaf Fire in Nevada (41.812° N, 116.324° W) were sampled during a WE-CAN research flight on 26 August 2018 (RF15). The fire originated on 17 August 2018 and was caused by lightning. Winds were 12 to 16 m s$^{-1}$ and consistently blowing from the southwest to the northeast. Figure 10 shows NH$_3$ and carbon monoxide (CO) measurements from RF15 that were collected during two crosswind intercepts of the South Sugarloaf Fire smoke plume. Intercepts of the smoke plume were performed at 4400 m AMSL at roughly 75 and 200 km downwind of the South Sugarloaf Fire, and correspond to roughly 1 to 4 hours of aging since emission. The first smoke plume intercept at 20:08 UTC was conducted with passivant addition to the NH$_3$ instrument; the second pass at 20:21 UTC was non-passivated. Linear regression analysis confirms that NH$_3$ and CO measurements are strongly correlated ($R^2 > 0.9$) during both plume intersects. CO measurements were simultaneously collected aboard the C-130 during WE-CAN using a similar compact model, QC-TILDAS instrument (Aerodyne). In contrast to the NH$_3$ instrument, the CO QC-TILDAS was operated at an

absorption wavelength of 2200 cm$^{-1}$, did not require a heated inertial inlet, was operated with a much lower sample flow rate (0.5 SLPM), and was recorded on a 1-Hz timescale. Additionally, ambient air was introduced to the CO instrument via a pressure-controlled, pumped bypass inlet system maintained at 1 SLPM and 265 Torr and constructed of 15 feet of ¼" o.d. stainless steel tubing. We expect a slightly longer residence time (e.g., 1-2 s) for CO compared to NH$_3$ given the significantly
lower sample flow rates and longer segments of tubing used with the CO instrument. During WE-CAN, the NH$_3$ instrument was typically zeroed between crosswind transects of a wildfire smoke plume when in background air and either just prior to or during turns. The instrument was zeroed every 10-20 mins during transits from Boise to the wildfires sampled with the frequency of zeros depending on the transit time. Zeros measured during WE-CAN research flights were typically collected for a period of 1 to 2 minutes, a duration much greater than the instrument response time, to ensure that zeros were measured
well within 90% of the final zero signal level. Prior to each research flight, the NH$_3$ instrument was overflowed with NH$_3$-free air for the duration of a 2-hour pre-flight exercise.

Differences in background mixing ratios of NH$_3$ and CO measured before and after the first transect of the smoke plume from the S. Sugarloaf fire are apparent in the magnified timeseries for each in Fig. 10. The differences in NH$_3$/CO ratio
observed at 20:14 UTC and 20:25 UTC following in-smoke measurements of NH$_3$ that exceeded 400 ppbv could have resulted from physical differences in plume chemistry, mixing or background composition on either side of the plume, an adsorption-related memory effect in the sample plumbing due to retention of NH$_3$ molecules adsorbed to the sampling surfaces (Williams et al., 1992), or a combination of both. Since the root of the differences are difficult to distinguish and may vary among the WE-CAN research flights, we utilized these differences to characterize the instrument time response
given the worst-case scenario that the differences in background observed in Fig. 10 are solely attributed to memory effects on the sampling surfaces. In this worst case, the response time for the NH$_3$ measurement following the plume transect to recover to near background mixing ratio levels observed prior to the plume transect (e.g., 1 ppbv) is roughly 250 s. The time frame most closely resembles $t_{99,obs}$ for the "typical" condition when the instrument is operated with or without passivant. This recovery time and "typical" cleanliness condition are within our expectations for the instrument during this research
flight (RF15) since the instrument had routinely been used to sample near source concentrations of NH$_3$ in smoke during several prior consecutive research flights without refreshing the sampling surfaces between flights.

Fine structure features in the time series (e.g., Fig. 10a and b) highlights a slightly faster time resolution for NH$_3$ compared to CO for the instruments as configured here. To quantify differences in the time resolution of the fine structure features
observed, NH$_3$ measurements were incrementally averaged from 1 to 5 seconds until linear regression analysis of scatter plots of CO versus the averaged NH$_3$ data points resulted in a maximum $R^2$ value. Averaging NH$_3$ to 3 seconds resulted in the best fit (e.g., highest $R^2$ value) for both the passivated and non-passivated cases, indicating that NH$_3$ measurements acquired during WE-CAN were equally faster than the CO measurements regardless of whether passivant was added. A similar time resolution observed for the passivated and non-passivated NH$_3$ measurements is consistent with the sampling
surfaces being relatively "clean" or having a "typical" level of cleanliness during this research flight. We also note that only a small fraction of NH$_3$ (<1%) is ejected from the sampling surfaces when passivant was re-added to the NH$_3$ instrument at 20:29 UTC following the second transect, thereby further indicating that only a small amount of NH$_3$ molecules were adsorbed to the  instrument sampling surfaces during this flight. No degradation in time resolution for the non-passivated NH$_3$ instrument during RF15, which was the second-to-last research flight of a 6-week field campaign where intense smoke
plumes with NH$_3$ mixing ratios ranging from 50 to 400 ppbv were routinely sampled for 4-6 hours every 1 to 3 days, further demonstrates that routine passivant addition throughout the field deployment was instrumental in preventing sampling surfaces from getting "dirty". Cleanings were only performed twice throughout the 6-week field campaign (roughly once every two weeks) when a gray-ish "smoky" residue could be visually observed on the inner sleeve of the glass inertial inlet; cleanings followed the procedure described in Sect. 4.

Since the time response of the CO measurement was limited by its sample flow rate and inlet configuration, we also compare NH$_3$ to acetonitrile (CH$_3$CN) measured by the PTR-ToF-MS. CH$_3$CN is well correlated with NH$_3$ in smoke, and may be more representative of a true 1-Hz tracer owing to operation of the instrument inlet at a flow rate of ~15 SLPM. However, there are no measurements from the PTR-ToF-MS during RF15, the research flight during which the NH$_3$ instrument was
systematically tested with and without passivant. Instead, we use measurements of CH$_3$CN from the Bear Trap Fire (RF09)

conducted on 09 August 2018 to perform a similar linear regression analysis of fine structure features of measured $NH_3$ versus $CH_3CN$, with $CH_3CN$ incrementally averaged up to 5 seconds. We find the best fits result from linear regressions of measured $NH_3$ with the 1-Hz reported and 2-second averaged $CH_3CN$ ($R^2$ is > 0.97 and within 0.001 of each other).

## 6 Conclusions

A closed-path QC-TILDAS instrument for measuring $NH_3$ was outfitted with an inertial inlet for filter-less separation of particles, a custom-designed aircraft inlet, a custom-built vibration isolation mounting plate, and the option for actively and continuously adding passivant to the sample stream. This flight-ready $NH_3$ instrument system was then deployed on the NSF/NCAR C-130 aircraft during test and research flights associated with the WE-CAN field campaign. The instrument was configured to measure large, rapid gradients in gas-phase $NH_3$, over a range of altitudes, in smoke (e.g., ash and particles), in the boundary layer (e.g., during turbulence and turns), in clouds, and in a hot aircraft cabin. Important design goals were to minimize motion sensitivity, maintain a reasonable detection limit, and minimize $NH_3$ "stickiness" on sampling surfaces to maintain fast time response in flight. The addition of a high frequency vibration to the laser objective in the QC-TILDAS and mounting the QC-TILDAS on a custom-designed vibration isolation plate were successful for reducing motion sensitivity. Allan variance analyses of 10-Hz data collected near 1.4 km AGL indicate that the in-flight instrument precision for this system as configured for the C-130 aircraft is 60 ppt at 1 Hz, with a corresponding $3\sigma$ detection limit of 180 ppt. Owing to variations observed in flight with respect to changes in cabin pressure and temperature and changes in altitude, the full range of the instrument's detection limit is likely closer to 400 pptv. The detection limit allowed measurement over a range of altitudes, and $NH_3$ mixing ratios in the free troposphere were frequently < 1 ppbv. Variations in $NH_3$ associated with turbulence and turns were also within the instrument's limit of detection.

Characterization of the instrument's time response in flight and on the ground with and without adding passivant showed that adding passivant to a "clean" or "typical" instrument system had little impact on the instrument's time response. This observation is consistent with previous studies using non-passivated instruments (Ellis et al., 2010; von Bobrutzki et al., 2010; Zöll et al., 2016; Whitehead et al., 2008) that clearly state the importance and necessity of careful instrument handling and rigorous experimental procedures for collecting high-quality, rapid-response measurements of $NH_3$. In contrast, and as highlighted in this work, passivant addition can greatly improve the time response of an instrument with "contaminated" sampling surfaces, thereby ensuring that high-quality, rapid-response $NH_3$ measurements can continue to be collected during field intensives where instrument components can be difficult to regularly access and keep clean (e.g., during research flights associated with an airborne field campaign). A comparison of passivated and non-passivated $NH_3$ measurements from a flight conducted near the end of the WE-CAN field campaign further indicates the utility of passivant addition for preventing build-up of water and basic species on instrument sampling surfaces over long periods of time, thereby helping to keep the instrument relatively clean throughout a several-week long field intensive.

$NH_3$ accumulation on sampling surfaces should be avoidable during normal operation of the instrument, even in $NH_3$-rich and humid environments, with frequent checks of the step-change response and routine cleanings whenever a temporal profile from a step change indicates that the instrument response time has degraded. This may be especially true when sampling on the ground since a substantial volume of the sampling surfaces associated with the aircraft inlet can be eliminated. However, in the case where cleaning is not possible (e.g., when contamination occurs in flight or just prior, or when ground-based instrumentation cannot be easily accessed), adding passivant can be a useful tool for recovering time response and acts as an insurance policy for being able to continuously collect high-quality, fast-response measurements from mission critical instruments. The option for passivant addition proved to be especially advantageous during the WE-CAN test flight period since it allowed for continuous collection of high-quality, fast-response $NH_3$ measurements even when the instrument was compromised by a known contamination. Continuous addition of passivant during WE-CAN research flights had the added advantage of maintaining an optimum level of instrument cleanliness even when used to sample concentrated wildfire smoke plumes from 15+ research flights over the 6-week duration of the WE-CAN field intensive.

The observations presented here indicate that the active continuous passivation technique combined with a closed-path, optical-based $NH_3$ instrument can have utility when rapid (1 Hz or greater) collection of $NH_3$ is critical to the project's scientific objectives (e.g., measuring fluxes, sampling from aircraft or another mobile research platform). Passivant addition can help maintain an optimum level of operation and data collection in $NH_3$-rich/humid environments and/or when contamination of sampling surfaces is likely, yet frequent cleaning is not possible. Passivant addition may not be necessary for fast operation, even in polluted environments, if sampling surfaces can be cleaned whenever a step-change response to $NH_3$ shows time response has degraded.

*Data availability.* $NH_3$ measurements from the WE-CAN field campaign utilized in this work can be found at https://doi.org/10.26023/2WAS-8Z23-QD0Z. CO data from WE-CAN can be accessed at https://doi.org/10.26023/Q888-WZRD-B70F. $CH_3CN$ data from WE-CAN can be access at https://doi.org/10.26023/K9F4-2CNH-EQ0W. Navigation, state parameters, and microphysics flight-level data provided by the UCAR/NCAR Earth Observing Laboratory are available from https://doi.org/10.26023/G766-BS71-9V03. Additionally, readers may contact the corresponding authors for data access.

*Author contributions.* IP prepared the manuscript with contributions from all co-authors. IP and JL designed and performed the laboratory and ground-based experiments and operated the $NH_3$ flight instrument during 2018 WE-CAN flights. All authors participated in instrument preparation and installation aboard the aircraft and in-flight data collection during the 2017 test flights.

*Competing interests.* The authors declare that they have no conflict of interest.

*Acknowledgments.* Funding for this work was provided by the National Science Foundation through grant number AGS-1650786. The authors wish to acknowledge Dr. Teresa Campos (NCAR) for providing CO measurements during WE-CAN; the scientists, engineers, and crew members of the WE-CAN Project Team that made these test and research flights possible; and the WE-CAN Data Archive Center at NCAR's Earth Observing Laboratory. We thank the two anonymous reviewers as well as Da Pan, Xuehui Guo, and Mark Zondlo for helpful comments and suggestions that have greatly improved this manuscript.

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

**Table 1. Summary of coefficients from fit to a bi-exponential decay and the resultant values for *%D*, *t₉₀* and *t₉₉* and the time responses derived directly form the observations (*t₉₀,ₒᵦₛ* and *t₉₉,ₒᵦₛ*) for time profiles generated from a step change in NH₃ mixing ratio under different levels of instrument "cleanliness".**

| Level of instrument "cleanliness" | $y_0$ | $A_1$ | $\tau_1$ | $A_2$ | $\tau_2$ | $\chi^2$ | %D | $t_{90}$ (s) | $t_{99}$ (s) | $t_{90,obs}$ (s) | $t_{99,obs}$ (s) |
|---|---|---|---|---|---|---|---|---|---|---|---|
| | | | | *Not passivated* | | | | | | | |
| Clean | 0.27±0.01 | 95.3±0.6 | 0.200±0.002 | 5.1±0.1 | 18.9±0.5 | 0.40 | 5.1±0.1 | 0.6±0.1 | 37±4 | 0.9±0.2 | 60±30 |
| Typical[a] | 0.41±0.01 | 85.6±0.7 | 0.441±0.006 | 7.8±0.1 | 85±1 | 0.83 | 8.4±0.1 | 1.6±0.2 | 220±23 | 5.6±1.1 | 335±40 |
| Contaminated | 0.62±0.02 | 68.3±0.5 | 8.2±0.1 | 17.2±0.1 | 306±3 | 0.65 | 20.1±0.2 | 180±19 | 1200±127 | 143±2 | 1700±100 |
| | | | | *Passivated with C8 compound* | | | | | | | |
| Clean | 0.32±0.01 | 90.8±0.6 | 0.225±0.003 | 7.9±0.1 | 18.5±0.4 | 0.52 | 8.0±0.1 | 0.8±0.1 | 45±5 | 1.7±0.2 | 90±55 |
| Typical[a] | 0.35±0.01 | 97.3±0.9 | 0.304±0.005 | 5.4±0.1 | 63±2 | 1.04 | 5.3±0.1 | 0.9±0.1 | 130±14 | 2.0±0.4 | 198±26 |
| Contaminated | 0.46±0.09 | 95.1±1.1 | 1.73±0.04 | 9.1±0.4 | 63±4 | 1.28 | 8.8±0.4 | 7±2 | 180±41 | 11±2 | 215±26 |
| | | | | *Passivated with C7 compound[b]* | | | | | | | |
| Typical[a] | 0.86±0.01 | 109.0±0.5 | 0.390±0.003 | 5.3±0.1 | 36.7±0.7 | 0.40 | 4.7±0.1 | 1.3±0.1 | 135±14 | 1.8±0.2 | 185±28 |

[a]Representative of the typical operating conditions of the instrument in an ambient field environment.

[b]For experiments performed in the lab after the contamination event and with the C7 compound, the aircraft inlet and inertial inlet were rinsed with solvents and dried with UZA and the sample tubing between the inertial inlet and QC-TILDAS was replaced. Several components along the sample pathway were intentionally cleaned but not replaced with pristine surfaces so that a mid-level of cleanliness could be assess and the benefits of adding C7 versus C8 passivant could be clearly observed. All sampling surfaces up to the optical cell in the QC-TILDAS were thoroughly cleaned following these tests and prior to the 2018 WE-CAN deployment.

**Table 2. The integrated sum of NH₃ detected (in units of ppbv) during intersect of a feedlot plume when the instrument was operated with passivant (orange shaded area in Fig. 8), without passivant (cyan shaded area), and during passivant ejection (red shaded area). Also reported for comparison is the total expected NH₃ signal from the non-passivated instrument plus ejection, the percent difference in the expected NH₃ signal from that measured with the passivated instrument, and the fraction of NH₃ ejected out of the total expected NH₃ signal.**

| Test Case (Date) | $NH_3$ measured without passivant | $NH_3$ ejected | $NH_3$ expected (=$NH_3$ without passivant + $NH_3$ ejected) | $NH_3$ measured with passivant | Percent Difference [100*(meas-exp) / meas] | Fraction of $NH_3$ ejected out of $NH_3$ expected |
|---|---|---|---|---|---|---|
| "Clean" conditions (13 July 2018) | 26,747.6 | 4,503.8 | 31,251.4 | 33,176.1 | 5.8 | 0.14 |
| "Contaminated" conditions (29 Sept 2017) | 2,564.0 | 14,159.5 | 16,723.5 | 24,353.4 | 31.3 | 0.85 |

**Figure Captions**

Figure 1: (a) Schematic of the instrument as configured for flight on the NSF/NCAR C-130 aircraft. The sample flow path (blue arrows) starts at the inlet tip, which is a short piece of 3/8" o.d., ¼" i.d. PFA tubing that protrudes slightly from the face of the aircraft inlet strut. A PFA injection block (1/4" i.d.) housed inside the aircraft inlet strut allows calibration gases and passivant to be added to the sample stream within a few centimetres of the inlet tip. A 71-cm length of 3/8" o.d., ¼" i.d. PFA tubing then directs ambient air into a quartz inertial inlet where particles >300 nm are separated from the sample stream, and a 36-cm length of 3/8" o.d., ¼" i.d. PFA tubing directs the sample flow from the inertial inlet to the QC-TILDAS. The particle-rich stream is pumped away (orange arrows). The sample flow path is heated to 40 ˚C where possible to minimize interactions of $NH_3$ with sampling surfaces. The sample flow path is purged overnight in the reverse flow direction with 40 sccm of $N_2$ injected near the pressure control valve. A 0-1000 Torr range baratron (denoted as $P$) measures pressure just upstream of the critical orifice in the inertial inlet for active continuous determination of the sample flow rate. An auxiliary draw acts to flush the dead volume formed near the base of the conical-shaped critical orifice in the inertial inlet. (b) Solid model of the QC-TILDAS vibration isolation mounting plate. (c) Photograph of the QC-TILDAS mounted to the vibration isolation plate while installed aboard the C-130 aircraft. (d) Photograph of the impinger used for active continuous passivant addition.

Figure 2: 10-Hz measurements collected while overblowing $NH_3$-free air at the inlet tip (a) in flight in the boundary layer near 1.4 km AGL and (b) on the ground in the laboratory. Upper traces represent the raw 10-Hz data collected; lower traces depict the Allan variance of the corresponding data set. An offset of -150 pptv was applied to the noise guidelines (gray and dashed lines) in panel (a) to account for differences induced by the vibration applied to the laser objective to reduce motion sensitivity in flight; the vibration was not applied during laboratory tests depicted in panel (b).

Figure 3: In-flight variations in zero signal level (in units of ppbv of $NH_3$) with respect to changes in (a) altitude, (b) cabin pressure, and (c) cabin temperature. A time series (d) illustrates the effects of motion sensitivity on the zero signal level as the aircraft initiates ascent and then levels off at a constant altitude. Gray symbols and lines represent the 1 s average of all of the 10-Hz data points collected in flight while overblowing the inlet tip with $NH_3$-free air. Colored symbols and error bars represent the average $NH_3$ zero signal and 3σ standard deviation for each altitude step of an ascent profile, 5 Torr increments in cabin pressure, and 2˚C increments in cabin temperature. Variations are largely within ±200 pptv (denoted by the light gray shaded areas). Gaps in the time series represent times when the instrument was performing a calibration or measuring ambient air.

Figure 4: Maneuvers performed at 2.6 km AGL in flight while overblowing the inlet tip with $NH_3$-free air show changes in $NH_3$ zero signal (in units of ppbv) with respect to aircraft accelerations (in units of g) in the (a) side-side, (b) up-down, and (c) fore-aft motions. Changes in $NH_3$ zero signal associated with (d) turbulence in the mixed boundary layer at 0.3 km AGL and (e) during turns at 2.6 km AGL are < 50 ppt when scaled by the $\Delta NH_3/g$ observed in each individual dimension during maneuvers. Side-side and fore-aft accelerations are offset by +0.5 g for display purposes in plots (d) and (e).

Figure 5: Normalized $NH_3$ signal (in %) versus elapsed time (in seconds) following a step change in $NH_3$ mixing ratio generated by switching off calibration gas at $t$=0 s. Temporal profiles and associated bi-exponential fits are shown for the non-passivated and passivated instrument operated under (a) "clean", (b) "typical", and (c) "contaminated" sampling surface conditions. Fits ranged from $t_0$ to 400, 1000, and 3000 s for the "clean", "typical", and "contaminated" cases, respectively, in accord with the elapsed time collected for each time profile.

Figure 6: Vertical profiles of $NH_3$ (in ppbv) and potential temperature (in K) from (a) the first and third test flight in 2017 and (b) the test flights in 2018 when the instrument was operated without passivant. $NH_3$ mixing ratios as high as 80 ppbv were observed in the mixed boundary layer during missed approaches at Greeley-Weld County Airport and over northeastern Colorado compared to average mixing ratios of ~0.8 ppbv near Akron, Colorado following several days of rain. (c) Histograms of the corresponding $NH_3$ measurements collected above 1.5 km AGL (dashed line) show that measurements were frequently larger than 200 ppt, especially measurements that were collected in the free troposphere.

Figure 7: Response times ($t_{90}$ and $t_{99}$) and %$D$ associated with increasing C8 passivant addition for a step change in NH$_3$ mixing ratio generated on top of a dry versus humidified background.

Figure 8: (upper) C-130 flight tracks and measured winds during the 29 Sept 2017 test flight. Colored segments of the flight tracks highlight enhancements in NH$_3$ measured downwind of concentrated animal operations (brown symbols sized by head of cattle) located southeast of Greeley-Weld County Airport. (lower) Time series of 1-Hz NH$_3$ mixing ratios measured using the "contaminated" QC-TILDAS instrument and kinetically calculated from a raw instrument signal obtained simultaneously aboard the C-130 aircraft by a Proton Transfer Reaction Time of Flight Mass Spectrometer (PTR-ToF-MS). Portions of the flight when the "contaminated" QC-TILDAS was non-passivated (at 13:30 MDT) and passivated (at 14:00 MDT) are highlighted by colored shaded areas in the time series. Passivant was disconnected from the QC-TILDAS instrument system between 13:23 and 13:34 MDT (as indicated by the gray shaded area); NH$_3$ ejection is observed when passivant is re-added to the system (red shaded area).

Figure 9: Time series of 1-Hz NH$_3$ mixing ratios measured with a "clean" QC-TILDAS during the test flight on 13 July 2018. Portions of the flight when the QC-TILDAS was passivated (at 13:46 MDT) and non-passivated (at 13:58 MDT) are highlighted by colored shaded areas in the time series. Passivant was disconnected from the instrument system between 13:47 and 13:58 MDT (as indicated by the gray shaded area); NH$_3$ ejection is observed when passivant is re-added to the system (red shaded area).

Figure 10: Time series of 1-Hz NH$_3$ (black lines) and CO (red lines) measured during a crosswind transect of the smoke plume from the South Sugarloaf Fire (RF15) on 26 August 2018. The transects represent nearly identical passes through the smoke plume with the only perturbation of the NH$_3$ instrument being operated (a) with passivant and (b) without passivant. Changes in fine structure features of NH$_3$ have the strongest $R^2$ correlation with CO when the NH$_3$ measurements are averaged to 3 s. A x50 magnified view of 1-Hz NH$_3$ (blue lines) and a x10 magnified view of CO (orange lines) shows differences in background levels of NH$_3$ compared to CO before and after each plume transect.

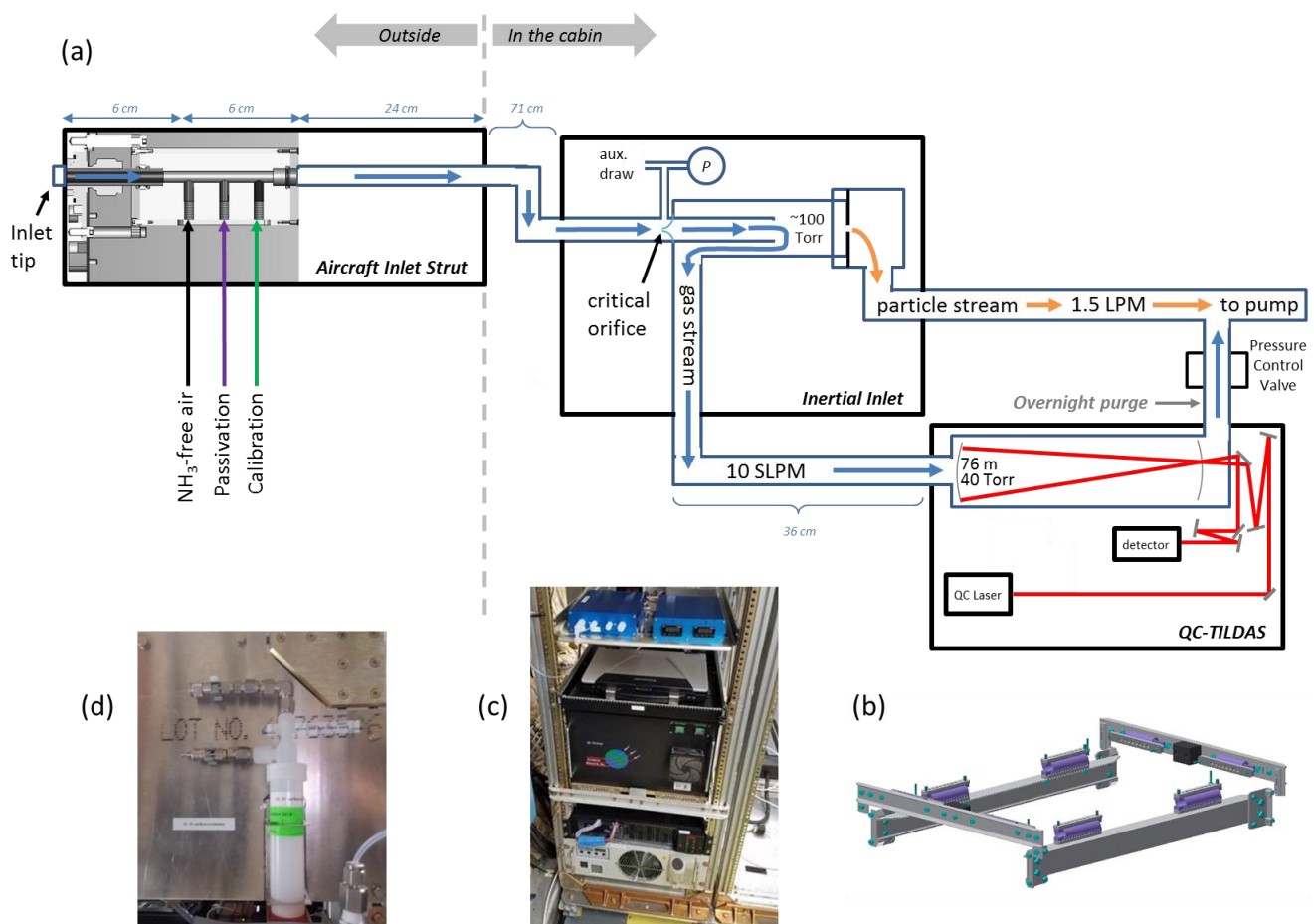

**Figure 1: (a)** Schematic of the instrument as configured for flight on the NSF/NCAR C-130 aircraft. The sample flow path (blue arrows) starts at the inlet tip, which is a short piece of 3/8" o.d., ¼" i.d. PFA tubing that protrudes slightly from the face of the aircraft inlet strut. A PFA injection block (1/4" i.d.) housed inside the aircraft inlet strut allows calibration gases and passivant to be added to the sample stream within a few centimetres of the inlet tip. A 71-cm length of 3/8" o.d., ¼" i.d. PFA tubing then directs ambient air into a quartz inertial inlet where particles >300 nm are separated from the sample stream, and a 36-cm length of 3/8" o.d., ¼" i.d. PFA tubing directs the sample flow from the inertial inlet to the QC-TILDAS. The particle-rich stream is pumped away (orange arrows). The sample flow path is heated to 40 ˚C where possible to minimize interactions of NH$_3$ with sampling surfaces. The sample flow path is purged overnight in the reverse flow direction with 40 sccm of N$_2$ injected near the pressure control valve. A 0-1000 Torr range baratron (denoted as *P*) measures pressure just upstream of the critical orifice in the inertial inlet for active continuous determination of the sample flow rate. An auxiliary draw acts to flush the dead volume formed near the base of the conical-shaped critical orifice in the inertial inlet. **(b)** Solid model of the QC-TILDAS vibration isolation mounting plate. **(c)** Photograph of the QC-TILDAS mounted to the vibration isolation plate while installed aboard the C-130 aircraft. **(d)** Photograph of the impinger used for active continuous passivant addition.

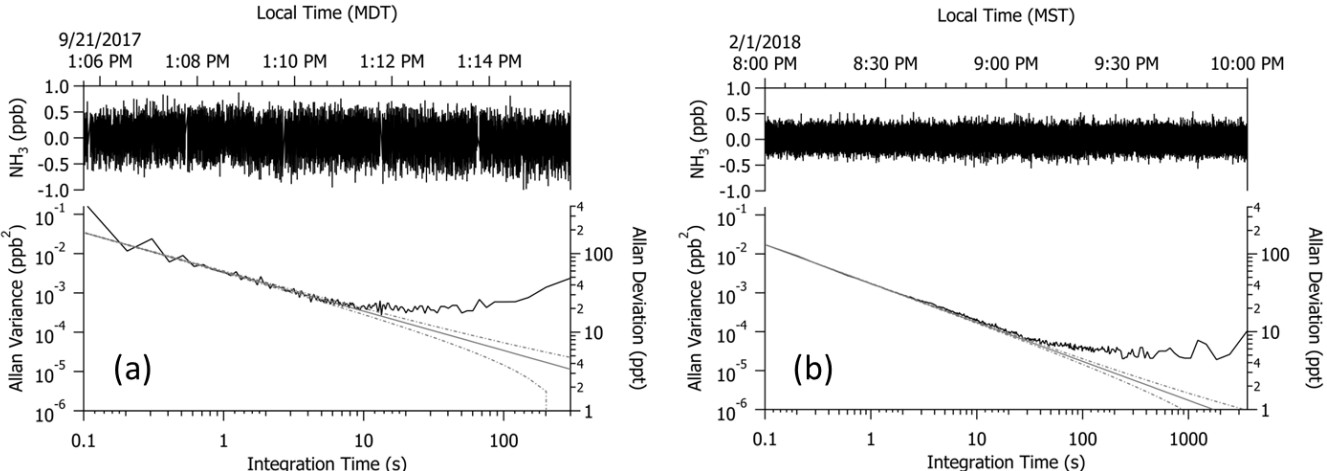

**Figure 2: 10-Hz measurements collected while overblowing NH₃-free air at the inlet tip (a) in flight in the boundary layer near 1.4 km AGL and (b) on the ground in the laboratory. Upper traces represent the raw 10-Hz data collected; lower traces depict the Allan variance in ppb² (left axis) and Allan deviation in ppt (right axis) of the corresponding data set. An offset of -150 pptv was applied to the noise guidelines (gray and dashed lines) in panel (a) to account for differences induced by the vibration applied to the laser objective to reduce motion sensitivity in flight; the vibration was not applied during laboratory tests depicted in panel (b).**

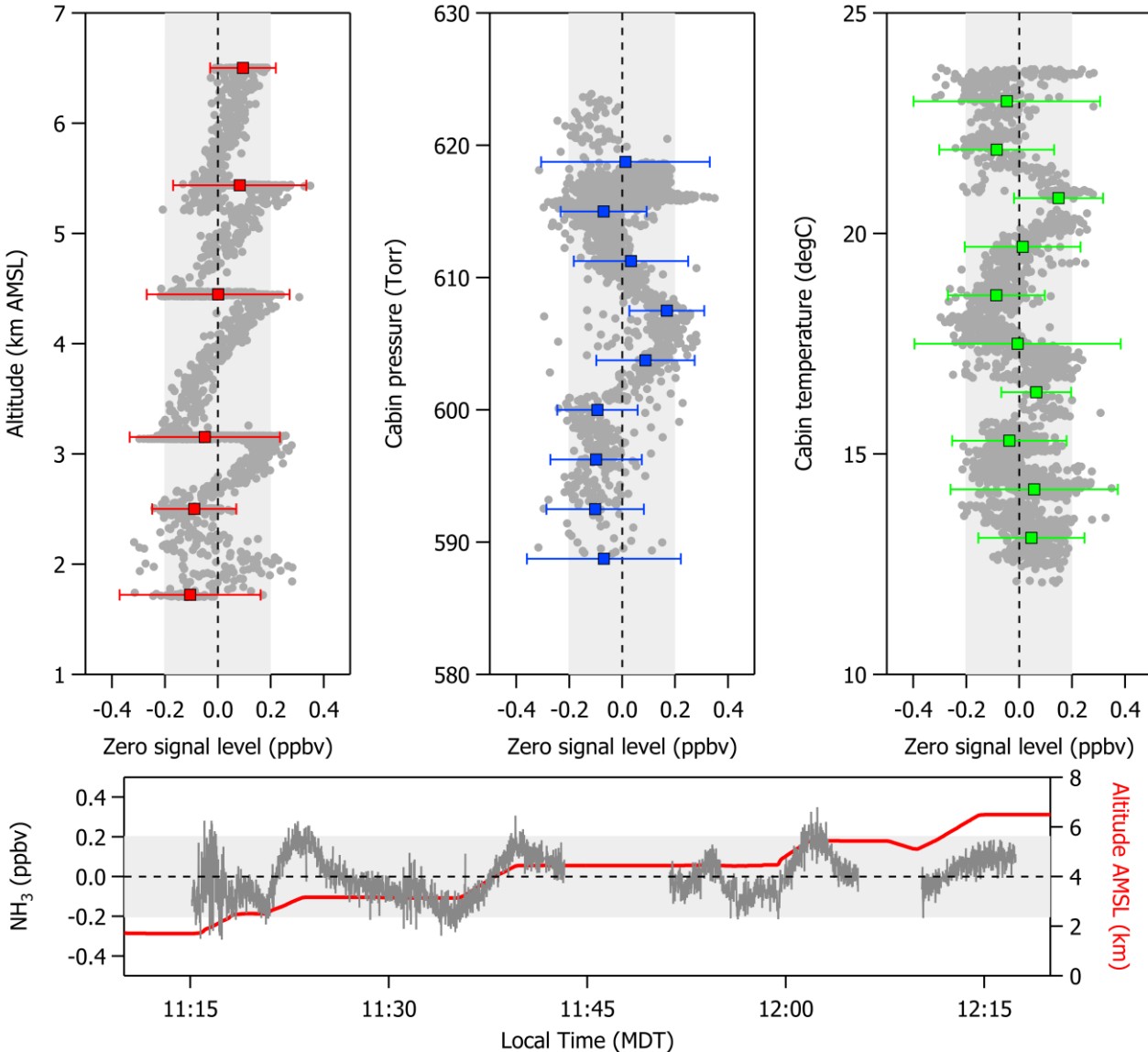

**Figure 3: In-flight variations in zero signal level (in units of ppbv of NH₃) with respect to changes in (a) altitude, (b) cabin pressure, and (c) cabin temperature. Gray symbols represent the 1 s average of all of the 10-Hz data points collected in flight while overblowing the inlet tip with NH₃-free air. Colored symbols and error bars represent the average NH₃ zero signal and 3σ standard deviation for each altitude step of an ascent profile, 5 Torr increments in cabin pressure, and 2°C increments in cabin temperature. Variations are largely within ±200 pptv (denoted by the light gray shaded areas). Gaps in the time series represent times when the instrument was performing a calibration or measuring ambient air.**

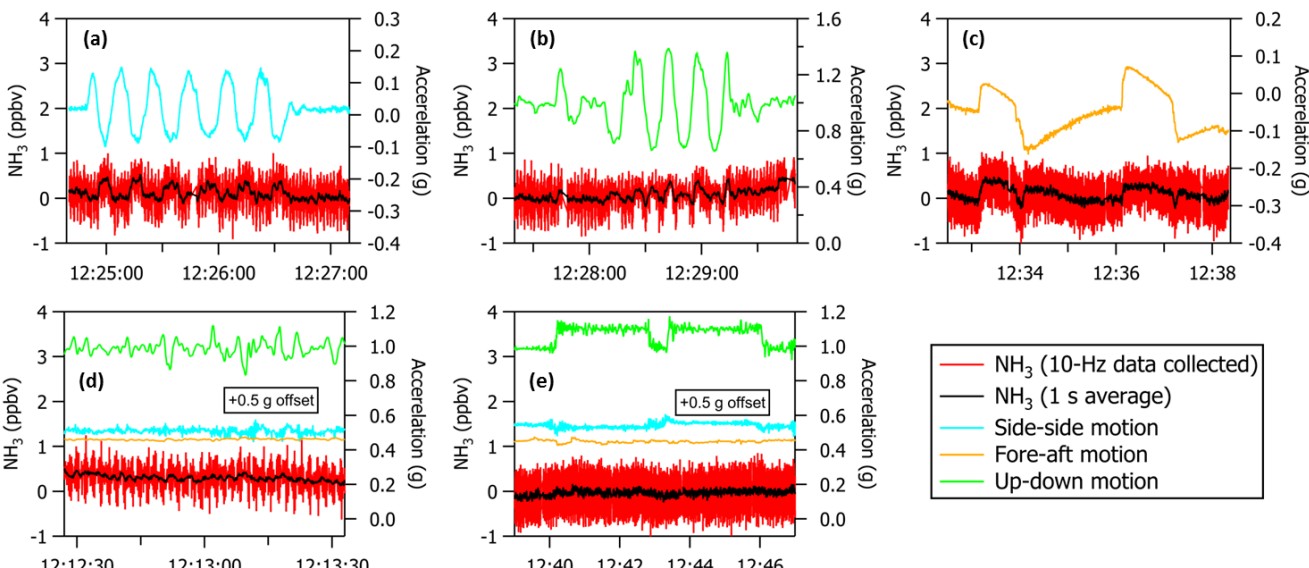

**Figure 4: Maneuvers performed at 2.6 km AGL in flight while overblowing the inlet tip with NH₃-free air show changes in NH₃ zero signal (in units of ppbv) with respect to aircraft accelerations (in units of g) in the (a) side-side, (b) up-down, and (c) fore-aft motions. Changes in NH₃ zero signal associated with (d) turbulence in the mixed boundary layer at 0.3 km AGL and (e) during turns at 2.6 km AGL are < 50 ppt when scaled by the ΔNH₃/g observed in each individual dimension during maneuvers. Side-side and fore-aft accelerations are offset by +0.5 g for display purposes in plots (d) and (e).**

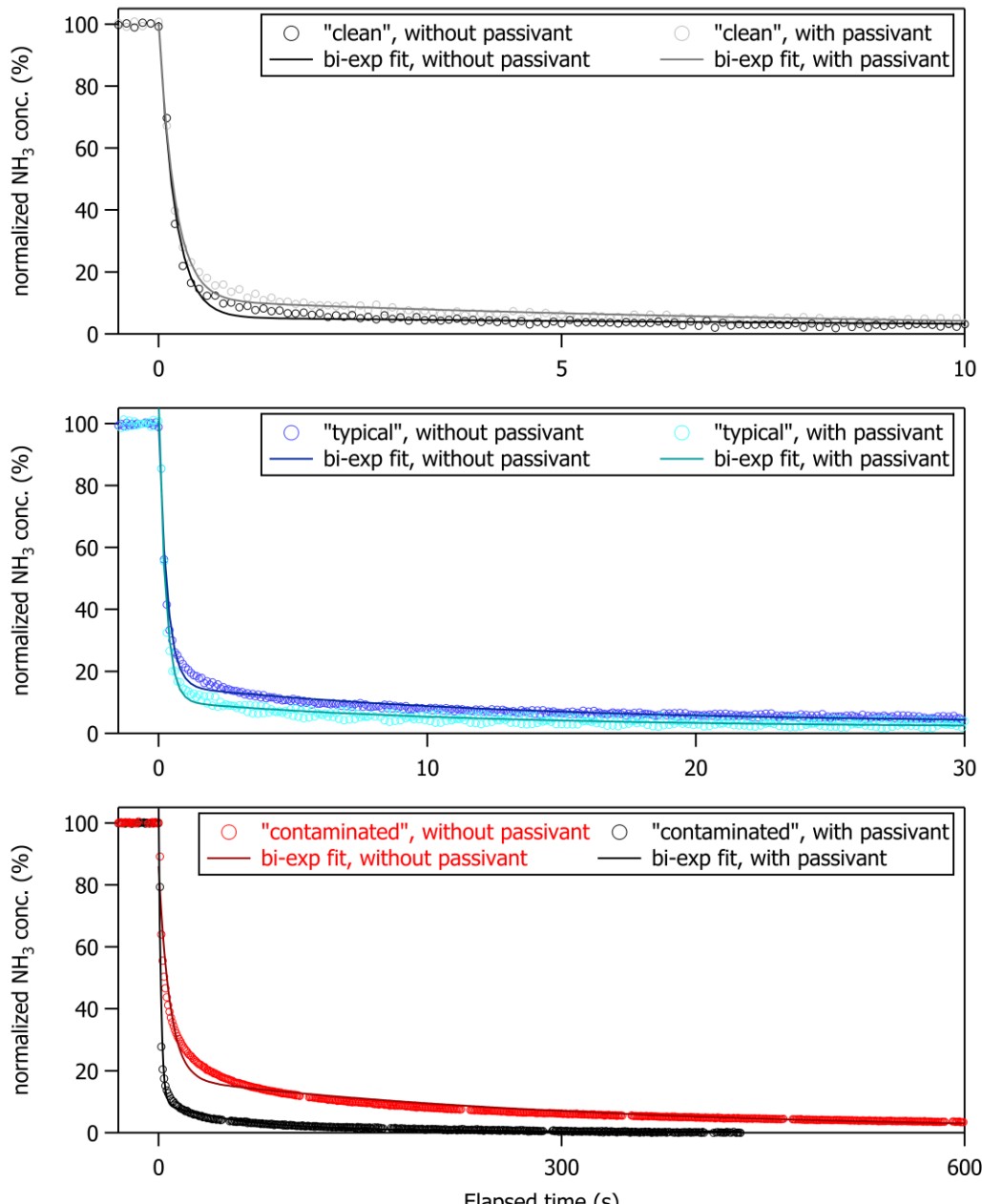

**Figure 5: Normalized NH₃ signal (in %) versus elapsed time (in seconds) following a step change in NH₃ mixing ratio generated by switching off calibration gas at *t*=0 s. Temporal profiles and associated bi-exponential fits are shown for the non-passivated and passivated instrument operated under (a) "clean", (b) "typical", and (c) "contaminated" sampling surface conditions. Fits ranged from *t₀* to 400, 1000, and 3000 s for the "clean", "typical", and "contaminated" cases, respectively, in accord with the elapsed time collected for each time profile.**

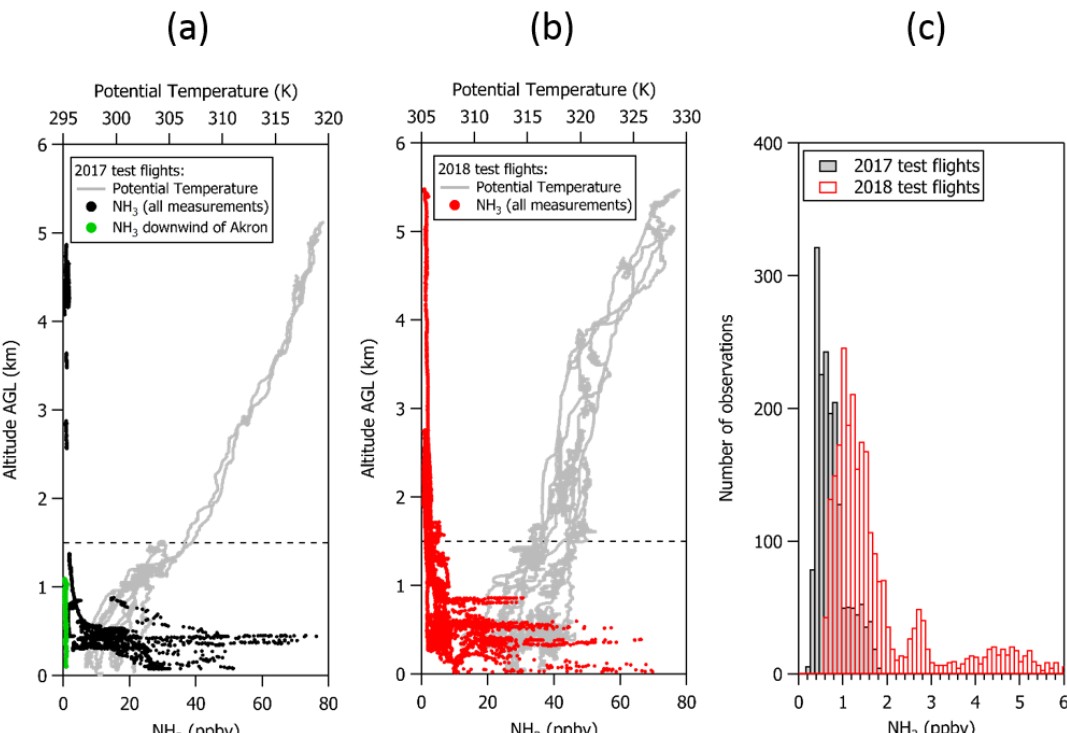

**Figure 6: Vertical profiles of NH₃ (in ppbv) and potential temperature (in K) from (a) the first and third test flight in 2017 and (b) the test flights in 2018 when the instrument was operated without passivant. NH₃ mixing ratios as high as 80 ppbv were observed in the mixed boundary layer during missed approaches at Greeley-Weld County Airport and over northeastern Colorado compared to average mixing ratios of ~0.8 ppbv near Akron, Colorado following several days of rain. (c) Histograms of the corresponding NH₃ measurements collected above 1.5 km AGL (dashed line) show that measurements were frequently larger than 200 ppt, especially measurements that were collected in the free troposphere.**

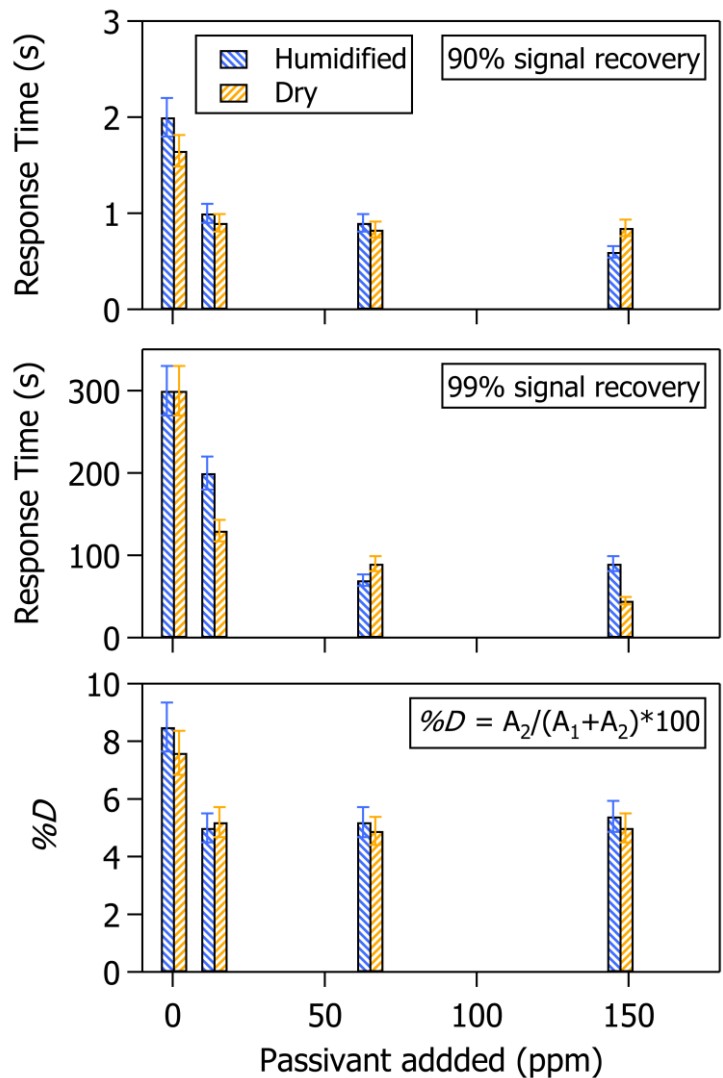

**Figure 7: Response times (*t*₉₀ and *t*₉₉) and %*D* associated with increasing C8 passivant addition for a step change in NH₃ mixing ratio generated on top of a dry versus humidified background.**

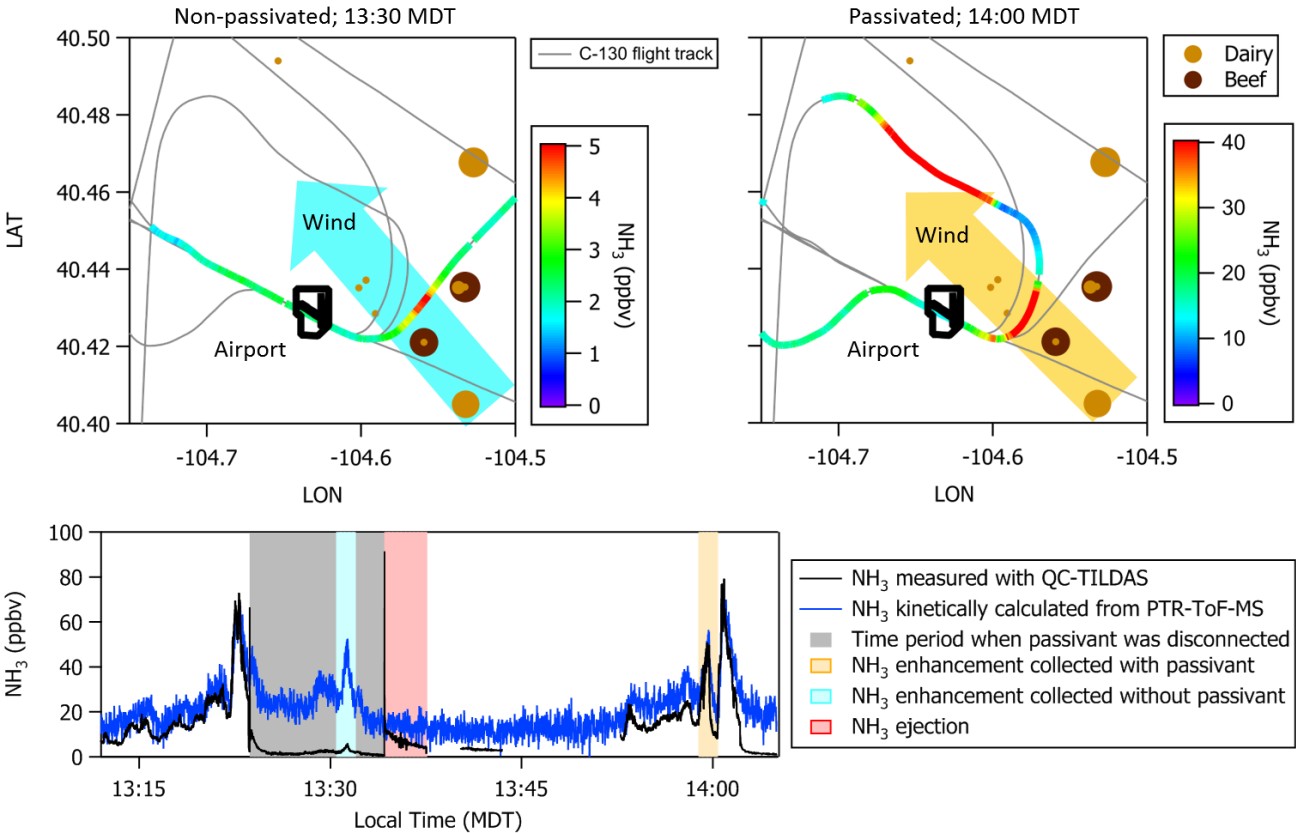

**Figure 8: (upper) C-130 flight tracks and measured winds during the 29 Sept 2017 test flight. Colored segments of the flight tracks highlight enhancements in NH₃ measured downwind of concentrated animal operations (brown symbols sized by head of cattle) located southeast of Greeley-Weld County Airport. (lower) Time series of 1-Hz NH₃ mixing ratios measured using the "contaminated" QC-TILDAS instrument and kinetically calculated from a raw instrument signal obtained simultaneously aboard the C-130 aircraft by a Proton Transfer Reaction Time of Flight Mass Spectrometer (PTR-ToF-MS). Portions of the flight when the "contaminated" QC-TILDAS was non-passivated (at 13:30 MDT) and passivated (at 14:00 MDT) are highlighted by colored shaded areas in the time series. Passivant was disconnected from the QC-TILDAS instrument system between 13:23 and 13:34 MDT (as indicated by the gray shaded area); NH₃ ejection is observed when passivant is re-added to the system (red shaded area).**

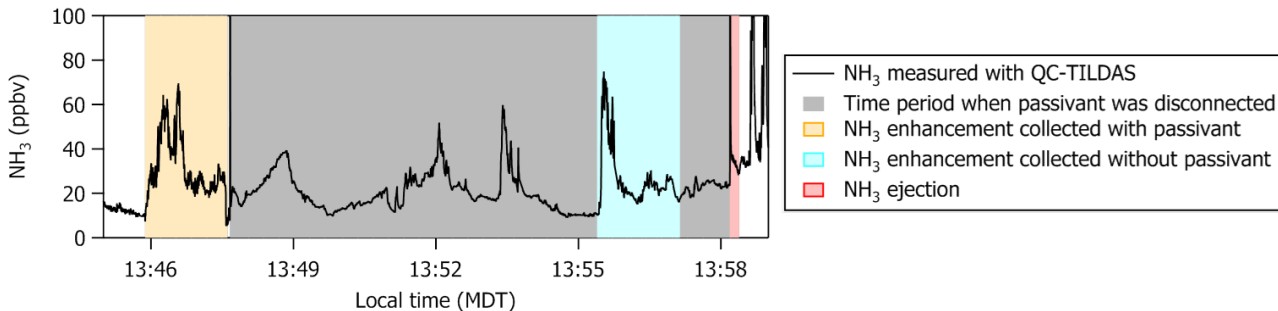

**Figure 9: Time series of 1-Hz NH₃ mixing ratios measured with a "clean" QC-TILDAS during the test flight on 13 July 2018. Portions of the flight when the QC-TILDAS was passivated (at 13:46 MDT) and non-passivated (at 13:58 MDT) are highlighted by colored shaded areas in the time series. Passivant was disconnected from the instrument system between 13:47 and 13:58 MDT (as indicated by the gray shaded area); NH₃ ejection is observed when passivant is re-added to the system (red shaded area).**

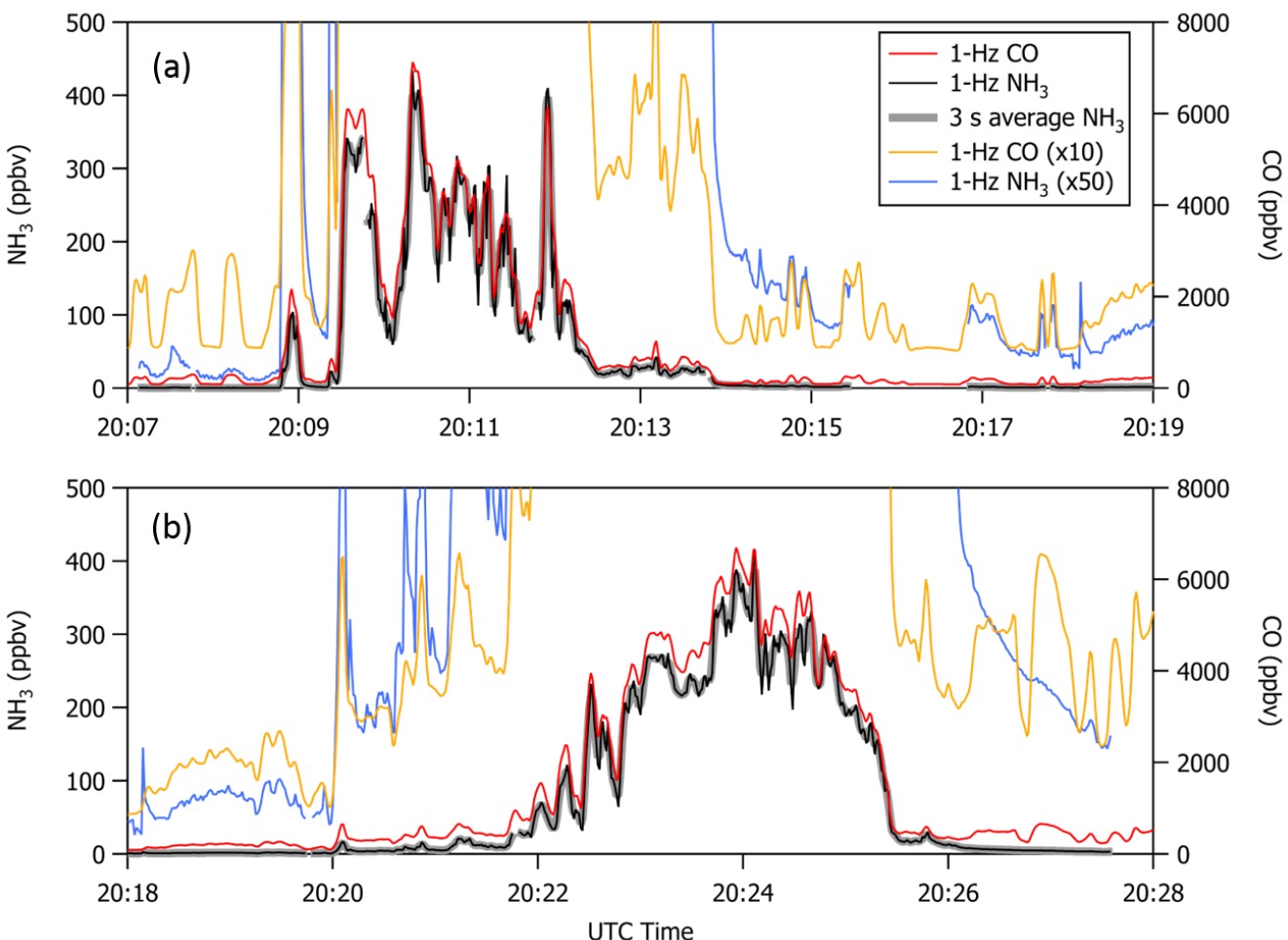

Figure 10: Time series of 1-Hz NH₃ (black lines) and CO (red lines) measured during a crosswind transect of the smoke plume from the South Sugarloaf Fire (RF15) on 26 August 2018. The transects represent nearly identical passes through the smoke plume with the only perturbation of the NH₃ instrument being operated (a) with passivant and (b) without passivant. Changes in fine structure features of NH₃ have the strongest $R^2$ correlation with CO when the NH₃ measurements are averaged to 3 s. A x50 magnified view of 1-Hz NH₃ (blue lines) and x10 magnified view of CO (orange lines) shows differences in background levels of NH₃ compared to CO before and after each plume transect.