# Peer review of "Evaluation of ambient ammonia measurements from a research aircraft using a closed-path QC-TILDAS spectrometer operated with active continuous passivation"

_Atmospheric Measurement Techniques, 2019_

## Referee Comment (RC1) · Anonymous Referee #2 · 7 Apr 2019

The manuscript 'Evaluation of ambient ammonia measurements from a research aircraft using a closed-path QC-TILDAS spectrometer operated with active continuous passivation' describes an instrument modified for airborne measurements of gas-phase ammonia ($NH_3$) and its characterization and performance during flights on the NSF/NCAR C-130 aircraft. Three modifications/additions were made to the previous $NH_3$ QC-TILDAS to improve its performance on an aircraft platform. These included mounting the instrument on a custom designed vibration isolation plate; addition of high frequency to an optical component; and the implementation of an active continu-

ous passivation system to improve time response. Implementation of the mechanical modifications minimized the motion sensitivity of the instrument reducing the 1 Hz precision to 60 ppt. While the active passivation, not absolutely necessary for a well maintained instrument, significantly improved the time response of a dirty instrument and will likely ease maintenance requirements for field campaigns in remote environments and/or over long periods of deployment time.

This paper is well written and appropriate for Atmospheric Measurement Techniques. It demonstrates an improvement in instrument performance from previous ground deployments and makes it a nice addition to the mix of available airborne NH3 instruments. I recommend publication after addressing a few minor comments listed below.

Specific Comments Page 1, Line 17 – Please give temperature range of the aircraft cabin instead of simply labeling it hot. This would give the reader context when considering using the instrument in other environments, such as, a trailer or tower.

Page 4, Line 26 – The inertial inlet description is sparse. The reader is not told what material it is constructed from until Page 7, Line 4 in the text or finds it buried in the caption of Figure 1. It is most logical for the reader to state that here in section 2.2.2 Inertial Inlet. In reference to figure 1, what is the size of the critical orifice? What temperature is it heated to? Also, it is a little misleading to say the 'The QC-TILDAS detector is typically operated with a heated inertial inlet . . .' since, from the literature and later in this manuscript, it seems other QC-TILDAS instruments measuring methane, carbon monoxide, ethane, for example, do not require or use an inertial inlet. Perhaps it should say 'The NH3 QC-TILDAS detector . . .'.

Page 6, Line 46 – While I applaud updating the cross-section used with optical absorption system described by Neuman et al., 2003, the number used here, 4.69e-18 cmˆ2, is slightly different than the 4.4e-18 cmˆ2 used in the 2003 manuscript not in contrast to. Furthermore, it appears that the +/- 0.03e-18 is the standard deviation of the average of the three cross sections listed here. What is the uncertainty of each and

then the uncertainty for the average? Are these two cross sections used in interpreting the absorption cell results here and in the previous manuscript within each other's uncertainty?

Page 7, line 4-7 – Since the sample flow rate is critical in calculating the calibration mixing ratios, the orifice size should be given in the text either here or in section 2.2.2.

Page 8, Section 4.3 – Some of the fits shown in Figure 5, particularly in panel b, do not match the data when it begins to flatten out. Would a triple exponential fit work better? Also, it would be easier for the reader to judge the fit if the data is present as symbols and the fit as a solid line.

Page 15, line 19 – I am concerned on the use of 'highlights the slightly faster time resolution of NH3 compared to CO.' without the qualifier 'for the instruments as configured here' to prevent misconstruing or over generalizing the observations in the future. It seems unlikely it would hold true if the instruments were configured with equal flow rates and tubing lengths.

Figure 1 – Please indicate the i.d of the PFA tubing used, the size of the critical orifice in the inertial inlet, and the temperature the inertial inlet is held at. From Figure 1 the length of the strut appears to be 12 cm, not the 36 cm stated in the text. What is the function of the aux. draw, which is not mentioned in the text or caption?

Figure 2 – With the lower panel in units of ppbˆ2 on a logarithmic scale, it is very difficult to see where the 60 ppt precision estimate comes from. Perhaps consider a lower right axis in ppt? Also, I suggest using the same scale in the lower panel for both plots.

[Figure]

---

## Author Comment (AC1) · 15 Apr 2019

We greatly thank Anonymous Reviewer #2 for their time and effort to provide detailed specific comments for this manuscript, which have greatly improved the accuracy and clarity of this work. Below is our response to each of the reviewer's specific comments.

We note the temperature of the aircraft cabin on Page 1, Line 17 (in the abstract) by adding "(e.g., average aircraft cabin temperatures expected to exceed 30 °C during summer deployments)".

We have amended Sect. 2.2.2 on Page 4, Line 26 and figure 1 with a more detailed description of the inertial inlet. This section now reads as: "The NH3 QC-TILDAS detector is typically operated with a heated inertial inlet positioned upstream of the spectrometer to provide filter-less separation of particles >300 nm from the sample stream, as shown in Fig. 1a. Coupling an inertial inlet with a QC-TILDAS has been well established following several laboratory and ground-based field experiments (Ellis et al., 2010; Ferrara et al., 2012; Tevlin et al., 2017; von Bobrutzki et al., 2010; Zöll et al., 2016). The inertial inlet is described in detail by Ellis et al. (2010) and Roscioli et al. (2016). Briefly, the inertial inlet used in these experiments consists of a quartz tube (12.7 mm o.d., 10.4 mm i.d.) with an integral, conical-shaped critical orifice roughly 1 mm in diameter positioned at about half the length of the tube, as shown in Fig. 1a. After passing through the orifice, gas (and particulates) are accelerated to a higher speed at a lower pressure (between 40 and 100 Torr) through the latter half of the 12.7 mm quartz tube, and then pass into a second quartz tube (25.2 mm o.d., 22.2 mm i.d.) that is sleeved around the 12.7 mm tube. The sample flow is split into two branches with approximately 90% of the total flow through the critical orifice (denoted by the blue arrow in Fig. 1a) being forced to make an 180 degree turn around the edge of the 12.7 mm tubing to continue to the spectrometer, and the other 10% (denoted by the orange arrow in Fig. 1a) being dumped via the straight section of 25.2 mm tube into the main pumping system. The inertia of particles with aerodynamic diameters greater than ~300 nm is too large to follow the gas stream around the 180 degree turn, thereby forcing the particles into the 10% of the flow stream that is directed to the pumping system. Ellis et al. (2010) reported that the inertial inlet, which acts like a form of virtual impactor, removes more than 50% of particles larger than 300 nm. A tee positioned immediately upstream of the critical orifice allows for pressure measurements using a baraton transducer (range 0-1000 Torr), which is used in determining the sample flow rate, and an auxiliary draw that allows the dead volume around the base of the conical-shaped critical orifice to be actively flushed. The flow rate of the auxiliary draw ranges from 160 to 500 sccm with changes in ambient pressure at the inlet tip. The inertial

inlet is housed in a fiberglass enclosure, with the inside of the enclosure maintained at 40 degrees C."

We agree with the reviewer that there is some confusion about how the uncertainty in the calibration source is determined in Sect. 2.3 on Page 6, Line 46. The text has been updated to include the uncertainties of the individual reported values from Froyd and Lovejoy (2012) (4.67 ± 0.08 x 10-18 cm2), Chen et al. (1998) (4.7 ± 0.5 x 10-18 cm2) and Cheng et al. (2006) (4.7 ± 0.5 x 10-18 cm2). Also, following careful consideration of the reviewer's comment, we now utilize the weighted average and associated propagated uncertainty of 4.7 ± 0.1 x 10-18 cm2 as a more appropriate treatment for combining the cross section values from the literature. The weighted mean absorption cross section utilized here is in agreement within the uncertainties with the value reported by Neuman et al. (2003) (e.g., 4.4 ± 0.3 x 10-18 cm2). We have also modified the text in Sect. 2.3 to clarify that the ±2% uncertainty in the weighted mean of the absorption cross section has been factored into the total estimated uncertainty (±7%) associated with the NH3 calibration source used in these experiments. Further, with only a ±2% uncertainty on the weighted average of the updated cross sections, the uncertainty in the absorption cross section is no longer the dominating factor in the total uncertainty of the calibration source using the NOAA UV optical absorption system. The ±7% uncertainty in the calibration source is factored into the overall instrument uncertainty (e.g., 200 pptv ± 12%), as described in Sect. 4.1. We have amended this portion of Sect 2.3 as follows: "In this work, we refine the uncertainty of the NOAA calibration of the emission rate of the permeation device used in these experiments by utilizing more recent assessments of the NH3 absorption cross section reported in the literature. Here, we use a weighted average of the NH3 absorption cross sections reported by Froyd and Lovejoy (2012) (4.67 ± 0.08 x 10-18 cm2), Chen et al. (1998) (4.7 ± 0.5 x 10-18 cm2) and Cheng et al. (2006) (4.7 ± 0.5 x 10-18 cm2). The weighted mean utilized here (4.7 ± 0.1 x 10-18 cm2) is in agreement within the uncertainties with the value reported by Neuman et al. (2003) (e.g., 4.4 ± 0.3 x 10-18 cm2). Combining in quadrature the ±2% uncertainty associated with the weighted mean of the absorption

cross section, the ±2.5% uncertainty in the stability of the permeation device between pre- and post-project calibrations with the NOAA UV optical absorption system, and a conservative estimate of ±6% for other sources of uncertainty associated with the NOAA calibration system, we determine a total estimated uncertainty of ±7% for the emission rate of the permeation device used in these experiments."

The size of the critical orifice ($\sim$1 mm) has been added to Sect. 2.2.2 on Page 7, line 4-7.

We too had considered in our initial analyses whether a triple exponential fit would work better for the time profiles shown in Fig. 5. Indeed, triple exponential fits do generate a more reasonable fit of the time profiles. However, in Section 4.3, we elected to report the results of bi-exponential fits in the original manuscript for the following reasons: 1) there is more physical basis for relating a bi-exponential fit to the experiments conducted in this work, 2) the results of a bi-exponential fit could be directly compared to similar results reported by Ellis et al. (2010) and Roscioli et al. (2016), and 3) the coefficient associated with the third time constant (A3) was <5% on average of the sum of the coefficients (e.g., [A3/(A1 + A2 + A3)]) and <23% on average of the sum of the coefficients associated with the latter two time constants (e.g., [A3/(A2 + A3)]). All the same, we agree that this discussion about the possibility of a triple exponential fit does have merit in this manuscript, and thus we have added the following discussion to the end of Sect. 4.3: "Indeed, a triple exponential decay with the functional form shown in Eq. (2): $y = y\_0 + A\_1 \exp((-(t-t\_0))/\tau\_1) + A\_2 \exp((-(t-t\_0))/\tau\_2) + A\_3 \exp((-(t-t\_0))/\tau\_3)$ produces better fits to the time profiles shown in Fig. 5. A triple exponential fit might have physical meaning in terms of the instrument time response if there is more than one time constant associated with the gas exchange rate through the sample flow pathway or if there is more than one time constant associated with the interaction of NH3 molecules with the sampling surfaces. In the case of multiple time constants associated with the gas exchange rate, it is possible that different residence times could arise from the different pressure regimes of the sample flow pathway (e.g.,

the portion of the sample flow path at ambient pressure upstream of the critical orifice in the inertial inlet versus the portion of the sample flow path downstream of the critical orifice at pressures between 40 and 100 Torr). In the case of NH3 molecules interacting with the sampling surfaces, additional time constants could be related to differing levels of cleanliness along the sample flow path. For example, inlet tubing and components were cleaned/replaced following contamination, but the optical cell in the QC-TILDAS was not; thus, more than one time constant might be most plausible, especially for the "typical" and "contaminated" time profiles that were collected following contamination. All the same, we elect to report the results of the bi-exponential fits in this work for the following reasons: 1) there is more physical basis for relating a bi-exponential fit to the passivation experiments conducted in this work, 2) the results can be directly compared to the results of bi-exponential fits for similar instrumentation reported by Ellis et al. (2010) and Roscioli et al. (2016), and 3) the coefficient associated with the third time constant (A3) is small (e.g., A3 is <5% on average of the sum of the coefficients (e.g., [A3/(A1 + A2 + A3)]) and <23% on average of the sum of the coefficients associated with the latter two time constants (e.g., [A3/(A2 + A3)])."

Figure 5 has been updated with symbols for the data and solid lines for the fits according to the reviewer's suggestion.

The phrase "for the instruments as configured here" has been added to the text on Page 15, Line 19.

Figure 1a, its figure caption, and relevant parts of Sect. 2.2.3 have been updated with the i.d of the PFA tubing used, the size of the critical orifice in the inertial inlet, and the temperature the inertial inlet. In general, 3/8" o.d., $\frac{1}{4}$" i.d. PFA tubing is used for the sample flow path. In terms of tubing lengths, we had intended to indicate that the full length of the sample flow path from the inlet tip to the inertial inlet is 107 cm. We agree that the current description of the inlet lengths is confusing and may not have been accurately labelled in the original version of the manuscript; therefore, we have modified Fig. 1a and Sect. 2.2.3 to clarify. The caption for Fig. 1 now reads as: "(a)

Schematic of the instrument as configured for flight on the NSF/NCAR C-130 aircraft. The sample flow path (blue arrows) starts at the inlet tip, which is a short piece of 3/8" o.d., $\frac{1}{4}$" i.d. PFA tubing that protrudes slightly from the face of the aircraft inlet strut. A PFA injection block (1/4" i.d.) housed inside the aircraft inlet strut allows calibration gases and passivant to be added to the sample stream within a few centimetres of the inlet tip. A 71-cm length of 3/8" o.d., $\frac{1}{4}$" i.d. PFA tubing then directs ambient air into a quartz inertial inlet where particles >300 nm are separated from the sample stream, and a 36-cm length of 3/8" o.d., $\frac{1}{4}$" i.d. PFA tubing directs the sample flow from the inertial inlet to the QC-TILDAS detector. The particle-rich stream is pumped away (orange arrows). The sample flow path is heated to 40 ЁŽC where possible to minimize interactions of NH3 with sampling surfaces. The sample flow path is purged overnight in the reverse flow direction with 40 sccm of N2 injected near the pressure control valve. A 0-1000 Torr range baratron (denoted as P) measures pressure just upstream of the critical orifice in the inertial inlet for active continuous determination of the sample flow rate. An auxiliary draw acts to flush the dead volume formed near the base of the conical-shaped critical orifice in the inertial inlet. (b) Solid model of the QC-TILDAS vibration isolation mounting plate. (c) Photograph of the QC-TILDAS mounted to the vibration isolation plate while installed aboard the C-130 aircraft. (d) Photograph of the impinger used for active continuous passivant addition."

According to the reviewer's suggestions, Figure 2 has been updated to also include a lower right axis in ppt, and both of the lower panels of the figure have been updated to display the same scale on the the right and left y-axes. The caption now reads as: "10-Hz measurements collected while overblowing NH3-free air at the inlet tip (a) in flight in the boundary layer near 1.4 km AGL and (b) on the ground in the laboratory. Upper traces represent the raw 10-Hz data collected; lower traces depict the Allan variance in ppb2 (left axis) and Allan deviation in ppt (right axis) of the corresponding data set. An offset was applied to the Allan variance in panel (a) to reflect the vibration applied to the laser objective to reduce motion sensitivity in flight; the vibration was not applied during laboratory tests depicted in panel (b)."

[Figure]

Figure 1

**(a)**

Outside | In the cabin

6 cm | 6 cm | 24 cm | 71 cm

Inlet tip

Aircraft Inlet Strut

aux. draw

P

~100 Torr

critical orifice

gas stream

particle stream → 1.5 LPM → to pump

Inertial Inlet

Pressure Control Valve

Overnight purge →

10 SLPM

76 m 40 Torr

detector

36 cm

QC Laser

QC-TILDAS

$NH_3$-free air | Passivation | Calibration

**(d)** | **(c)** | **(b)**

LOT NO.

**Fig. 1.**

Figure 2

[Figure]

**Fig. 2.**

Figure 5

[Figure]

**Fig. 3.**

---

## Short Comment (SC1) · 4 May 2019

**Comment on "Evaluation of ambient ammonia measurements from a research aircraft using a closed-path QC-TILDAS spectrometer operated with active continuous passivation" by Da Pan, Xuehui Guo, and Mark Zondlo**

The manuscript assesses the performance of a closed-path, airborne-based ammonia instrument (Aerodyne Res. Inc.) by demonstrating the performance of active passivation under flight conditions. Ammonia is incredibly challenging to measure anywhere (with any technique) due to the significant adsorption/desorption effects on instrument/inlet surfaces, particularly on an airborne-based platform where temperatures, humidities, and ambient ammonia concentrations can vary dramatically. The authors show greatly improved performance when using the passivated versus unpassivated flows for sampling large NH3 concentrations (10s-100s ppbv NH3) from farms and biomass burning plume. The documentation of the instrument performance versus flight maneuvers was particularly valuable. This manuscript represents a large advance in airborne-based ammonia measurements, and the authors' experiences on using passivant additions in addition to /in lieu of frequent cleaning are important for future implementation of closed-path ammonia instruments in specific but also ammonia sampling more generally (laboratory experiments, calibrations, etc.). However, there remain some areas that require greater clarification to put the research in the proper context.

**1. The response times, and applicability, to smaller $NH_3$ variations should be discussed (and backgrounds relevant to very low free tropospheric values, <ppbv). The detection limit needs better justification.**

In this study, the step change of $NH_3$ was created by turning off the calibration gas. The change is around 85 – 115 ppbv. This variation is uncommon for sites away from source regions. At high $NH_3$ concentrations and large variations, $NH_3$ observations may be less impacted by surface interaction because a "clean" sampling line only has a finite number of adsorption sites which could be quickly fill up under this condition. This effect has been reported by Ellis et al.(1), and it may explain why passivation did not help to increase the response time of the instrument. At low $NH_3$ concentrations, a greater fraction of $NH_3$ molecules may interact with the inner surface.

Roscioli et al. showed that $t_{90}$ of a similar instrument was 12 sec for a step change of 3 ppbv (from 0 to 3 ppbv) without passivation (2). When 4 ppmv passivant was applied, $t_{90}$ decreased to 2 sec for the same step change. The instrument can be considered clean since it was flushed with $NH_3$-free and low $NH_3$ gases. Therefore, even a "clean" closed-path instrument may not be capable for high-frequency (>1 Hz) field application with small $NH_3$ variations without passivation, and passivant additions may still not work for fast operation (> 1 Hz) under clean conditions. The authors should discuss in more details about the applicability and effectiveness of passivation to field applications with relatively low $NH_3$ concentrations (e.g. flux measurements in rural area and airborne observations away from sources).

Fig. 10: Because of the nature of the very large concentrations measured, it is hard to discern just how well the instrument/technique can observe cleaner, free tropospheric ammonia levels after seeing large plumes. For example, while the correlation is impressive in Fig. 10 and shows the value of this overall approach, with a several second response time for NH3, the "peaks" and "valleys" may still be attenuated to some extent. It would be helpful to show a plot against a true 1 Hz tracer correlation instead of two instruments with 2-3 second response times.

There are also some differences between the start and end of the plumes in Fig. 10 in terms of the NH3/CO ratio. As one progresses in the plume, the NH3/CO ratio seems to get higher, which would be consistent if the background is growing. Differences in plume chemistry across the transect may be a reason for this, too. However, outside the plume (start/end of timeseries), the NH3/CO level isn't the same, either.

Related to this overall point, on the large y-scale axis (500 ppbv NH3) in Fig. 10, while the concentration looks to be "close" to zero, in reality on this scale it could be numerous ppbv NH3. Even if the instrument response time is nominally on the order of a few seconds, going from 400 ppbv NH3 to sub-ppbv NH3 could take a long time and may result in biases in clean conditions in the free troposphere. It may be helpful to show a vertical profile of NH3 in the ascent out of Greeley (where high agricultural emission concentrations exist) and compare it to another, short-lived boundary layer tracer that would be highly enhanced in the boundary layer vs. the free troposphere. On the left below, I show a vertical profile of NH3 from the NASA DISCOVER-AQ California in the San Joaquin Valley (taken from Sun et al., 2015), where the importance of the authors' large improvement is clearly validated (needed), but where concerns of going very high NH3 to nominally sub-ppbv NH3 at high altitude could still be an issue even with this approach. Also attached on the right is a profile of NO2, ethane, and CN in the Greeley area from DISCOVER-AQ Colorado. I'm not sure if these are necessarily the best tracers per se from a quick look, but one can see very sharp gradients at the top of the boundary layer, and it would be illustrative to see how the shape of the NH3 profile compares to other ~ short-lived tracers when ascending/descending across the mixed layer height. WE-CAN should have plenty of such measurements on the C-130 to compare.

Figure 3: It is hard to see on Fig. 3a, but the constant altitude segments seem to show quite a bit of variability in the background, say, from -0.2 to +0.2 ppbv NH3 within an altitude level for a given 1 Hz measurement. This calls to question as well the accuracy of anything < 0.5 ppbv, given that the background is changing by 0.4 ppbv. What were the ascent rates/g's after each constant altitude leg? The 50 pptv NH3 sensitivity to typical flight maneuvers mentioned in the text doesn't seem consistent with Fig. 3a. If most of the variability is due to the ascent portions (g-forces), then perhaps a 1 Hz timeseries of the constant leg would be helpful. Also, how "polluted" was NH3 prior to overfilling the inlet for a zero for these flights upon takeoff in Broomfield? How often was it zeroed vs. sampling? 10% duty cycle? Entire flight? 50%? The wording wasn't clear in the text for this portion of the flight.

Table 2: I really appreciated the mass balance in Table 2/discussion (neat experiment!), though even here differences of ~ 10% of counting molecules still could mean significant backgrounds still exist relative to very clean conditions (though I recognize this mass balance counting is within the instrument uncertainty).

In summary of all of the above, taking 3 times the 1 Hz precision doesn't seem justified for the detection limit, nor an assessment of instrument accuracy at low concentrations. It seems the instrument is well-designed for fires/agriculture but future work is still needed for clean conditions after such large plumes (or more justification in the manuscript). This is particularly true when going from dirty to clean conditions, given the many sampling biases that still may exist for ammonia.

[Figure]

(Sun et al., 2015)

**3. Validity of using bi-exponential decay model and meaning of the parameters should be addressed:**

The bi-exponential decay model is essential to the discussion about instrument response time. The authors used the bi-exponential decay model to determine the response times of the instrument to associate gas exchange and the interaction of $NH_3$ molecules with sampling surfaces. The fit results were also used to extrapolate the 90% and 99% signal recovery times ($t_{90}$ and $t_{99}$). Therefore, it is necessary to address the validity of the bi-exponential decay model.

The bi-exponential decay model was first introduced to characterize response time of QC-TILDAS to $NH_3$ changes by Ellis et al.(1). However, the validity of the model was not discussed in the original work. Here, we propose to use the following a simplified surface-air exchange model to derive the bi-exponential decay model and discuss its validity.

After a step change, changes of the mixing ratio of $NH_3$ inside the instrument $\chi$ is caused by 1) the difference of $NH_3$ mixing ratio between the gas currently inside the chamber and the newly introduced gas ($\chi_0$); 2) adsorption or desorption to the inner surface of the instrument. These processes can be expressed as

$$\frac{d\chi}{dt} = \frac{Q}{V}(\chi_0 - \chi) + \kappa(\chi_s(t) - \chi)$$

where $Q$ is the flow rate and $V$ is the inner volume of the instrument; $\kappa$ is the conductance of $NH_3$ between surface and air interface; $\chi_s$ is the compensation point of the inner surface (adsorption occurs when $\chi > \chi_s$, desorption occurs when $\chi < \chi_s$). The compensation point is a function of time and its variation depends on historical changes of $NH_3$ concentration inside the instrument. When there are no phase changes and chemical reactions, and the surface is not saturated by $NH_3$ or exhausted of $NH_3$ during the process, $\chi_s$ could be simplified as

$$\frac{d\chi_s}{dt} = \kappa(\chi - \chi_s).$$

When the surface is clean such that $\chi_s \ll \chi$, $\chi_s$ equation can be approximated as

$$\frac{d\chi_s}{dt} = \kappa\chi.$$

For the step change described in this study, $\chi_0=0$. Combine all the equations, we have

$$\frac{d^2\chi}{dt^2} + \left(\frac{Q}{V} + \kappa\right)\frac{d\chi}{dt} = \kappa^2\chi.$$

The general solution to the differential equation is

$$\chi = A_1 \exp\left[-\frac{1}{2}\left(\sqrt{\left(\frac{Q}{V}\right)^2 + 2\left(\frac{Q}{V}\right)\kappa + 5\kappa^2} + \left(\frac{Q}{V}\right) + \kappa\right)t\right]$$

$$+ A_2 \exp\left[\frac{1}{2}\left(\sqrt{\left(\frac{Q}{V}\right)^2 + 2\left(\frac{Q}{V}\right)\kappa + 5\kappa^2} - \left(\frac{Q}{V}\right) - \kappa\right)t\right]$$

$$= A_1 \exp\left(-\frac{t}{\tau_1}\right) + A_2 \exp\left(-\frac{t}{\tau_2}\right)$$

It can be seen from above derivation that the bi-exponential decay model approximates the universal solution of the differential equations, but it only works under certain conditions - the most important one is the relative cleanliness of the surface. After certain time, $\chi$ will approach $\chi_s$ and the solution to the differential equations becomes significantly more complicated and is unlikely to follow bi-exponential decay model. The authors should clarify the applicability of the bi-exponential decay model.

Given the validity of the bi-exponential decay model, it may be more reliable to derive $t_{90}$ and $t_{99}$ using observed time series directly if the measurements are not noisy.

$\tau_1$ and $\tau_2$ represents the combined effects of both gas exchange and air-surface exchange instead of representing the effects separately. Therefore, the statements about $\tau_1$ and $\tau_2$ from line 22 to line 24 on page 10 should be removed.

**3. Uncertainty of response time may not be representative:**

The uncertainty of the response time is currently estimated using error propagation of the fitted results. However, given the exponential natural of the issue, parameters like $t_{90}$ and $t_{99}$ may have a skewed distribution (i.e. log-normal distribution) with a long tail. This behavior may not be correctly captured by error propagation method. If fitted results are used, Monte Carlo method should be used. If the real time series is used, $t_{90}$ could be estimated as the standard deviation of time stamps of observations with $NH_3$ between the $90 - \sigma_{obs}$ percentile and the $90 + \sigma_{obs}$ percentile.
* * *
Summary:  The authors have shown a marked improvement in the use of airborne-based ammonia measurements. In fact, these measurements are the most impressive and reliable to date in the literature and have set a new standard for all future campaigns (airborne and ground-based). The technique has applicability to Picarro and other closed-path sensors, as well as calibration methods for open-path sensors (which, indeed, have to be "closed" typically for calibration). However, there are still many gremlins for airborne ammonia, particularly with its enormous dynamic range in concentration

and adsorption issues, which get magnified for sub-ppbv NH3 levels that are expected in the free (or at least upper) troposphere (Asian UT monsoon levels excepted, possibly). I hope the points above allow for some clarifications that strengthen the manuscript.

****others****

Page 11, line 4-10: The manuscript never mentions how the boundary layer height was determined. Was it known accurately in each case or simply assumed to be <1 km?

Page 12, line 40-45: The authors should be aware that the relationship between water content and $NH_3$ adsorption is not necessarily linear. The interaction mechanism varies significantly depending on the amount of water present. A previous study by Vaittinen et al., 2018, has demonstrated this. Therefore, the two scenarios (dry vs 80% humidified) tested here may not be representative enough to tell the whole story.

Page 13, line 40-43: It is not clear what criteria the authors used to determine that the $NH_3$ transmission shows little difference between the non-passivant and with-passivant transects. (By the response time/maximum reading/amount of the $NH_3$ measured?)

Figure 6a: It would be helpful to show the exact boundary layer height for this profile for better clarity.

Figure 8 (upper): I am confused by the brown circles/dots labeled as beef and dairy. Do the small dots indicate smaller facilities as compared to the large circles? The two circles on the upper and lower right presumably refer to dairy but there is a dot in the center of each circle. Does this mean the facilities have both dairy and beef?

The total power output of the system should be described, since power seemed to be an issue even on an aircraft.

Abstract: "Flight-ready" in the abstract seems redundant for the topic; also "custom" is mentioned three times in the first sentence.

For the Allan plot, what offset was applied and how much? As written, it is confusing.

SilcoNert 2000 has been shown to work very well for ammonia and water vapor adsorption – can the authors – Pogany et al., Meas. Sci. Tech., https://iopscience.iop.org/article/10.1088/0957-0233/27/11/115012/meta

**References**

1.  Ellis R*, et al.* (2010) Characterizing a quantum cascade tunable infrared laser differential absorption spectrometer (QC-TILDAS) for measurements of atmospheric ammonia. *Atmospheric Measurement Techniques* 3(2):397-406.
2.  Roscioli J, Zahniser M, Nelson D, Herndon S, & Kolb C (2015) New Approaches to measuring sticky molecules: improvement of instrumental response times using active passivation. *The Journal of Physical Chemistry A* 120(9):1347-1357.

3.  Sun, K. et al., JGR, 2015.

---

## Referee Comment (RC2) · Anonymous Referee #1 · 14 May 2019

This is a clearly written manuscript that documents the performance of a closed-path absorption spectrometer for the measurement of NH3 aboard an aircraft, with a particular emphasis on the utility of an active passivation technique. The manuscript is appropriate for AMT, and should be published after addressing the following issues:

General comments: In several places (e.g. P2, L29; P3, L32; P4, L27; P4, L33), the manuscript uses the term 'detector' to refer to the instrument itself, whereas in other places, including in Figure 1, 'detector' is used to refer to the MCT detector that collects the transmitted radiation, but is not in contact with the gas flow of the system.

[Figure]

I found the more general use of the term somewhat distracting/confusing and would suggest using either 'QC-TILDAS', 'spectrometer', or 'instrument' in whichever case is appropriate.

In the section discussing the vibrational and structural issues, the authors mention (P5, L35) 'reinforcing' the 'strain relief'. It was not clear to me if this involved providing more slack in the sampling lines, or making them more rigid. A little more information would be helpful. In Section 4.1, the authors describe a zero overflow experiment. Does the (> 500 sccm) refer to the difference between the flow of zero air being delivered and the flow pulled by the instrument? Clarification would be useful.

In Section 5.1, the authors explore the impact of inlet aging and the use of the passivant on the time response of the system. While the proportion of the time response governed by the slow, "adsorptive", term was typically quite low (D < 10%), the magnitude of the step change in concentration was rather large (50 ppb), so caution should be taken in extrapolating that result to ambient observations.

Section 5.2.1 presents an interesting case study in which two intercepts of an intense NH3 plume led to much different sampling efficiencies depending on whether or not the passivant was being added to the inlet, as the result of a recent pre-flight contamination. I found this section a bit confusing because the time period between 13:20 and 13:23, when both the QC-TILDAS and the PTR-TOF-MS measured enhanced (and consistent) NH3 is not described. One infers that the passivant was being used at the time, however it's not clear.

Specific comments:

P2, L11 – NH3 is regulated under the Gothenburg protocol in some parts of the world.

P3, L33 – The 'D' in QC-TILDAS has traditionally stood for 'differential', not 'direct'

P7, L41 (and subsequently) 'Hydroscopic' should be 'hygroscopic'

Figure 8 caption – 'colorded' should be 'colored'

[Figure]

---

## Author Comment (AC3) · 11 Jun 2019

Response to Interactive Short Comment SC1 on "Evaluation of ambient ammonia measurements from a research aircraft using a closed- path QC-TILDAS spectrometer operated with active continuous passivation" by Da Pan, Xuehui Guo, and Mark Zondlo

We thank the Zondlo group for their extremely thoughtful and constructive feedback. This discussion exemplifies the value of this particular forum. Please find the commenters suggestions highlighted in bold font below followed by our reply in plain text.

The manuscript assesses the performance of a closed-path, airborne-based ammonia instrument (Aerodyne Res. Inc.) by demonstrating the performance of active passivation under flight conditions. Ammonia is incredibly challenging to measure anywhere (with any technique) due to the significant adsorption/desorption effects on instrument/inlet surfaces, particularly on an airborne-based platform where temperatures, humidities, and ambient ammonia concentrations can vary dramatically. The authors show greatly improved performance when using the passivated versus unpassivated flows for sampling large NH3 concentrations (10s-100s ppbv NH3) from farms and biomass burning plume. The documentation of the instrument performance versus flight maneuvers was particularly valuable. This manuscript represents a large advance in airborne-based ammonia measurements, and the authors' experiences on using passivant additions in addition to /in lieu of frequent cleaning are important for future implementation of closed-path ammonia instruments in specific but also ammonia sampling more generally (laboratory experiments, calibrations, etc.). However, there remain some areas that require greater clarification to put the research in the proper context.

**1**. The response times, and applicability, to smaller NH3 variations should be discussed (and backgrounds relevant to very low free tropospheric values, <ppbv). The detection limit needs better justification.

Thank you for these comments. See our responses below each specific comment/suggestion related to these topics.

In this study, the step change of NH3 was created by turning off the calibration gas. The change is around 85 – 115 ppbv. This variation is uncommon for sites away from source regions. At high NH3 concentrations and large variations, NH3 observations may be less impacted by surface interaction because a "clean" sampling line only has a finite number of adsorption sites which could be quickly fill up under this condition. This effect has been reported by Ellis et al.(1), and it may explain why passivation did not help to increase the response time of the instrument. At low NH3 concentrations, a greater fraction of NH3 molecules may interact with the inner surface.

We agree that a limitation of these experiments is a lack of tests with lower mixing ratios of NH3, which could affect the applicability of these results to some specific applications, namely the remote free troposphere. As the commenters point out, the effect of decreasing instrument response time with increasing step change concentration has already been extensively characterized by Ellis et al. Therefore, the main goal of this paper is to further the works of Ellis et al. (2010) and Roscioli et al. (2016) by characterizing the effects of passivant addition on instrument time response in flight. For this discussion we also note that the results of Ellis et al. were reproducible in our own experiments when time profiles were generated with various levels of step change concentrations. One of which included a step change from ambient levels of NH3 (e.g., between 5 and 12 ppbv) to zero, which was generated by switching on the overflow injection of NH3-free air at the inlet tip after a period of sampling near

homogeneous levels of ambient  $NH_3$  for several minutes. All the same, we agree with the commenters that the text should be amended to clarify that these results may be most applicable to near source sampling because the instrument utilized for these experiments was optimized for sampling large and rapid gradients of  $NH_3$  in smoke.

In response to this and the next comment, we have added the following text to Section 4.4:

"All the same, further measurements are recommended for assessing sampling biases that could arise during field measurements of low mixing ratios of NH3 in clean environments following long periods of exposure to near source level concentrations. The potential for an adsorption-related "memory effect" of NH3 (e.g., Williams et al., 1992) on the sampling surfaces following long-term exposure to high concentrations of NH3 is discussed in following sections."

**And we added this text to Section 5.1.1:**

"In this study, the instrument response is rigorously tested with a single step change of NH3 created by turning off a 50 ppbv calibration gas mixture. We note that such large variations in NH3 mixing ratio may not been full applicable to field measurements in unpolluted regions away from concentrated sources of NH3. As described by Ellis et al. (2010), large gradients in NH3 may be less impacted by surface interactions because "clean" sampling surfaces only have a finite number of adsorption sites that could be quickly filled under high NH3 conditions. At lower NH3 concentrations, a greater fraction of NH3 molecules may interact with the inner surfaces. This could explain why passivation did not help to increase the response time of the instrument."

Roscioli et al. showed that  $t_{90}$  of a similar instrument was 12 sec for a step change of 3 ppbv (from 0 to 3 ppbv) without passivation (2). When 4 ppmv passivant was applied,  $t_{90}$  decreased to 2 sec for the same step change. The instrument can be considered clean since it was flushed with NH3-free and low NH3 gases. Therefore, even a "clean" closed-path instrument may not be capable for high-frequency (>1 Hz) field application with small NH3 variations without passivation, and passivant additions may still not work for fast operation (> 1 Hz) under clean conditions. The authors should discuss in more details about the applicability and effectiveness of passivation to field applications with relatively low NH3 concentrations (e.g. flux measurements in rural area and airborne observations away from sources).

We greatly appreciate your comments with respect to the comparison of time responses that we collected with those reported by Roscioli et al. (2016). While we frame the results of the time response tests so that they can be directly compared to the works of Ellis et al. (2010) and Roscioli et al. (2016), we believe that it is difficult to compare the level of cleanliness of two different instrument systems. In this particular case, it may not be fair to say that our "clean" (aka. pristine, out-of-the-box, never-used-in-the-field) instrument is the same level of cleanliness as the copiously flushed, yet previously deployed instrument used by Roscioli et al. We are fortunate to know the history of both instruments, and surmise that the QC-TILDAS utilized by Roscioli et al. (2016) more closely resembles an instrument with a mid-level of cleanliness (similar to what we define as a "typical" operating condition) rather than the pristine, out-of-the-box condition that we referred to as "clean" in this work. This is because the instrument utilized by Roscioli et al. was a dual channel QC-TILDAS optimized to measure NH3 as well as HNO3 (aka. two well-known sticky molecules). Prior to lab tests, the instrument was deployed aboard the NSF/NCAR C-130 research aircraft in the 2014 FRAPPE field campaign where it had been exposed to near source levels of NH3 and urban emissions in the Colorado Front Range. Further, during the lab tests, the dual channel instrument was used to test several strong bases as passivant agents for NH3 and

several strong acids as passivant agents for HNO3. In our experience with contamination and cleaning of sampling surfaces, we found that the instrument flow path could only be truly cleaned by replacing tubing and inlet components where possible and by performing several cycles of ethanol/water rinse followed by week-long periods of flushing the sample flow path with NH3-free air. The dual channel instrument was solely flushed with NH3-free air prior to lab experiments, but sampling surfaces were not systematically cleaned and replaced. For these reasons, we categorize the instrument used by Roscioli et al. as an instrument operating under a middle level or cleanliness, like our "typical" operating condition. We also point out that Roscioli et al. (2016) found that the NH3 time response of the instrument continually improved with increasing passivation concentration, presumably with an eventual lower limit somewhere at or above the volumetric flush time. Therefore, the degree to which one wants to achieve >1 Hz sensitivity can be determined by the amount of passivant added to the sampling system. In the case of the flights discussed here, we made a compromise between passivant use and time response in order to achieve a reasonable temporal response while not using an excessive amount of passivant. All the same, we agree with the commenters that further discussion is warranted about the applicability and effectiveness of active continuous passivation to field applications with relatively low NH3 concentrations, such as flux measurements in rural areas and measurements away from sources. We now include this statement in Sect. 4 and reiterate this point in Sect. 5. See the text pasted in response to comment above.

Fig. 10: Because of the nature of the very large concentrations measured, it is hard to discern just how well the instrument/technique can observe cleaner, free tropospheric ammonia levels after seeing large plumes. For example, while the correlation is impressive in Fig. 10 and shows the value of this overall approach, with a several second response time for NH3, the "peaks" and "valleys" may still be attenuated to some extent. It would be helpful to show a plot against a true 1 Hz tracer correlation instead of two instruments with 2-3 second response times.

We agree with the commenters suggestion of trying to compare with a true 1 Hz tracer. We have identified acetonitrile, measured with the PTR-ToF-MS, as a better 1-Hz tracer that is well correlated with NH3 in smoke. Unfortunately, we do not have measurements from the PTR-ToF-MS during RF15 when the aircraft performed transects of the S. Sugarloaf fire and the NH3 instrument was systematically operated with and without passivant in flight through smoke. Instead, we use measurements of acetonitrile from the Bear Trap Fire (RF09) conducted on 09 August 2018 to perform a similar linear regression comparison of fine structure features as that described earlier in Sect. 5.2.2. Briefly, we conducted linear regression analysis of scatter plots of NH3 versus CH3CN incrementally averaged from 1 to 5 seconds until linear regression resulted in a maximum  $R^2$  value. We found the best fits resulted from regressions of measured NH3 with the 1-Hz reported and 2-second averaged CH3CN ( $R^2 > 0.97$  and within 0.001 of each other). A timeseries of  $NH_3$  and  $CH_3CN$  from an example plume transect of the Bear Trap Fire in RF09 is included here for discussion with the commenters and mentioned briefly in the text, but since we cannot produce the same figure for RF15 when the NH3 instrument was systematically tested with and without passivant, we feel that adding this as a figure to the paper does not add much to the discussion. We also note that there is little hysteresis in the recovery of background ratios of NH3 to CH3CN following the plume transect in RF09.

We assert that the NH3 observations are well correlated with the 1-Hz reported CH3CN data as well as the 2-second average of CH3CN in the discussion by adding the following text to the end of Sect. 5.2.2: "Since the time response of the CO measurement was limited by its sample flow rate and inlet configuration, we also compare NH3 to acetonitrile (CH3CN) measured by the PTR-ToF-MS. CH3CN is well correlated with NH3 in smoke, and may be more representative of a true 1-Hz tracer owing to operation of the instrument inlet at a flow rate of ~15 SLPM. However, there are no measurements from the PTR-ToF-MS during RF15, the research flight during which the NH3 instrument was systematically tested with and without passivant. Instead, we use measurements of CH3CN from the Bear Trap Fire (RF09) conducted on 09 August 2018 to perform a similar linear regression analysis of fine structure features of measured NH3 versus CH3CN, with CH3CN incrementally averaged up to 5 seconds. We find the best fits result from linear regressions of measured NH3 with the 1-Hz reported and 2-second averaged CH3CN ( $R^2$ is > 0.97 and within 0.001 of each other)."

There are also some differences between the start and end of the plumes in Fig. 10 in terms of the  $NH_3/CO$  ratio. As one progresses in the plume, the  $NH_3/CO$  ratio seems to get higher, which would be consistent if the background is growing. Differences in plume chemistry across the transect may be a reason for this, too. However, outside the plume (start/end of timeseries), the  $NH_3/CO$  level isn't the same, either.

We recognize that there could be differences in the  $NH_3/CO$  ratio between the start and end of the plumes in Fig. 10. Indeed, differences in background mixing ratios of  $NH_3$  compared to CO before and after the first transect of the S. Sugarloaf fire are apparent when the  $NH_3$  and CO signals in Fig. 10 are magnified by a factor of 50 and 10, respectively. To highlight this difference, we have amended the time series in Fig. 10 as follows:

---

## Author Response (AR2)

**Response to Interactive Comment RC1 by Anonymous Referee #2 on manuscript # amt-2019-11:**

We greatly thank Anonymous Reviewer #2 for their time and effort to provide detailed specific comments for this manuscript, which have greatly improved the accuracy and clarity of this work. We have responded to each of the specific comments below with the reviewer's comments shown in bold font and our responses immediately following them in plain font.

**Specific Comments from Anonymous Reviewer #2:**

**Page 1, Line 17 – Please give temperature range of the aircraft cabin instead of simply labeling it hot. This would give the reader context when considering using the instrument in other environments, such as, a trailer or tower.**

The following text has been added to the abstract for perspective about what a hot aircraft cabin could mean in terms of temperature…"(e.g., average aircraft cabin temperatures expected to exceed 30 ℃ during summer deployments)".

**Page 4, Line 26 – The inertial inlet description is sparse. The reader is not told what material it is constructed from until Page 7, Line 4 in the text or finds it buried in the caption of Figure 1. It is most logical for the reader to state that here in section 2.2.2 Inertial Inlet. In reference to figure 1, what is the size of the critical orifice? What temperature is it heated to? Also, it is a little misleading to say the 'The QC-TILDAS detector is typically operated with a heated inertial inlet . . .' since, from the literature and later in this manuscript, it seems other QC-TILDAS instruments measuring methane, carbon monoxide, ethane, for example, do not require or use an inertial inlet. Perhaps it should say 'The NH3 QC-TILDAS detector . . .'.**

We have amended Sect. 2.2.2 and figure 1 with details about the inertial inlet. This section now reads as: "The $NH_3$ QC-TILDAS is typically operated with a heated inertial inlet positioned upstream of the spectrometer to provide filter-less separation of particles >300 nm from the sample stream, as shown in Fig. 1a. Coupling an inertial inlet with a QC-TILDAS has been well established following several laboratory and ground-based field experiments (Ellis et al., 2010; Ferrara et al., 2012; Tevlin et al., 2017; von Bobrutzki et al., 2010; Zöll et al., 2016). The inertial inlet is described in detail by Ellis et al. (2010) and Roscioli et al. (2016). Briefly, the inertial inlet used in these experiments consists of a quartz tube (12.7 mm o.d., 10.4 mm i.d.) with an integral, conical-shaped critical orifice roughly 1 mm in diameter positioned at about half the length of the tube, as shown in Fig. 1a. After passing through the orifice, gas (and particulates) are accelerated to a higher speed at a lower pressure (between 40 and 100 Torr) through the latter half of the 12.7 mm quartz tube, and then pass into a second quartz tube (25.2 mm o.d., 22.2 mm i.d.) that is sleeved around the 12.7 mm tube. The sample flow is split into two branches with approximately 90% of the total flow through the critical orifice (denoted by the blue arrow in Fig. 1a) being forced to make an 180˚ turn around the edge of the 12.7 mm tubing to continue to the spectrometer, and the other 10% (denoted by the orange arrow in Fig. 1a) being dumped via the straight section of 25.2 mm tube into the main pumping system. The inertia of particles with aerodynamic diameters greater than ~300 nm is too large to follow the gas stream around the 180˚ turn, thereby forcing the particles into the 10% of the flow stream that is directed to the pumping system. Ellis et al. (2010) reported that the inertial inlet, which acts like a form of virtual impactor, removes more than 50% of particles larger than 300 nm. A tee positioned immediately upstream of the critical orifice allows for pressure measurements using a baraton transducer (range 0-1000 Torr), which is used in determining the sample flow rate, and an auxiliary draw that allows the dead volume around the base of the conical-shaped critical orifice to be actively flushed. The flow rate of the auxiliary draw ranges from 160 to 500 sccm with changes in ambient pressure at the inlet tip. The inertial inlet is housed in a fiberglass enclosure, with the inside of the enclosure maintained at 40°C."

**Page 6, Line 46 – While I applaud updating the cross-section used with optical absorption system described by Neuman et al., 2003, the number used here, 4.69e-18 cmˆ2, is slightly different than the 4.4e-18 cmˆ2 used in the 2003 manuscript not in contrast to. Furthermore, it appears that the +/- 0.03e-18 is the standard deviation of the average of the three cross sections listed here. What is the uncertainty of each and then the uncertainty for the average? Are these two cross sections used in interpreting the absorption cell results here and in the previous manuscript within each other's uncertainty?**

We agree with the reviewer that there is some confusion about how the uncertainty in the calibration source is determined in Sect. 2.3. The text has been updated to include the uncertainties of the individual reported values from Froyd and Lovejoy (2012) ($4.67 \pm 0.08 \times 10^{-18}$ cm$^2$), Chen et al. (1998) ($4.7 \pm 0.5 \times 10^{-18}$ cm$^2$) and Cheng et al. (2006) ($4.7 \pm 0.5 \times 10^{-18}$ cm$^2$). Also, following careful consideration of the reviewer's comment, we now utilize the weighted average and associated propagated uncertainty of $4.7 \pm 0.1 \times 10^{-18}$ cm$^2$ as a more appropriate treatment for combining the cross section values from the literature. The weighted mean absorption cross section utilized here is in agreement within the uncertainties with the value reported by Neuman et al. (2003) (e.g., $4.4 \pm 0.3 \times 10^{-18}$ cm$^2$). We have also modified the text in Sect. 2.3 to clarify that the $\pm 2\%$ uncertainty in the weighted mean of the absorption cross section has been factored into the total estimated uncertainty ($\pm 7\%$) associated with the NH$_3$ calibration source used in these experiments. Further, with only a $\pm 2\%$ uncertainty on the weighted average of the updated cross sections, the uncertainty in the absorption cross section is no longer the dominating factor in the total uncertainty of the calibration source using the NOAA UV optical absorption system. The $\pm 7\%$ uncertainty in the calibration source is factored into the overall instrument uncertainty (e.g., 200 pptv $\pm 12\%$), as described in Sect. 4.1. We have amended this portion of Sect 2.3 as follows: "In this work, we refine the uncertainty of the NOAA calibration of the emission rate of the permeation device used in these experiments by utilizing more recent assessments of the NH$_3$ absorption cross section reported in the literature. Here, we use a weighted average of the NH$_3$ absorption cross sections reported by Froyd and Lovejoy (2012) ($4.67 \pm 0.08 \times 10^{-18}$ cm$^2$), Chen et al. (1998) ($4.7 \pm 0.5 \times 10^{-18}$ cm$^2$) and Cheng et al. (2006) ($4.7 \pm 0.5 \times 10^{-18}$ cm$^2$). The weighted mean utilized here ($4.7 \pm 0.1 \times 10^{-18}$ cm$^2$) is in agreement within the uncertainties with the value reported by Neuman et al. (2003) (e.g., $4.4 \pm 0.3 \times 10^{-18}$ cm$^2$). Combining in quadrature the $\pm 2\%$ uncertainty associated with the weighted mean of the absorption cross section, the $\pm 2.5\%$ uncertainty in the stability of the permeation device between pre- and post-project calibrations with the NOAA UV optical absorption system, and a conservative estimate of $\pm 6\%$ for other sources of uncertainty associated with the NOAA calibration system, we determine a total estimated uncertainty of $\pm 7\%$ for the emission rate of the permeation device used in these experiments."

**Page 7, line 4-7 – Since the sample flow rate is critical in calculating the calibration mixing ratios, the orifice size should be given in the text either here or in section 2.2.2.**

The size of the critical orifice (~1 mm) has been added to Sect. 2.2.2.

**Page 8, Section 4.3 – Some of the fits shown in Figure 5, particularly in panel b, do not match the data when it begins to flatten out. Would a triple exponential fit work better? Also, it would be easier for the reader to judge the fit if the data is present as symbols and the fit as a solid line.**

In our initial analysis, we too had considered whether a triple exponential fit would work better. Indeed, triple exponential fits do generate a more reasonable fit of the time profiles shown in Fig. 5. However, we elected to report the results of bi-exponential fits in the original manuscript for the following reasons: 1) there is more physical basis for relating a bi-exponential fit to the experiments conducted in this work, 2) the results of a bi-exponential fit could be directly compared to similar results reported by Ellis et al. (2010) and Roscioli et al. (2016), and 3) the coefficient associated with the third time constant ($A_3$) was <5% on average of the sum of the coefficients (e.g., $[A_3/(A_1 + A_2 + A_3)]$) and <23% on average of the sum of the coefficients associated with the latter two time constants (e.g., $[A_3/(A_2 + A_3)]$). All the same, we agree that this discussion about the possibility of a triple exponential fit does have merit in this manuscript, and thus we have added the following discussion to the end of Sect. 4.3:

"From Fig. 5, it appears that a bi-exponential fit does not always do a good job of approximating the observations. Indeed, reduced chi-square values from bi-exponential fit of the decay profiles ranged from 0.4 to 1.3. A triple exponential decay with the functional form shown in Eq. (2):

$$y = y_0 + A_1 exp\left(\frac{-(t-t_0)}{\tau_1}\right) + A_2 exp\left(\frac{-(t-t_0)}{\tau_2}\right) + A_3 exp\left(\frac{-(t-t_0)}{\tau_3}\right) \qquad (2)$$

produces better fits to the time profiles shown in Fig. 5. Albeit, the coefficient associated with the third time constant ($A_3$) is small (e.g., $A_3$ is <5% on average of the sum of the coefficients (e.g., $[A_3/(A_1 + A_2 + A_3)]$)) and <23% on average of the sum of the coefficients associated with the latter two time constants (e.g., $[A_3/(A_2 + A_3)]$)). While the physical basis for using a triple exponential fit is not forthright, it is possible that there is more than one time constant associated with the gas exchange rate through the sample flow pathway, the interaction of $NH_3$ molecules with the sampling surfaces, or a combination of these effects."

Figure 5 has also been updated with symbols according to the reviewer's suggestions.

**Page 15, line 19 – I am concerned on the use of 'highlights the slightly faster time resolution of NH3 compared to CO.' without the qualifier 'for the instruments as configured here' to prevent misconstruing or over generalizing the observations in the future. It seems unlikely it would hold true if the instruments were configured with equal flow rates and tubing lengths.**

The phrase "for the instruments as configured here" has been added to the text on Page 15, Line 19.

**Figure 1 – Please indicate the i.d of the PFA tubing used, the size of the critical orifice in the inertial inlet, and the temperature the inertial inlet is held at. From Figure 1 the length of the strut appears to be 12 cm, not the 36 cm stated in the text. What is the function of the aux. draw, which is not mentioned in the text or caption?**

Figure 1a, its figure caption, and relevant parts of Sect. 2.2.3 have been updated with the recommended information. In terms of tubing lengths, we had intended to indicate that the full length of the sample flow path from the inlet tip to the inertial inlet is 107 cm. We agree that the current description of the inlet lengths is confusing and may not have been accurately labelled in the original version of the manuscript; therefore, we have modified Fig. 1a and Sect. 2.2.3 to clarify.

**Figure 2 – With the lower panel in units of ppb^2 on a logarithmic scale, it is very difficult to see where the 60 ppt precision estimate comes from. Perhaps consider a lower right axis in ppt? Also, I suggest using the same scale in the lower panel for both plots.**

Figure 2 has been updated according to the reviewer's suggestions.

**Response to Interactive Comment RC2 by Anonymous Referee #1 on Manuscript # amt-2019-11:**

We would like to thank Reviewer #1 for their time and effort to provide detailed comments that have greatly improved the clarity of this work. We have responded to each comment below with the reviewer's comments shown in bold font and our responses immediately following them in plain font.

**This is a clearly written manuscript that documents the performance of a closed-path absorption spectrometer for the measurement of NH₃ aboard an aircraft, with a particular emphasis on the utility of an active passivation technique. The manuscript is appropriate for AMT, and should be published after addressing the following issues:**

**General comments: In several places (e.g. P2, L29; P3, L32; P4, L27; P4, L33), the manuscript uses the term 'detector' to refer to the instrument itself, whereas in other places, including in Figure 1, 'detector' is used to refer to the MCT detector that collects the transmitted radiation, but is not in contact with the gas flow of the system. I found the more general use of the term somewhat distracting/confusing and would suggest using either 'QC-TILDAS', 'spectrometer', or 'instrument' in whichever case is appropriate.**

We have changed this term throughout the manuscript according to the reviewer's suggestions.

**In the section discussing the vibrational and structural issues, the authors mention (P5, L35) 'reinforcing' the 'strain relief'. It was not clear to me if this involved providing more slack in the sampling lines, or making them more rigid. A little more information would be helpful.**

We have amended the last sentence of Sect. 2.2.4 to read as: "However, this motion sensitivity could be minimized by keeping tubing lengths to a minimum and reinforcing the strain relief of the sample tubing connected to the QC-TILDAS enclosure inlet and outlet ports (e.g., rigidly securing all flexible tubing to the frame of equipment rack with cable ties) prior to installation on the aircraft."

**In Section 4.1, the authors describe a zero overflow experiment. Does the (> 500 sccm) refer to the difference between the flow of zero air being delivered and the flow pulled by the instrument? Clarification would be useful.**

Yes, we mean the 500 sccm flow to be the difference between the flow of zero air being supplied to the inlet and the flow pulled by the instrument. We have added the following sentence to clarify: "An overflow > 500 sccm (e.g., the difference between the flow of zero air being supplied to the inlet and the instrument's sample flow) was maintained to ensure that the sample stream was truly NH₃-free during this test.

**In Section 5.1, the authors explore the impact of inlet aging and the use of the passivant on the time response of the system. While the proportion of the time response governed by the slow, "adsorptive", term was typically quite low ($D < 10\%$), the magnitude of the step change in concentration was rather large (50 ppb), so caution should be taken in extrapolating that result to ambient observations.**

We have added the following caveat to Sect. 5.1: "While the proportion of the time response governed by the slow, "adsorptive", term was typically quite low ($D < 10\%$), the magnitude of the step change concentration utilized here is large (e.g., 50 ppb), so caution should be taken when extrapolating these results to ambient observations away from concentrated source regions."

**Section 5.2.1 presents an interesting case study in which two intercepts of an intense NH3 plume led to much different sampling efficiencies depending on whether or not the passivant was being added to the inlet, as the result of a recent pre-flight contamination. I found this section a bit confusing because the time period between 13:20 and 13:23, when both the QC-TILDAS and the PTR-TOF-MS measured enhanced (and consistent) NH₃ is not described. One infers that the passivant was being used at the time, however it's not clear.**

We have added the following clarification to this section in the paragraph where the PTR-$NH_3$ measurements are described: "Passivant was not added to the PTR-ToF-MS; active continuous passivation was only applied the QC-TILDAS-based instrument during the selected times described above. It is clear by visual comparison to the PTR-ToF-MS that the non-passivated, "contaminated" QC-TILDAS instrument did not capture all of the expected ambient $NH_3$. This is evident from the differences in measured $NH_3$ mixing ratios reported in Fig. 8 during the time period between 13:20 and 13:23 when the QC-TILDAS was operated without passivant. During this time period the PTR-ToF-MS consistently measured more $NH_3$ than the QC-TILDAS, with the enhancement measured by the PTR during the plume intersect at 13:30 MDT showing an expected mixing ratio of ~45 ppbv. According to PTR-$NH_3$, the integrated $NH_3$ signal during the plume intersect at 13:30 MDT was only 14% less than the integrated $NH_3$ signal measured during the plume intersect at 14:00 MDT, and thus a significant enhancement in $NH_3$ should have been observed by the QC-TILDAS-based instrument. However, the non-passivated, "contaminated" QC-TILDAS-based instrument measured only a fraction of the $NH_3$ expected during the plume transect at 13:30 MDT, with the only attributable difference being $NH_3$ molecules adsorbing to the sampling surfaces."

**Specific comments:**
**P2, L11 – $NH_3$ is regulated under the Gothenburg protocol in some parts of the world.**

We have added the following to the introduction: "While $NH_3$ is regulated under the Gothenburg protocol in some parts of the world (e.g., http://www.unece.org/environmental-policy/conventions/air/guidance-documents-and-other-methodological-materials/gothenburg-protocol.html), it remains an unregulated pollutant in the U.S. (Gilliland et al., 2008)."

**P3, L33 – The 'D' in QC-TILDAS has traditionally stood for 'differential', not 'direct'**

We thank the reviewer for pointing out that prior usages of the acronym QC-TILDAS have referred to the 'D' as 'differential'. Aerodyne Research Inc., the manufacturer of the mini-TILDAS $NH_3$ monitor used in these experiments, has recently changed the 'D' to stand for 'direct' since they feel it better reflects the measurement method. While there are prior publications that use 'differential', newer papers and manufacturer's spec/product sheets (e.g., http://www.aerodyne.com/sites/default/files/Product%20sheet%20NH3.pdf) are now using the word 'direct'.

**P7, L41 (and subsequently) 'Hydroscopic' should be 'hygroscopic'**

We have made this correction.

**Figure 8 caption – 'colorded' should be 'colored'**

We have made this correction.

**Response to the interactive short comment on manuscript # amt-2019-11:**

**The manuscript assesses the performance of a closed-path, airborne-based ammonia instrument (Aerodyne Res. Inc.) by demonstrating the performance of active passivation under flight conditions. Ammonia is incredibly challenging to measure anywhere (with any technique) due to the significant adsorption/desorption effects on instrument/inlet surfaces, particularly on an airborne-based platform where temperatures, humidities, and ambient ammonia concentrations can vary dramatically. The authors show greatly improved performance when using the passivated versus unpassivated flows for sampling large $NH_3$ concentrations (10s-100s ppbv $NH_3$) from farms and biomass burning plume. The documentation of the instrument performance versus flight maneuvers was particularly valuable. This manuscript represents a large advance in airborne-based ammonia measurements, and the authors' experiences on using passivant additions in addition to /in lieu of frequent cleaning are important for future implementation of closed-path ammonia instruments in specific but also ammonia sampling more generally (laboratory experiments, calibrations, etc.). However, there remain some areas that require greater clarification to put the research in the proper context.**

**1. The response times, and applicability, to smaller $NH_3$ variations should be discussed (and backgrounds relevant to very low free tropospheric values, <ppbv). The detection limit needs better justification.**

Thank you for these comments. See our responses below each specific comment/suggestion related to these topics.

**In this study, the step change of $NH_3$ was created by turning off the calibration gas. The change is around 85 – 115 ppbv. This variation is uncommon for sites away from source regions. At high $NH_3$ concentrations and large variations, $NH_3$ observations may be less impacted by surface interaction because a "clean" sampling line only has a finite number of adsorption sites which could be quickly fill up under this condition. This effect has been reported by Ellis et al.(2010), and it may explain why passivation did not help to increase the response time of the instrument. At low $NH_3$ concentrations, a greater fraction of $NH_3$ molecules may interact with the inner surface.**

We agree that a limitation of these experiments is a lack of tests with lower mixing ratios of $NH_3$, which could affect the applicability of these results to some specific applications, namely the remote free troposphere. As the commenters point out, the effect of decreasing instrument response time with increasing step change concentration has already been extensively characterized by Ellis et al. Therefore, the main goal of this paper is to further the works of Ellis et al. (2010) and Roscioli et al. (2016) by characterizing the effects of passivant addition on instrument time response in flight. For this discussion we also note that the results of Ellis et al. were reproducible in our own experiments when time profiles were generated with various levels of step change concentrations. One of which included a step change from ambient levels of $NH_3$ (e.g., between 5 and 12 ppbv) to zero, which was generated by switching on the overflow injection of $NH_3$-free air at the inlet tip after a period of sampling near homogeneous levels of ambient $NH_3$ for several minutes. All the same, we agree with the commenters that the text should be amended to clarify that these results may be most applicable to near source sampling because the instrument utilized for these experiments was optimized for sampling large and rapid gradients of $NH_3$ in smoke.

In response to this and the next comment, we have added the following text to Section 4.4:
"All the same, further measurements are recommended for assessing sampling biases that could arise during field measurements of low mixing ratios of $NH_3$ in clean environments following long periods of exposure to near source level concentrations. The potential for an adsorption-related "memory effect" of $NH_3$ (e.g., Williams et al., 1992) on the sampling surfaces following long-term exposure to high concentrations of $NH_3$ is discussed in following sections."

And we added this text to Section 5.1.1: "In this study, the instrument response is rigorously tested with a single step change of $NH_3$ created by turning off a 50 ppbv calibration gas mixture. We note that such large variations in $NH_3$ mixing ratio may

not been full applicable to field measurements in unpolluted regions away from concentrated sources of $NH_3$. As described by Ellis et al. (2010), large gradients in $NH_3$ may be less impacted by surface interactions because "clean" sampling surfaces only have a finite number of adsorption sites that could be quickly filled under high $NH_3$ conditions. At lower $NH_3$ concentrations, a greater fraction of $NH_3$ molecules may interact with the inner surfaces. This could explain why passivation 5    did not help to increase the response time of the instrument."

**Roscioli et al. showed that $t_{90}$ of a similar instrument was 12 sec for a step change of 3 ppbv (from 0 to 3 ppbv) without passivation (2). When 4 ppmv passivant was applied, $t_{90}$ decreased to 2 sec for the same step change. The instrument can be considered clean since it was flushed with $NH_3$-free and low $NH_3$ gases. Therefore, even a "clean"** 10    **closed-path instrument may not be capable for high-frequency (>1 Hz) field application with small $NH_3$ variations without passivation, and passivant additions may still not work for fast operation (> 1 Hz) under clean conditions. The authors should discuss in more details about the applicability and effectiveness of passivation to field applications with relatively low $NH_3$ concentrations (e.g. flux measurements in rural area and airborne observations away from sources).**

We greatly appreciate your comments with respect to the comparison of time responses that we collected with those reported by Roscioli et al. (2016). While we frame the results of the time response tests so that they can be directly compared to the works of Ellis et al. (2010) and Roscioli et al. (2016), we believe that it is difficult to compare the level of cleanliness of two different instrument systems. In this particular case, it may not be fair to say that our "clean" (aka. pristine, out-of-the-box,
20   never-used-in-the-field) instrument is the same level of cleanliness as the copiously flushed, yet previously deployed instrument used by Roscioli et al. We are fortunate to know the history of both instruments, and surmise that the QC-TILDAS utilized by Roscioli et al. (2016) more closely resembles an instrument with a mid-level of cleanliness (similar to what we define as a "typical" operating condition) rather than the pristine, out-of-the-box condition that we referred to as "clean" in this work. This is because the instrument utilized by Roscioli et al. was a dual channel QC-TILDAS optimized to
25   measure $NH_3$ as well as $HNO_3$ (aka. two well-known sticky molecules). Prior to lab tests, the instrument was deployed aboard the NSF/NCAR C-130 research aircraft in the 2014 FRAPPE field campaign where it had been exposed to near source levels of $NH_3$ and urban emissions in the Colorado Front Range. Further, during the lab tests, the dual channel instrument was used to test several strong bases as passivant agents for $NH_3$ and several strong acids as passivant agents for $HNO_3$. In our experience with contamination and cleaning of sampling surfaces, we found that the instrument flow path
30   could only be truly cleaned by replacing tubing and inlet components where possible and by performing several cycles of ethanol/water rinse followed by week-long periods of flushing the sample flow path with $NH_3$-free air. The dual channel instrument was solely flushed with $NH_3$-free air prior to lab experiments, but sampling surfaces were not systematically cleaned and replaced. For these reasons, we categorize the instrument used by Roscioli et al. as an instrument operating under a middle level or cleanliness, like our "typical" operating condition. We also point out that Roscioli et al. (2016) found
35   that the $NH_3$ time response of the instrument continually improved with increasing passivation concentration, presumably with an eventual lower limit somewhere at or above the volumetric flush time. Therefore, the degree to which one wants to achieve >1 Hz sensitivity can be determined by the amount of passivant added to the sampling system. In the case of the flights discussed here, we made a compromise between passivant use and time response in order to achieve a reasonable temporal response while not using an excessive amount of passivant. All the same, we agree with the commenters that
40   further discussion is warranted about the applicability and effectiveness of active continuous passivation to field applications with relatively low $NH_3$ concentrations, such as flux measurements in rural areas and measurements away from sources. We now include this statement in Sect. 4 and reiterate this point in Sect. 5. See the text pasted in response to comment above.

**Fig. 10: Because of the nature of the very large concentrations measured, it is hard to discern just how well the**
45   **instrument/technique can observe cleaner, free tropospheric ammonia levels after seeing large plumes. For example, while the correlation is impressive in Fig. 10 and shows the value of this overall approach, with a several second response time for $NH_3$, the "peaks" and "valleys" may still be attenuated to some extent. It would be helpful to show a plot against a true 1 Hz tracer correlation instead of two instruments with 2-3 second response times.**

We agree with the commenters suggestion of trying to compare with a true 1 Hz tracer. We have identified acetonitrile, measured with the PTR-ToF-MS, as a better 1-Hz tracer that is well correlated with $NH_3$ in smoke. Unfortunately, we do not have measurements from the PTR-ToF-MS during RF15 when the aircraft performed transects of the S. Sugarloaf fire and the $NH_3$ instrument was systematically operated with and without passivant in flight through smoke. Instead, we use

5  measurements of acetonitrile from the Bear Trap Fire (RF09) conducted on 09 August 2018 to perform a similar linear regression comparison of fine structure features as that described earlier in Sect. 5.2.2. Briefly, we conducted linear regression analysis of scatter plots of $NH_3$ versus $CH_3CN$ incrementally averaged from 1 to 5 seconds until linear regression resulted in a maximum $R^2$ value. We found the best fits resulted from regressions of measured $NH_3$ with the 1-Hz reported and 2-second averaged $CH_3CN$ ($R^2 > 0.97$ and within 0.001 of each other). A timeseries of $NH_3$ and $CH_3CN$ from an

10  example plume transect of the Bear Trap Fire in RF09 is included here for discussion with the commenters and mentioned briefly in the text, but since we cannot produce the same figure for RF15 when the $NH_3$ instrument was systematically tested with and without passivant, we feel that adding this as a figure to the paper does not add much to the discussion. We also note that there is little hysteresis in the recovery of background ratios of $NH_3$ to $CH_3CN$ following the plume transect in RF09.

[Figure]

15  We assert that the $NH_3$ observations are well correlated with the 1-Hz reported $CH_3CN$ data as well as the 2-second average of $CH_3CN$ in the discussion by adding the following text to the end of Sect. 5.2.2: "Since the time response of the CO measurement was limited by its sample flow rate and inlet configuration, we also compare $NH_3$ to acetonitrile ($CH_3CN$) measured by the PTR-ToF-MS. $CH_3CN$ is well correlated with $NH_3$ in smoke, and may be more representative of a true 1-

20  Hz tracer owing to operation of the instrument inlet at a flow rate of ~15 SLPM. However, there are no measurements from the PTR-ToF-MS during RF15, the research flight during which the $NH_3$ instrument was systematically tested with and without passivant. Instead, we use measurements of $CH_3CN$ from the Bear Trap Fire (RF09) conducted on 09 August 2018 to perform a similar linear regression analysis of fine structure features of measured $NH_3$ versus $CH_3CN$, with $CH_3CN$ incrementally averaged up to 5 seconds. We find the best fits result from linear regressions of measured $NH_3$ with the 1-Hz

25  reported and 2-second averaged $CH_3CN$ ($R^2$ is > 0.97 and within 0.001 of each other)."

**There are also some differences between the start and end of the plumes in Fig. 10 in terms of the $NH_3$/CO ratio. As one progresses in the plume, the $NH_3$/CO ratio seems to get higher, which would be consistent if the background is growing. Differences in plume chemistry across the transect may be a reason for this, too. However, outside the**

30  **plume (start/end of timeseries), the $NH_3$/CO level isn't the same, either.**

We recognize that there could be differences in the NH₃/CO ratio between the start and end of the plumes in Fig. 10. Indeed, differences in background mixing ratios of NH₃ compared to CO before and after the first transect of the S. Sugarloaf fire are apparent when the NH₃ and CO signals in Fig. 10 are magnified by a factor of 50 and 10, respectively. To highlight this difference, we have amended the time series in Fig. 10 as follows:

[Figure]

Updated caption for Fig. 10:
Time series of 1-Hz NH₃ (black lines) and CO (red lines) measured during a crosswind transect of the smoke plume from the South Sugarloaf Fire (RF15) on 26 August 2018. The transects represent nearly identical passes through the smoke plume with the only perturbation of the NH₃ instrument being operated (a) with passivant and (b) without passivant. Changes in fine structure features of NH₃ have the strongest $R^2$ correlation with CO when the NH₃ measurements are averaged to 3 s. A x50 magnified view of 1-Hz NH₃ (blue lines) and x10 magnified view of CO (orange lines) shows differences in background levels of NH₃ compared to CO before and after each plume transect.

[As context for this discussion, we note that the instrument inlet was overblown with NH₃-free air for the duration of a 2-hour pre-flight exercise prior to take off. Following take-off, the instrument sampled a maximum of 5 ppbv NH₃ during ascent out of Boise and was then exposed to < 1 ppbv NH₃ for 1-hour during transit to the S. Sugarloaf fire.]
As the commenters suggest, there could be several causes for the differences in NH₃/CO ratio observed before and after the plume transect in Fig. 10. One reason could be physical differences in plume chemistry, mixing, or background composition. Another could be "memory effects" in the sample plumbing due to retention of NH₃ adsorbed to sampling surfaces following exposure to NH₃ mixing ratios in excess of 400 ppbv (Williams, et al., 1992). The observation could also reflect some combination of both. While distinct differences in background are apparent in Fig. 10, we note that differences in

background before and after the plume were not always observed during WE-CAN research flights (e.g., RF07 conducted on 06 August 2018 described in the next section of this response). Since the root of the differences are not immediately obvious and because differences seem to vary among the WE-CAN research flights, we now also include a response time for the signal recovery shown in Fig. 10 assuming the worst-case scenario that the differences in background are solely attributed to memory effects on the sampling surfaces. In this worst case, the time for the $NH_3$ measurement following the plume transect to recover to near background mixing ratio levels observed prior to the plume transect (e.g., 1 ppbv) is roughly 250 s. This time frame most closely resembles $t_{99,obs}$ for the "typical" operating condition when operated with or without passivant. To highlight the commenter's points, we have amended Sect. 5.2.2 with the following text: "Differences in background mixing ratios of $NH_3$ and CO measured before and after the first transect of the smoke plume from the S. Sugarloaf fire are apparent in the magnified timeseries for each in Fig. 10. The differences in $NH_3$/CO ratio observed at 20:14 UTC and 20:25 UTC following in-smoke measurements of $NH_3$ that exceeded 400 ppbv could have resulted from physical differences in plume chemistry, mixing or background composition on either side of the plume, adsorption-related memory effects in the sample plumbing due to retention of $NH_3$ molecules adsorbed to the sampling surfaces (Williams et al., 1992), or a combination of both. Since the root of the differences are difficult to distinguish and may vary among the WE-CAN research flights, we utilized these differences to characterize the instrument time response given the worst-case scenario that the differences in background observed in Fig. 10 are solely attributed to memory effects on the sampling surfaces. In this worst case, the response time for the $NH_3$ measurement following the plume transect to recover to near background mixing ratio levels observed prior to the plume transect (e.g., 1 ppbv) is roughly 250 s. The time frame most closely resembles $t_{99,obs}$ for the "typical" condition when the instrument is operated with or without passivant. This recovery time and "typical" cleanliness condition are within our expectations for the instrument during this research flight (RF15) since the instrument had routinely been used to sample near source concentrations of $NH_3$ in smoke during several prior consecutive research flights without refreshing the sampling surfaces between flights."

**Related to this overall point, on the large y-scale axis (500 ppbv $NH_3$) in Fig. 10, while the concentration looks to be "close" to zero, in reality on this scale it could be numerous ppbv $NH_3$. Even if the instrument response time is nominally on the order of a few seconds, going from 400 ppbv $NH_3$ to sub ppbv $NH_3$ could take a long time and may result in biases in clean conditions in the free troposphere. It may be helpful to show a vertical profile of $NH_3$ in the ascent out of Greeley (where high agricultural emission concentrations exist) and compare it to another, short-lived boundary layer tracer that would be highly enhanced in the boundary layer vs. the free troposphere. On the left below, I show a vertical profile of $NH_3$ from the NASA DISCOVER-AQ California in the San Joaquin Valley (taken from Sun et al., 2015), where the importance of the authors' large improvement is clearly validated (needed), but where concerns of going very high $NH_3$ to nominally sub-ppbv $NH_3$ at high altitude could still be an issue even with this approach. Also attached on the right is a profile of $NO_2$, ethane, and CN in the Greeley area from DISCOVER-AQ Colorado. I'm not sure if these are necessarily the best tracers per se from a quick look, but one can see very sharp gradients at the top of the boundary layer, and it would be illustrative to see how the shape of the $NH_3$ profile compares to other ~ short-lived tracers when ascending/descending across the mixed layer height. WE-CAN should have plenty of such measurements on the C-130 to compare.**

[Figure]

(Sun et al., 2015)

We agree that comparing vertical profiles from an ascent and descent is a great suggestion. While we do not have extensive enough vertical information from the test flights near Greeley, we do have a spiral over the California Central Valley about 60 miles southeast of Sacramento during WE-CAN research flight RF07 conducted on 06 Aug 2018. The spiral aimed to sample aged smoke in the Central Valley, and thus consisted of a descent followed by a spiraling ascent spanning between 4.5 km and 1.2 km AGL. As such, the observations (shown below) likely reflect a combination of aged smoke and agricultural emissions. Changes in $NH_3$ are consistent with changes observed for other tracers. While it may appear that there is some hysteresis in $NH_3$ compared to CO and $CH_4$ around 1.5 km as the aircraft ascends through the mixed layer, a closer look (e.g., $NH_3$ magnified x10 in the figure below) shows that $NH_3$ mixing ratios immediately drop to $\leq 200$ pptv. To put these observations into context of the "memory effect" discussion above, it should be noted that the maximum $NH_3$ mixing ratio prior to ascent during the spiral in RF07 was < 15 ppbv compared to the background measured in RF15 following a smoke plume transect where $NH_3$ was > 400 ppbv. A systematic analysis of the WE-CAN research flights for physical differences in plume chemistry, mixing, background composition, and hysteresis with plume concentration are beyond the scope of this work, but several of these topics are forthcoming in WE-CAN publications. Therefore, we include the discussion of a worst-case scenario of hysteresis in the manuscript, and only provide the following plots of vertical profiles for discussion with the commenters.

[Figure]

**Figure 3:** It is hard to see on Fig. 3a, but the constant altitude segments seem to show quite a bit of variability in the background, say, from -0.2 to +0.2 ppbv $NH_3$ within an altitude level for a given 1 Hz measurement. This calls to question as well the accuracy of anything < 0.5 ppbv, given that the background is changing by 0.4 ppbv. What were the ascent rates/g's after each constant altitude leg? The 50 pptv $NH_3$ sensitivity to typical flight maneuvers mentioned in the text doesn't seem consistent with Fig. 3a. If most of the variability is due to the ascent portions (g-forces), then perhaps a 1 Hz timeseries of the constant leg would be helpful. Also, how "polluted" was $NH_3$ prior to overfilling the inlet for a zero for these flights upon takeoff in Broomfield? How often was it zeroed vs. sampling? 10% duty cycle? Entire flight? 50%? The wording wasn't clear in the text for this portion of the flight.

The symbols and error bars in Fig. 3 represent the mean and 3s standard deviation of the mean measured zero level during constant altitude legs. Here, we purposefully depict the 3s standard deviation to illustrate the range of variability with respect to 3 times the Allan deviation, which we defined as the instrument's limit of detection. However, it is true that the zero signal level in Fig. 3 spans ±200 pptv around zero, or 400 pptv total. While we continue to report 3 times the precision as determined from the Allan variance since this is how the detection limit for similar instruments is reported in the literature, we have added the following to Sect. 4.1: "We note that the true detection limit of the instrument in flight may be better represented by the full range of variability about the mean zero signal level from the observations in Fig. 3 (e.g., an instrument detection limit of 400 pptv)." We have also added this information to the abstract.

Ascent profiles were typically performed at ~1000 ft/min. The 50 pptv sensitivity reported in Fig 4 and Sect. 4.2 is specific to turbulence and turns. To further clarify the variations in $NH_3$ zero signal level with altitude in Fig. 3, accelerations at the onset of an ascent at 1000 ft/min were measured to be 0.4 g for the up-down motion, 0.1 g for the side-side motion, and 0.07 g in the fore-aft motion. Given the accelerations during ascent and the slopes of the measured motion sensitivities (in units of ppbv/g) determined from Fig. 4., it is reasonable to expect as much as 400 ppt of variability from motion sensitivity. Sect. 4.1 and 4.2 have been updated with the following text to clarify the observations during ascent between constant altitude legs in Fig. 3: "It should also be noted that large accelerations in the up-down and fore-aft dimensions are also significant at the onset of vertical ascent. Accelerations measured in the up-down and fore-aft motions at the onset of a 1000 ft/min vertical ascent were measured to be 0.4 g and 0.08 g, respectively. Given the slopes above, these accelerations correspond to a maximum change in $NH_3$ zero signal level of 400 pptv during ascent, which is consistent with the variability in zero signal level observed in Fig. 3 when ascending between constant altitude legs."

We have also added a time series of the vertical ascent profile while overblowing the inlet tip with $NH_3$-free air to the bottom of Fig. 3, as shown here.

[Figure]

Updated caption for Fig. 3:
In-flight variations in zero signal level (in units of ppbv of $NH_3$) with respect to changes in (a) altitude, (b) cabin pressure, and (c) cabin temperature. A time series (d) illustrates the effects of motion sensitivity on the zero signal level as the aircraft

initiates an ascent and then levels off at a constant altitude. Gray symbols and lines represent the 1 s average of all of the 10-Hz data points collected in flight while overblowing the inlet tip with $NH_3$-free air; the red line in the time series is altitude AMSL. Colored symbols and error bars in the vertical profiles represent the average $NH_3$ zero signal and 3s standard deviation for each constant altitude level, 5 Torr increments in cabin pressure, and 2˚C increments in cabin temperature. Variations are largely within ±200 pptv (denoted by the light gray shaded areas).

For further context, we have also added the following detail to Sect. 4.2: "For these experiments, the instrument inlet was continuously overflowed with $NH_3$-free air for the duration of a 3-hour pre-flight exercise prior to take off. Overflowing the inlet was purposefully done to keep the instrument system free of contaminants (e.g., exhaust from other aircraft and ground-based support equipment) prior to sampling in flight."

We added similar info to Sect. 5.2.2: "During WE-CAN, the $NH_3$ instrument was typically zeroed between crosswind transects of a wildfire smoke plume when in background air and either just prior to or during turns. The instrument was zeroed every 10-20 mins during transits from Boise to the wildfires sampled with the frequency of zeros depending on the transit time. Zeros measured during WE-CAN research flights were typically collected for a period of 1 to 2 minutes, a duration much greater than the instrument response time, to ensure that zeros were measured well within 90% of the final zero signal level. Prior to each research flight, the $NH_3$ instrument was overflowed with $NH_3$-free air for the duration of a 2-hour pre-flight exercise."

**Table 2: I really appreciated the mass balance in Table 2/discussion (neat experiment!), though even here differences of ~ 10% of counting molecules still could mean significant backgrounds still exist relative to very clean conditions (though I recognize this mass balance counting is within the instrument uncertainty).**

**In summary of all of the above, taking 3 times the 1 Hz precision doesn't seem justified for the detection limit, nor an assessment of instrument accuracy at low concentrations. It seems the instrument is well designed for fires/agriculture but future work is still needed for clean conditions after such large plumes (or more justification in the manuscript). This is particularly true when going from dirty to clean conditions, given the many sampling biases that still may exist for ammonia.**

We have updated the discussion about detection limit given the 400 pptv variability in Fig. 3. We have also added notes about sampling biases in accord with the responses above.

**2. Validity of using bi-exponential decay model and meaning of the parameters should be addressed:**

**The bi-exponential decay model is essential to the discussion about instrument response time. The authors used the bi-exponential decay model to determine the response times of the instrument to associate gas exchange and the interaction of $NH_3$ molecules with sampling surfaces. The fit results were also used to extrapolate the 90% and 99% signal recovery times ($t_{90}$ and $t_{99}$). Therefore, it is necessary to address the validity of the bi-exponential decay model.**

**The bi-exponential decay model was first introduced to characterize response time of QC-TILDAS to $NH_3$ changes by Ellis et al.(1). However, the validity of the model was not discussed in the original work. Here, we propose to use the following a simplified surface-air exchange model to derive the biexponential decay model and discuss its validity.**

**After a step change, changes of the mixing ratio of $NH_3$ inside the instrument $\chi$ is caused by 1) the difference of $NH_3$ mixing ratio between the gas currently inside the chamber and the newly introduced gas ($\chi_0$); 2) adsorption or desorption to the inner surface of the instrument. These processes can be expressed as**

$$\frac{d\chi}{dt} = \frac{Q}{V}(\chi_0 - \chi) + \kappa(\chi_s(t) - \chi)$$

where Q is the flow rate and V is the inner volume of the instrument; $\kappa$ is the conductance of $NH_3$ between surface and air interface; $\chi_s$ is the compensation point of the inner surface (adsorption occurs when $\chi > \chi_s$, desorption occurs when $\chi < \chi_s$). The compensation point is a function of time and its variation depends on historical changes of $NH_3$ concentration inside the instrument. When there are no phase changes and chemical reactions, and the surface is not saturated by $NH_3$ or exhausted of $NH_3$ during the process, $\chi_s$ could be simplified as

$$\frac{d\chi_s}{dt} = \kappa(\chi - \chi_s).$$

When the surface is clean such that $\chi_s \ll \chi$, $\chi_s$ equation can be approximated as

$$\frac{d\chi_s}{dt} = \kappa\chi.$$

For the step change described in this study, $\chi_0 = 0$. Combine all the equations, we have

$$\frac{d^2\chi}{dt^2} + \left(\frac{Q}{V} + \kappa\right)\frac{d\chi}{dt} = \kappa^2\chi.$$

The general solution to the differential equation is

$$\chi = A_1 \exp\left[-\frac{1}{2}\left(\sqrt{\left(\frac{Q}{V}\right)^2 + 2\left(\frac{Q}{V}\right)\kappa + 5\kappa^2} + \left(\frac{Q}{V}\right) + \kappa\right)t\right]$$
$$+ A_2 \exp\left[\frac{1}{2}\left(\sqrt{\left(\frac{Q}{V}\right)^2 + 2\left(\frac{Q}{V}\right)\kappa + 5\kappa^2} - \left(\frac{Q}{V}\right) - \kappa\right)t\right]$$
$$= A_1 \exp\left(-\frac{t}{\tau_1}\right) + A_2 \exp\left(-\frac{t}{\tau_2}\right)$$

It can be seen from above derivation that the bi-exponential decay model approximates the universal solution of the differential equations, but it only works under certain conditions - the most important one is the relative cleanliness of the surface. After certain time, $\chi$ will approach $\chi_s$ and the solution to the differential equations becomes significantly more complicated and is unlikely to follow biexponential decay model. The authors should clarify the applicability of the bi-exponential decay model.

Given the validity of the bi-exponential decay model, it may be more reliable to derive $t_{90}$ and $t_{99}$ using observed time series directly if the measurements are not noisy.

$\tau_1$ and $\tau_2$ represents the combined effects of both gas exchange and air-surface exchange instead of representing the effects separately. Therefore, the statements about $\tau_1$ and $\tau_2$ from line 22 to line 24 on page 10 should be removed.

We greatly appreciate the time and effort by the commenters to provide us with a detailed surface-air exchange model for the bi-exponential decay. We would be happy to include your model in this paper or future papers if a peer-reviewed reference can be provided. However without that, we feel that derivations of this model are beyond the scope of this paper, and it seems more appropriate for the commenters to develop this model as the originators of these concepts. All the same, we agree that the bi-exponential fits do not always provide a perfect representation of the observations, which could indicate

instances where the model fails to adequately describe the physical system. As suggested by the commenters, we now also include values for $t_{90}$ and $t_{99}$ in Table 1 that are directly determined from the observations in Fig. 5. The new parameters are denoted as $t_{90,obs}$ and $t_{99,obs}$. However, we also continue to frame the results using the existing bi-exponential decay model in the literature for the following reasons: 1) as a way to provide context for fitting the time response profiles, 2) for consistency with the approach utilized in the peer-reviewed literature for similar instruments (Zahniser et al., 1995; Ellis et al., 2010; Roscioli et al., 2016), and 3) for ease of comparison to prior assessments with similar instrumentation by Ellis et al. (2010) and Roscioli et al. (2016).

**3. Uncertainty of response time may not be representative:**

**The uncertainty of the response time is currently estimated using error propagation of the fitted results. However, given the exponential natural of the issue, parameters like $t_{90}$ and $t_{99}$ may have a skewed distribution (i.e. log-normal distribution) with a long tail. This behavior may not be correctly captured by error propagation method. If fitted results are used, Monte Carlo method should be used. If the real time series is used, $t_{90}$ could be estimated as the standard deviation of time stamps of observations with NH3 between the $90 - \sigma_{obs}$ percentile and the $90 + \sigma_{obs}$ percentile.**

For comparison to the uncertainties derived from error propagation of the bi-exponential fit coefficients, we now also include an observation-based determination of the uncertainties for $t_{90,obs}$ and $t_{99,obs}$ in Sect. 4.3. These uncertainties reflect the $\Delta t$ spread in times associated with the 90±1% and 99±1% signal recovery levels, where ±1% on the signal recovery level corresponds to ±0.5 ppbv for a 50 ppbv step change, which is within the instrument's limit of detection.
* * *
**Summary: The authors have shown a marked improvement in the use of airborne-based ammonia measurements. In fact, these measurements are the most impressive and reliable to date in the literature and have set a new standard for all future campaigns (airborne and ground-based). The technique has applicability to Picarro and other closed-path sensors, as well as calibration methods for open-path sensors (which, indeed, have to be "closed" typically for calibration). However, there are still many gremlins for airborne ammonia, particularly with its enormous dynamic range in concentration and adsorption issues, which get magnified for sub-ppbv NH3 levels that are expected in the free (or at least upper) troposphere (Asian UT monsoon levels excepted, possibly). I hope the points above allow for some clarifications that strengthen the manuscript.**

****others****
**Page 11, line 4-10: The manuscript never mentions how the boundary layer height was determined. Was it known accurately in each case or simply assumed to be <1 km?**

A well-mixed layer below roughly 1 km was initially assumed. There could be differences in the structure of the boundary layer for the different test flights, which could be due to the colder/wetter ambient conditions during the test flights in September 2017 compared to the warmer/drier conditions during test flights in July 2018. Unfortunately, we have very few parameters to compare from the test flights as the instrument payload was minimal in 2017 and not all instruments were fully operational/optimized at the time of the WE-CAN test flights in 2018. On the other hand, we reliably have potential temperature, which was collected during each flight as part of the aircraft's standard suite of measurements. Vertical profiles of potential temperature do indicate a planetary boundary layer height was primarily between 1 and 1.5 km for both the 2017 and 2018 test flights. Although, there could have been more than one mixed layer during the 2018 test flights. We have added the following text to Sect. 4.4: "All the same, further measurements are recommended for assessing sampling biases that could arise during field measurements of low mixing ratios of $NH_3$ in clean environments following long periods of exposure to near source level concentrations. The potential for a "memory effect" of $NH_3$ on the sampling surfaces following long-term exposure to high concentrations of $NH_3$ is discussed in following sections." We have also modified Fig. 6 to include potential temperature and a rough guideline for the boundary layer height.

[Figure]

Updated caption for Fig. 6:

Vertical profiles of $NH_3$ (in ppbv) and potential temperature (in K) from (a) the first and third test flight in 2017 and (b) the test flights in 2018 when the instrument was operated without passivant. $NH_3$ mixing ratios as high as 80 ppbv were observed in the mixed boundary layer during missed approaches at Greeley-Weld County Airport and over northeastern Colorado compared to average mixing ratios of ~0.8 ppbv near Akron, Colorado following several days of rain. (c) Histograms of the corresponding $NH_3$ measurements collected above 1.5 km AGL (dashed line) show that measurements were frequently larger than 200 ppt, especially measurements that were collected in the free troposphere.

10 **Page 12, line 40-45: The authors should be aware that the relationship between water content and $NH_3$ adsorption is not necessarily linear. The interaction mechanism varies significantly depending on the amount of water present. A previous study by Vaittinen et al., 2018, has demonstrated this. Therefore, the two scenarios (dry vs 80% humidified) tested here may not be representative enough to tell the whole story.**

15 We appreciate the commenter's points. The Vaittinen et al. (2018) and Pogany et al. (2016) references have been added to Sect. 5.1.3 and we have amended the section with the following text: "We only measured two extreme relative humidity conditions for these tests, even though the relationship of surface interactions may be non-linear and vary greatly depending on the fraction of water vapor added as suggested by Pogany et al. (2016) and Vaittinen et al. (2018)."

20 And, "...a caveat of these tests is that the humidity levels tested here may not provide enough information to fully characterize the effects of passivant addition over the full range of dry to humid sampling conditions. Further characterization of the humidity dependence with and without passivant addition is recommended prior to future deployments of this instrument system (or similar QC-TILDAS instruments) in humid field environments."

25 **Page 13, line 40-43: It is not clear what criteria the authors used to determine that the $NH_3$ transmission shows little difference between the non-passivant and with-passivant transects. (By the response time/maximum reading/amount of the $NH_3$ measured?)**

The comparison is based on the amount of $NH_3$ measured by the detector. We have amended the text on Pg. 13 to clarify.

**Figure 6a: It would be helpful to show the exact boundary layer height for this profile for better clarity.**

We have amended Fig. 6 to include vertical profiles of potential temperature, which was measured as part of the aircraft's standard suite of parameters during the test flights. We agree that this is a helpful addition because of the subtle differences in mixed layers between the 2017 and 2018 test flight cases described above.

**Figure 8 (upper): I am confused by the brown circles/dots labeled as beef and dairy. Do the small dots indicate smaller facilities as compared to the large circles? The two circles on the upper and lower right presumably refer to dairy but there is a dot in the center of each circle. Does this mean the facilities have both dairy and beef?**

We greatly appreciate the commenters finding these typos. The smaller dots are meant to indicate smaller facilities in terms of head of cattle, and some of the beef and dairy animal operations are collocated. All the same, the original plot did have some defects in symbol outlines and layering that complicated its appearance. The plot, legend, and caption have been updated to clarify these differences.

**The total power output of the system should be described, since power seemed to be an issue even on an aircraft.**

We have added a Sect. 2.4 to describe the power, weight and space utilized by the instrument. The following text has been added in this new section. "The instrument system described above in the configuration that it was utilized aboard the C-130 aircraft requires the space of an entire NSF/NCAR G-V aircraft equipment rack (approximate dimensions 21.5" W x 28" D x 50" H). The equipment without the rack weighed approximately 150 kg and included a 30 kg uninterruptable power supply (UPS) and a 10 kg display laptop. The total power used by the instrument system was 1600 watts, with roughly one third of this total (600 watts) being dedicated to the main pumping system (Agilent, model Triscroll 600, 100 lbs installed). It is possible that the power, weight and space required for this instrument system can be reduced for future deployments by eliminating the UPS and display laptop. It may also be possible to reduce the size of the pump if different field applications allow for a lower sample flow rate to be used."

**Abstract: "Flight-ready" in the abstract seems redundant for the topic; also "custom" is mentioned three times in the first sentence.**

Agreed. We have implemented these changes.

**For the Allan plot, what offset was applied and how much? As written, it is confusing.**

We agree that the application of the offset is misleading as currently written. Owing to the vibration applied to the laser objective, the noise guidelines were offset by -150 ppt to align with the observations. The observations were not adjusted. We have updated Sect. 4.1 and the figure caption to clarify.

**SilcoNert 2000 has been shown to work very well for ammonia and water vapor adsorption – can the authors – Pogany et al., Meas. Sci. Tech., https://iopscience.iop.org/article/10.1088/0957-0233/27/11/115012/meta.**

We have added (Pogany et al., 2016) and (Vaittinen et al., 2018) to Sect. 5.1.3 to aid in discussion of the limitations of the humidity tests performed with and without passivant addition in this work. We have also added mention of other materials used in the past as potential passivant coatings to Sect. 2.2.5 via addition of the (Pogany et al., 2016) and (Yokelson et al., 2003) references. This reads: "Prior studies have shown inlet coatings such as a halocarbon wax (Yokelson et al., 2003) and SilcoNert 2000 (Pogány et al., 2016) can prevent the adsorption of $NH_3$ and water vapor on instrument sampling surfaces. While current coating technology can provide relatively non-sticky surfaces, we note that in field environments, these

surface treatments can quickly become overcoated with dust, salt, and other condensables, that ultimately compromise their non-stick properties. Continual re-application of a non-stick coating via the active continuous passivation method described here mitigates this issue."

**List of all relevant changes made in the manuscript:**

1. The following text has been added to the abstract for perspective about what a hot aircraft cabin could mean in terms of temperature…"(e.g., average cabin temperatures expected to exceed 30 ºC during summer deployments)".

2. Updates to the wording in the abstract accord to the commenter's and reviewer's suggestions. Specifically, with respect to "Flight-ready" and "custom" being used several times in the text.

3. Addition of the following text to the Sect. 1: While $NH_3$ is regulated under the Gothenburg protocol in some parts of the world (e.g., http://www.unece.org/environmental-policy/conventions/air/guidance-documents-and-other-methodological-materials/gothenburg-protocol.html), it remains an unregulated pollutant in the U.S. (Gilliland et al., 2008)."

4. Sect. 2.2.2 has been amended to include additional details about the inertial inlet.

5. The size of the critical orifice has been added to Sect. 2.2.2.

6. Clarification has been regarding the strain relief in Sect. 2.2.4.

7. Sect. 2.3 has been updated with use of a weighted average to determine an average value and associated uncertainty for the $NH_3$ absorption cross section, which is used to determine the permeation rate of the calibration source. The text in this section has been amended accordingly.

8. The phrase "for the instruments as configured here" has been added to the text on Page 15, Line 19.

9. 'Hydroscopic' was replaced with 'hygroscopic' throughout.

10. The word 'detector' was replaced with QC-TILDAS, instrument, or spectrometer throughout the manuscript according to the reviewer's suggestion.

11. Added a new Sect. 2.4 to include power, weight, and space occupied by the instrument aboard the C-130 as configured for these experiments.

12. Clarification about the overflow of $NH_3$-free air at the inlet was added to Sect. 4.1 by adding the following sentence to the first paragraph: "An overflow > 500 sccm (e.g., the difference between the flow of zero air being supplied to the inlet and the instrument's sample flow) was maintained to ensure that the sample stream was truly $NH_3$-free during this test."

13. We added clarification in Sect. 4.1 about how the measurements were collected for Fig. 3 by adding: "For these experiments, the instrument inlet was continuously overflowed with $NH_3$-free air for the duration of a 3-hour pre-flight exercise prior to take off. Overflowing the inlet was purposefully done to keep the instrument system free of contaminants (e.g., exhaust from other aircraft and ground-based support equipment) prior to sampling in flight."

14. We updated the discussion of the detection limit according to the commenter's suggestions by adding: "We note that the true detection limit of the instrument in flight may be better represented by the full range of variability about the mean zero signal level from the observations in Fig. 3 (e.g., an instrument detection limit of 400 pptv)." to Sect. 4.1.

15. We updated the discussion in Sect. 4.2 about motion sensitivity according to the commenter's suggestions by adding: "It should also be noted that large accelerations in the up-down and fore-aft dimensions are also significant at the onset of vertical ascent. Accelerations measured in the up-down and fore-aft motions at the onset of a 1000 ft/min vertical ascent were measured to be 0.4 g and 0.08 g, respectively. Given the slopes above, these accelerations correspond to a maximum change in $NH_3$ zero signal level of 400 pptv during ascent, which is consistent with the variability in zero signal level observed in Fig. 3 when ascending between constant altitude legs.

16. Discussion about the possibility of using a triple exponential decay to fit the time profile shown in Fig. 5 has been added to Sect. 4.3. Figure 5 has also been updated according to the reviewer's suggestions.

17. Significant updates were made to all of Sect. 4.3 according to the commenter's suggestions. Specifically, we added: "For consistency with the approaches used in the peer-reviewed literature for characterizing the time response of a QC-TILDAS instrument and for ease of comparison to the values reported by Ellis et al. (2010) and Roscioli et al. (2016), we show the results of the bi-exponential fits in Table 1. However, the possibility remains that the time profiles collected here are not perfectly represented by the bi-exponential air-surface exchange model described by Eq. 1. Therefore, we also utilize the observations in Fig. 5 to directly derive the 90% and 99% signal recovery times (denoted as $t_{90, obs}$ and $t_{99, obs}$). In this case, uncertainties reflect the $\Delta t$ spread in the observations associated with the

90±1% and 99±1% signal recovery levels, where ±1% on the signal recovery level corresponds to ±0.5 ppbv for a 50 ppbv step change, which is well within the limit of detection. As indicated in Fig. 5 and Table 1, the instrument time response has a clear dependence with the cleanliness of the instrument sampling surfaces. Specifically, an instrument with "clean" sampling surfaces has a much faster time response ($t_{90,\ obs} < 1$ s) compared to an instrument with "dirty, or "contaminated", sampling surfaces ($t_{90,\ obs} = 143$ s). This effect is apparent regardless of how the $t_{90}$ is determined."

18. The following was added to the end of Sect. 4.4 to address the concerns of the reviewer and the commenters: "All the same, further measurements are recommended for assessing sampling biases that could arise during field measurements of low mixing ratios of $NH_3$ in clean environments following long periods of exposure to near source level concentrations. The potential for an adsorption-related "memory effect" of $NH_3$ (e.g., Williams et al., 1992) on the sampling surfaces following long-term exposure to high concentrations of $NH_3$ is discussed in following sections."

19. The following was added to Sect. 5.1.1 according to suggestions by the reviewer and the commenters: "In this study, the instrument response is rigorously tested with a single step change of $NH_3$ created by turning off a 50 ppbv calibration gas mixture. We note that such large variations in $NH_3$ mixing ratio may not been full applicable to field measurements in unpolluted regions away from concentrated sources of $NH_3$. As described by Ellis et al. (2010), large gradients in $NH_3$ may be less impacted by surface interactions because "clean" sampling surfaces only have a finite number of adsorption sites that could be quickly filled under high $NH_3$ conditions. At lower $NH_3$ concentrations, a greater fraction of $NH_3$ molecules may interact with the inner surfaces. This could explain why passivation did not help to increase the response time of the instrument.".

20. According to the reviewer's suggestions, we also added, "While the proportion of the time response governed by the slow, "adsorptive", term was typically quite low ($D < 10\%$), the magnitude of the step change concentration utilized here is large (e.g., 50 ppb), so caution should be taken when extrapolating these results to ambient observations away from concentrated source regions."

21. We added the following to the first paragraph of Sect. 5.1.3: "We only measured two extreme relative humidity conditions for these tests, even though the relationship of surface interactions may be non-linear and vary greatly depending on the fraction of water vapor added as suggested by Pogány et al. (2016) and Vaittinen et al. (2018)."

22. We added the following to the end of Sect. 5.1.3: "All the same, we reiterate that a caveat of these tests is that the humidity levels tested here may not provide enough information to fully characterize the effects of passivant addition over the full range of dry to humid sampling conditions. Further characterization of the humidity dependence with and without passivant addition is recommended prior to future deployments of this instrument system (or similar QC-TILDAS instruments) in humid field environments.

23. We added the following to the end of Sect. 5.1.4: "Prior studies have shown inlet coatings such as a halocarbon wax (Yokelson et al., 2003) and SilcoNert 2000 (Pogány et al., 2016) can prevent the adsorption on $NH_3$ and water vapor on instrument sampling surfaces. While current coating technology can provide relatively non-sticky surfaces, we note that in field environments, these surface treatments can quickly become overcoated with dust, salt, and other condensables, that ultimately compromise their non-stick properties. Continual re-application of a non-stick coating via the active continuous passivation method described here mitigates this issue. "

24. We updated the language in Sect. 5.2.1 to clarify the observations in Fig. 8. It now read as: "Passivant was not added to the PTR-ToF-MS; active continuous passivation was only applied the QC-TILDAS-based instrument during the selected times described above. It is clear by visual comparison to the PTR-ToF-MS that the non-passivated, "contaminated" QC-TILDAS instrument did not capture all of the expected ambient $NH_3$. This is evident from the differences in measured $NH_3$ mixing ratios reported in Fig. 8 during the time period between 13:20 and 13:23 when the QC-TILDAS was operated without passivant. During this time period the PTR-ToF-MS consistently measured more $NH_3$ than the QC-TILDAS, with the enhancement measured by the PTR during the plume intersect at 13:30 MDT showing an expected mixing ratio of ~45 ppbv. According to PTR-$NH_3$, the integrated $NH_3$ signal during the plume intersect at 13:30 MDT was only 14% less than the integrated $NH_3$ signal measured during the plume intersect at 14:00 MDT, and thus a significant enhancement in $NH_3$ should have been observed by the QC-TILDAS-based instrument. However, the non-passivated, "contaminated" QC-TILDAS-based

instrument measured only a fraction of the NH$_3$ expected during the plume transect at 13:30 MDT, with the only attributable difference being NH$_3$ molecules adsorbing to the sampling surfaces."

25. The following was added to Sect. 5.2.2 according to the commenter's suggestions: "During WE-CAN, the NH$_3$ instrument was typically zeroed between crosswind transects of a wildfire smoke plume when in background air and either just prior to or during turns. The instrument was zeroed every 10-20 mins during transits from Boise to the wildfires sampled with the frequency of zeros depending on the transit time. Zeros measured during WE-CAN research flights were typically collected for a period of 1 to 2 minutes, a duration much greater than the instrument response time, to ensure that zeros were measured well within 90% of the final zero signal level. Prior to each research flight, the NH$_3$ instrument was overflowed with NH$_3$-free air for the duration of a 2-hour pre-flight exercise. Differences in background mixing ratios of NH$_3$ and CO measured before and after the first transect of the smoke plume from the S. Sugarloaf fire are apparent in the magnified timeseries for each in Fig. 10. The differences in NH$_3$/CO ratio observed at 20:14 UTC and 20:25 UTC following in-smoke measurements of NH$_3$ that exceeded 400 ppbv could have resulted from physical differences in plume chemistry, mixing or background composition on either side of the plume, an adsorption-related memory effect in the sample plumbing due to retention of NH$_3$ molecules adsorbed to the sampling surfaces (Williams et al., 1992), or a combination of both. Since the root of the differences are difficult to distinguish and may vary among the WE-CAN research flights, we utilized these differences to characterize the instrument time response given the worst-case scenario that the differences in background observed in Fig. 10 are solely attributed to memory effects on the sampling surfaces. In this worst case, the response time for the NH$_3$ measurement following the plume transect to recover to near background mixing ratio levels observed prior to the plume transect (e.g., 1 ppbv) is roughly 250 s. The time frame most closely resembles $t_{99,obs}$ for the "typical" condition when the instrument is operated with or without passivant. This recovery time and "typical" cleanliness condition are within our expectations for the instrument during this research flight (RF15) since the instrument had routinely been used to sample near source concentrations of NH$_3$ in smoke during several prior consecutive research flights without refreshing the sampling surfaces between flights. "

26. We also amended the text in Sect. 5.2.2 according to the above changes: "A similar time resolution observed for the passivated and non-passivated NH$_3$ measurements is consistent with the sampling surfaces being relatively "clean" or having a "typical" level of cleanliness during this research flight. We also note that only a small fraction of NH$_3$ (<1%) is ejected from the sampling surfaces when passivant was re-added to the NH$_3$ instrument at 20:29 UTC following the second transect, thereby further indicating that only a small amount of NH$_3$ molecules were adsorbed to the instrument sampling surfaces during this flight.

27. We also added the following to the end of Sect. 5.2.2 according to the commenters suggestions: "Since the time response of the CO measurement was limited by its sample flow rate and inlet configuration, we also compare NH$_3$ to acetonitrile (CH$_3$CN) measured by the PTR-ToF-MS. CH$_3$CN is well correlated with NH$_3$ in smoke, and may be more representative of a true 1-Hz tracer owing to operation of the instrument inlet at a flow rate of ~15 SLPM. However, there are no measurements from the PTR-ToF-MS during RF15, the research flight during which the NH$_3$ instrument was systematically tested with and without passivant. Instead, we use measurements of CH$_3$CN from the Bear Trap Fire (RF09) conducted on 09 August 2018 to perform a similar linear regression analysis of fine structure features of measured NH$_3$ versus CH$_3$CN, with CH$_3$CN incrementally averaged up to 5 seconds. We find the best fits result from linear regressions of measured NH$_3$ with the 1-Hz reported and 2-second averaged CH$_3$CN ($R^2$ is > 0.97 and within 0.001 of each other)."

28. The conclusions were updated accordingly.

29. DOI's for the data sets were updated with permanent DOI data links.

30. We added the following acknowledgment: "We thank the two anonymous reviewers as well as Da Pan, Xuehui Guo, and Mark Zondlo for helpful comments and suggestions that have greatly improved this manuscript."

31. (Pogany et al., 2016), (Vaittinen et al., 2018), Williams et al., 1992) and (Yokelson et al., 2003) were added to the references.

32. Table 1 was updated to include $t_{90,obs}$ and $t_{99,obs}$

33. Figure 1a, its figure caption, and relevant parts of Sect. 2.2.3 have been updated with the information requested by the reviewer.

34. Figure 2 has been updated according to the reviewer's suggestions.

35. The caption for Fig. 2 was updated to clarify that the offset was applied to the guidelines in the Allan Variance plots not the data.
36. Fig. 3 was updated to include a time series (Panel d) to illustrate the effects of motion sensitivity on the zero signal level as the aircraft initiates ascent and then levels off at a constant altitude.
37. Fig. 6 was updated to include vertical profiles of potential temperature as an indicator of the boundary layer height.
38. The brown symbols in Fig. 8 were updated to clarify that they are sized by number of head of cattle.
39. CO measurements shown in Figure 10 were updated following a release of updated final data (version R1) from the WE-CAN field campaign. Traces in the figure were also re-ordered to highlight the differences between $NH_3$ and CO measurements. The conclusions drawn from Figure 10 remain unchanged.
40. Fig. 10 was updated with magnified views of the $NH_3$ and CO measurements to highlight differences in background measured before and after transect of a concentrated smoke plume.
41. All figure captions were updated according to the modifications listed above.

[revised manuscript text omitted]